# IRSp53 controls plasma membrane shape and polarized transport at the nascent lumen in epithelial tubules

Sara Bisi[1,6], Stefano Marchesi[1,6], Abrar Rizvi[1,6], Davide Carra [1], Galina V. Beznoussenko[1], Ines Ferrara[2], Gianluca Deflorian[3], Alexander Mironov[1], Giovanni Bertalot[4], Federica Pisati[3], Amanda Oldani[1], Angela Cattaneo[3], Ghazaleh Saberamoli[1], Salvatore Pece[4,5], Giuseppe Viale[4], Angela Bachi [1], Claudio Tripodo[1,2], Giorgio Scita [1,5 ✉] & Andrea Disanza [1]

It is unclear whether the establishment of apical–basal cell polarity during the generation of epithelial lumens requires molecules acting at the plasma membrane/actin interface. Here, we show that the I-BAR-containing IRSp53 protein controls lumen formation and the positioning of the polarity determinants aPKC and podocalyxin. Molecularly, IRSp53 acts by regulating the localization and activity of the small GTPase RAB35, and by interacting with the actin capping protein EPS8. Using correlative light and electron microscopy, we further show that IRSp53 ensures the shape and continuity of the opposing plasma membrane of two daughter cells, leading to the formation of a single apical lumen. Genetic removal of IRSp53 results in abnormal renal tubulogenesis, with altered tubular polarity and architectural organization. Thus, IRSp53 acts as a membrane curvature-sensing platform for the assembly of multi-protein complexes that control the trafficking of apical determinants and the integrity of the luminal plasma membrane.

[1] IFOM, the FIRC Institute of Molecular Oncology, Via Adamello 16, 20139 Milan, Italy. [2] Department of Health Sciences, Human Pathology Section, University of Palermo School of Medicine, Via del Vespro 129, 90127 Palermo, Italy. [3] Cogentech, S.R.L., Via Adamello 16, 20139 Milan, Italy. [4] European Institute of Oncology (IEO) IRCCS, Via Ripamonti 435, 20141 Milan, Italy. [5] Department of Oncology and Haemato-Oncology, University of Milan, Via Santa Sofia 9/1, 20122 Milan, Italy. [6] These authors contributed equally: Sara Bisi, Stefano Marchesi, Abrar Rizvi. ✉email: Giorgio.scita@ifom.eu

Many internal epithelial organs consist of a polarized cell monolayer that surrounds a central apical lumen. Polarization requires interactions between the signaling complexes and scaffolds that define cortical domains with membrane-sorting machinery[1]. This architectural organization and the morphogenesis processes that underlie it can be reproduced by plating cells on pliable matrigel in a matrigel-containing medium that provides the mechanochemical cues for formation of a polarized hollow sphere (cyst). The de-novo apical–basal polarity arises from successive divisions of a single, nonpolarized cyst-forming cell[2,3]. The first symmetry-breaking event occurs at the first cell division, when the midbody is formed[4]. Around the midbody, the apical membrane initiation site (AMIS) is assembled, which establishes the location of the nascent lumen. Assembly of the AMIS is mediated by both microtubules and branched actin filaments, which promote recruitment and anchoring of vesicles[5]. Transmembrane proteins, such as podocalyxin (PODXL; classical apical marker, also known as GP135) and Crumbs3, are transcytosed from the plasma membrane facing the extracellular matrix (ECM) toward the first cell–cell contact site[6]. This occurs via RAB11-RAB8 endo/exosomes trafficking and through a direct anchoring with RAB35 at the AMIS[6,7]. Thus, coordination between actin cytoskeletal dynamics and membrane trafficking is essential for the initiation of a polarized central lumen.

The molecular players that link the spatially restricted actin dynamics and the polarized delivery of endosomal vesicles remain incompletely understood. What determines the shape of the opposing apical membrane during the initial phase of lumenogenesis is also not clear. Proteins that can sense and shape the curvature of the plasma membrane at the AMIS and physically link it with the underlying cytoskeleton will be critical here. We postulated that IRSp53 can fulfill this function.

IRSp53 regulates the interplay between the plasma membrane and the actin cytoskeleton during directional migration and invasion of cells[8–11]. Accordingly, IRSp53 localizes at the tips of both filopodia and lamellipodia[12]. At these sites, IRSp53 senses and promotes membrane curvature through its I-BAR domain, and acts as an effector of either CDC42 or RAC1 GTPases by ensuring the recruitment of a number of actin regulatory proteins[8,9,13–17]. These proteins include the nucleator-promoting WAVE complex, essential for branched polymerization of actin, and the linear actin elongators VASP and mDia1, which together with EPS8 (an actin capping and cross-linking protein[18]) initiate and promote the extension of filopodia[8,19,20]. Through its I-BAR domain, IRSp53 undergoes a phase separation that facilitates protein clustering[21] and the recruitment of Ezrin, a member of the ezrin-radixin-moesin (ERM) protein family, that links the actin cortex to the cell membranes[22]. The subsequent binding of activated CDC42 to this cluster leads to inhibition of the weak capping activity of IRSp53. It further promotes a structural change in IRSp53 that facilitates the interaction of IRSp53, via its now-liberated SH3 domain, with actin linear elongators and cross-linkers to support the growth of filopodia[8,23]. IRSp53 has also been involved in the assembly of cell–cell and cell–ECM adhesions downstream of "polarity-regulating kinase partitioning-defective 1b" (PAR1B), and in the polarized architectural organization of Madin Darby canine kidney (MDCK) epithelial cells in vitro[24,25]. The molecular pathways and interactors IRSp53 uses in these processes and their physiological relevance at the organism level, however, remains poorly understood.

Here, we investigated these issues using three-dimensional cultures of renal MDCK and intestinal Caco-2 cysts, as well as genetically modified zebrafish and mice. Our data indicate that IRSp53 is pivotal for the assembly of the RAB35–IRSp53–EPS8 macromolecular complex. We further show that this complex is necessary for correct kidney morphogenesis and lumenogenesis, through its control of polarized transport and the shape of the apical plasma membrane.

## Results

**IRSp53 localizes to the apical lumen in vivo and in vitro.** To study the physiological role of IRSp53 in epithelial morphogenesis, we examined its expression (Supplementary Fig. 1A) and localization in several epithelial tissues characterized by glandular or tubular structures (Fig. 1a–d). Invariably, IRSp53 localized to the luminal-facing side of the human epithelial cells in human (Fig. 1a, b), adult mice (Fig. 1c), and in the pronephric ducts of developing zebrafish (Fig. 1d). We confirm the apical localization of endogenous IRSp53, as well as of expressed GFP-IRSp53 in two-dimensional (2D) monolayers and 3D cysts of MDCK (Fig. 2a and Supplementary Fig. 2A) and colorectal adenocarcinoma (Caco-2) epithelial (Supplementary Fig. 2B–C) cells in vitro.

**The loss of IRSp53 disrupts cyst morphogenesis.** To determine whether IRSp53 is involved in the establishment of apical–basal polarity and lumen formation, IRSp53 was depleted in MDCK cells using CRISPR/Cas9, shRNA[24], and siRNAs, and in Caco-2 cells using CRISPR/Cas9. Single MDCK-control and IRSp53-depleted cells were seeded onto Matrigel for 6 days to allow the generation of hollow cysts[1,26]. As expected, most of the control cysts (>75%) developed a single, actin-positive central lumen that was surrounded by an epithelial monolayer, which was decorated by apical proteins, such as PODXL (Fig. 2b). Conversely, <50% of the MDCK IRSp53-silenced cysts formed a single apical lumen, while the remaining showed either multiple or aberrant lumens (Fig. 2b). A significant increase in multi-luminal aberrant cysts was also seen in two independently generated Caco-2 cell CRISPR–IRSp53-KO clones, as compared to the control Caco-2 cells (Fig. 2c). The aberrant multi-luminal phenotype was rescued upon re-expression of GFP-IRSp53 in both the IRSp53-silenced MDCK and Caco-2 IRSp53-KO cells (Fig. 2c). Thus, IRSp53 has a crucial role in the morphogenesis of these epithelial organoids.

**IRSp53 controls the distribution and trafficking of PODXL.** Next, we examined the localization and dynamics of IRSp53 in the very early phases of cystogenesis (Fig. 3a)[6]. In the first 8–12 h after plating of the control cells, PODXL localized to the peripheral surface of the cells, before accumulating in vesicular-like, recycling structures, as previously shown[7,27–31]. Subsequently, PODXL was delivered to the opposing membrane between the two daughter cells, where the apical lumen was formed de novo (Fig. 3a, b). IRSp53 became prominently enriched along the newly formed, opposing intercellular membrane well before PODXL recruitment (Fig. 3b, Supplementary Movie 1). At later stages, IRSp53 and PODXL were concentrated at the AMIS, and when the cyst was formed, IRSp53 and PODXL co-stained along the lumen (Fig. 3b). In the early phases of cystogenesis, IRSp53 and PODXL also co-localized in vesicle-like structures (Fig. 3b), which, after forming at the peripheral plasma membrane, were subsequently internalized and targeted to the AMIS and the nascent lumen (Fig. 3c–d, Supplementary Movies 2 and 3). These findings suggest that IRSp53 and PODXL co-traffic to or from endosomal recycling compartments and that IRSp53 controls the trafficking or targeting of PODXL to the AMIS.

To distinguish among these possibilities, we analyze the cellular localization of GFP-IRSp53 relative to a variety of proteins that mark distinct trafficking compartments. GFP-IRSp53, which showed a distribution identical to that of the endogenous protein

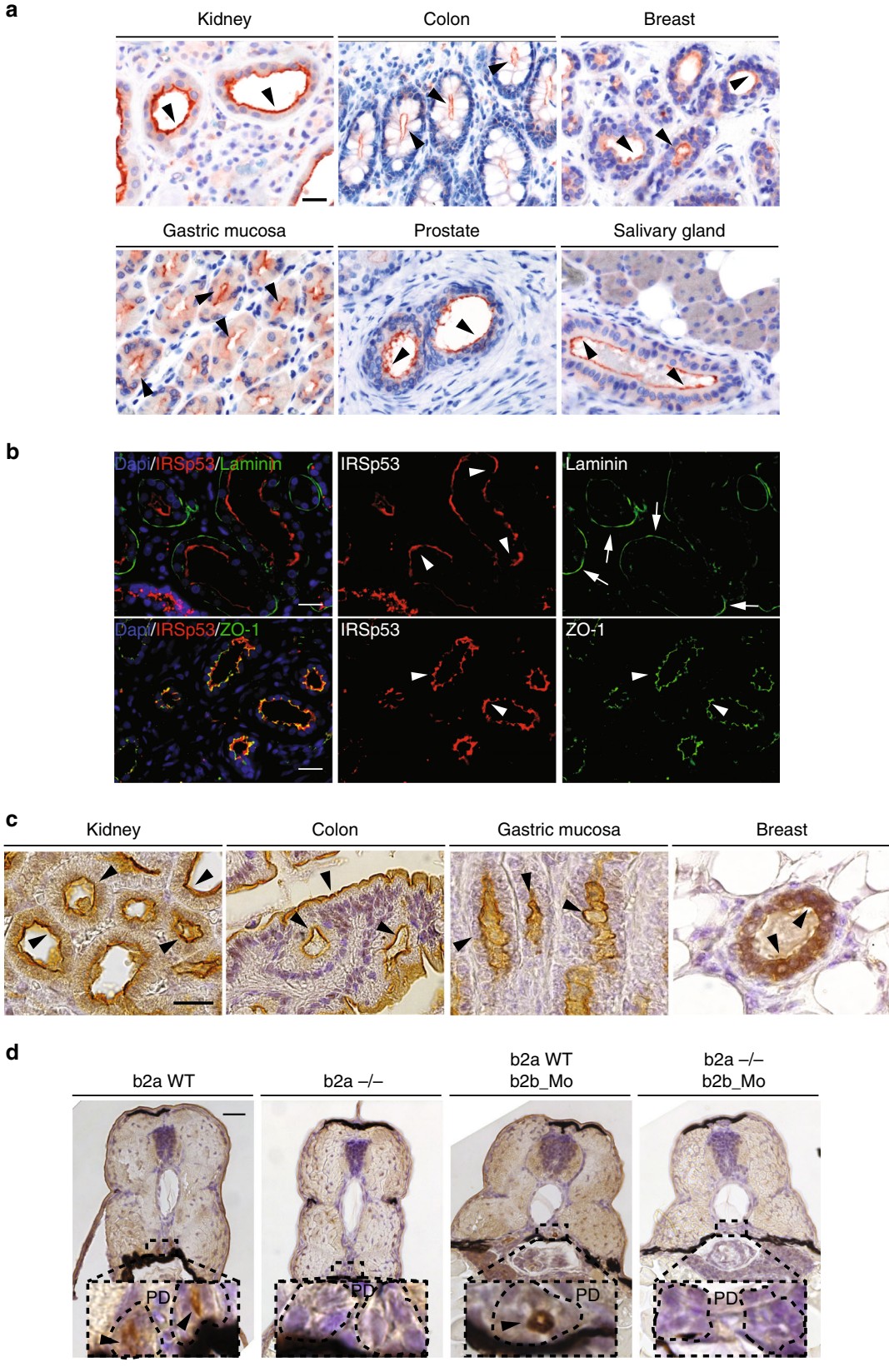

(Fig. 2a, and Supplementary Fig. 2A), partially localized with PODXL and actin in vesicle-like structures (Fig. 4a). These structures were also positive for RAB7, RAB11a, and RAB35 (Fig. 4a), but not for the early and late endosome markers "early endosome antigen 1" (EEA1) and "lysosome-associated

membrane glycoprotein 1" (LAMP1), respectively, or the Golgi marker Giantin (Supplementary Fig. S2D).

To determine whether IRSp53 is involved in the control of PODXL trafficking, we took advantage of the observation that PODXL is internalized from the apical membrane and undergoes

**Fig. 1 IRSp53 is restricted to the luminal side in various epithelial tissues of vertebrates. a** IHC analysis of IRSp53 expression and localization in the indicated human tissues and organs. Arrowheads indicate the apical/luminal enrichment of IRSp53. Scale bar, 50 μm. **b** IF analysis of human kidney. Samples were stained with anti-IRSp53 (red), anti-Laminin (green) and DAPI (upper panels) or with anti-IRSp53 (red) anti-ZO-1 (green) and DAPI (lower panels). Arrowheads and arrows indicate the apical/luminal localization of IRSp53 and ZO-1, or the basal enrichment of Laminin, respectively. Scale bar, 50 μm. **c** IHC analysis of the expression and localization of IRSp53 in the indicated murine tissues and organs. Arrowheads indicate the apical/luminal enrichment of IRSp53. Scale bar, 20 μm. **d** IHC analysis of Baiap2a and Baiap2b expression and localization in zebrafish embryo (Inset: PD) at 72 hpf in the indicated genetic backgrounds: *baiap2a wild-type* (b2a WT), *baiap2a* mutant (b2a −/−), *baiap2b* morphant (b2a WT b2b_Mo) and *baiap2a* mutant *baiap2b* morphant (b2a −/− b2b_Mo). Arrowheads indicate the apical/luminal enrichment of IRSp53 in the pronephric duct. Scale bar, 50 μm (insets, 200 μm).

trafficking to the so-called vacuolar apical compartment (VAC) upon calcium removal in MDCK polarized monolayers[32]. To verify that PODXL is internalized under these conditions, MDCK cell monolayers were incubated with a PODXL antibody that recognized the extracellular portion of PODXL in vivo. Detection of PODXL with a secondary fluorescence-conjugated anti-PODXL IgG after stripping of the cell surface with acid washes revealed that PODXL was internalized in the VACs (Supplementary Fig. 3A). Of note, IRSp53 and PODXL colocalize at VACs upon calcium removal in polarized monolayer (Supplementary Fig. 3B). The removal of IRSp53 significantly delayed the relocalization of PODXL to the VACs, in comparison to the control cells (Supplementary Fig. S3C–E).

If IRSp53 is required for correct trafficking or anchoring of PODXL in the early phases of cystogenesis, its ablation should lead to mislocalization of PODXL. Consistently, while in control cells PODXL was enriched in a single, centrally-located luminal spot at the four-cell stage, in the *IRSp53*-KO and knocked-down cells, we detected two or more PODXL-positive foci (Fig. 4b, c). Of note, the majority of *IRSp53*-silenced, early phase, multi-luminal cysts retained an apically-restricted distribution of PODXL. However, a significant, albeit small, proportion of the cysts showed an inverted apical–basal polarity that was characterized by mislocalization of PODXL to the outer, ECM-facing, plasma membrane (Fig. 4b, c). We performed a similar experiment also in *IRSp53*-KO Caco-2 cells by monitoring the distribution of aPKC, which is the most commonly-used apical marker in this cell system[26,33,34]. Likewise in MDCK cells, IRSp53 depletion resulted in formation of aPKC multi-foci at the 3–4-cell stage during cystogenesis, as compared to the control Caco-2 cells (Supplementary Fig. 4A).

Collectively, these findings indicate that IRSp53 is implicated in the trafficking and/or correct targeting of apical proteins, and specifically of PODXL, to the de-novo forming lumen.

**IRSp53 coordinates RAB35 and EPS8 pathways in lumen formation.** The RAB GTPases are key regulators of PODXL transcytosis[35]. Among these, we focused our attention on RAB35. RAB35 is involved in clathrin-mediated endocytosis[36] and was recently shown to localize early at the AMIS, where it serves as a physical anchor for the targeting of PODXL in MDCK cell cystogenesis[7,35]. Of note, the complete genetic loss of RAB35 was shown to lead to a complete but transient inversion of polarity accompanied by accumulation of PODXL in intracellular vesicles and only subsequently to the formation of multilumen[32]. Conversely, a partial reduction of its expression caused the formation of multiple lumen[10]. Collectively, these findings indicate that tampering with RAB35 activity impact on cystogenesis, PODXL trafficking and establishment of apico-basal polarity.

During the initial phases of cyst development, IRSp53 and RAB35 were enriched along the opposing cell–cell membrane prior to the arrival of PODXL (Fig. 3b and Supplementary Fig. 4B). At the later stages, IRSp53, RAB35, and PODXL were concentrated at the AMIS (Fig. 3b and Supplementary Fig. 4B). IRSp53, RAB35, and PODXL also colocalized in intracellular

vesicle-like structures (Figs. 3b and 4a, and Supplementary Fig. 4B). More relevantly, silencing of RAB35, as previously reported[7,35] and partly reproduced by IRSp53 depletion, resulted in the complete inversion of the apical–basal polarity and in the formation of multi-lumen cysts (Fig. 4c). The penetrance of polarity inversion caused by the silencing of IRSp53 (Fig. 4b) is almost as robust as in the case of RAB35 loss. This is, however, expected since IRSp53 only partially affects RAB35 localization and activity and further impacts on other independent pathways (see below). Stated differently, the loss of IRSp53 mimics a partial loss of RAB35 function in epithelial polarity. Notably, IRSp53 retained its localization to the AMIS in RAB35-silenced cells that showed inverted PODXL polarity (Supplementary Fig. 4C).

To determine whether IRSp53 and RAB35 interact physically, co-immunoprecipitation experiments were performed. Here, ectopically expressed IRSp53 interacted with RAB35 only upon serum starvation, which suggested that IRSp53 might preferentially associate with the inactive GDP-bound form of RAB35 (Fig. 5a). This possibility was verified using the recombinant purified dominant-negative RAB35S22N and constitutively active RAB35Q67L mutants of RAB35, and the recombinant purified RAB35WT, loaded either with GDP or GTPγS, in in vitro pull down and overlay assays. Here, IRSp53 bound directly, albeit weakly to RAB35 and, surprisingly, showed higher apparent affinity for inactive RAB35S22N or RAB35WT-GDP compared to the RAB35Q67L or RAB35WT-GTP forms respectively (Supplementary Fig. 4D, E). Mapping of the interaction surfaces showed that the I-BAR domain of IRSp53 is necessary and sufficient for this association with RAB35S22N (Fig. 5b). In addition, an IRSp53 mutant that lacked these critical positively charged residues (the I-BAR-4K domain mutant; IRSp53 I-BAR*: K142E, K143E, K146E, K147E[37]) did not interact with RAB35S22N under in vitro binding or co-immunoprecipitation conditions (Fig. 5c, d).

The direct interaction between IRSp53 and RAB35 suggested that these two proteins might function together in the regulation of PODXL localization during cystogenesis. Thus, perturbation of their association should affect PODXL trafficking to the AMIS and lumenogenesis. To test this, we lentivirally transduced IRSp53 mutants in each of its critical domains (i.e., I-BAR, CRIB, SH3, PDZ-binding domains)[10] into *IRSp53*-silenced MDCK cells and *IRSp53*-KO Caco-2 cells and performed cystogenesis assays to examine PODXL and aPKC localization. Expression of the I-BAR-4K mutant, which retains its structural integrity[37] but does not bind to RAB35 failed in rescuing the localization of PODXL to a single spot at the AMIS in these MDCK cells (Fig. 5e). This mutant was still present on the apical lumen, albeit its localization was more diffuse and less restricted than wild-type (WT) IRSp53 (see also below). Similar findings were obtained upon expression of I-BAR-4K in Caco-2 *IRSp53*-KO cells (Supplementary Fig. 5A). Furthermore, in the Caco-2 *IRSp53*-KO cells, analysis of the IRSp53 mutants showed that the I-BAR and SH3 domains were essential for correct IRSp53 localization at the apical lumen (Fig. 6a). Thus, IRSp53 required its functional I-BAR domain for its role in lumen formation in

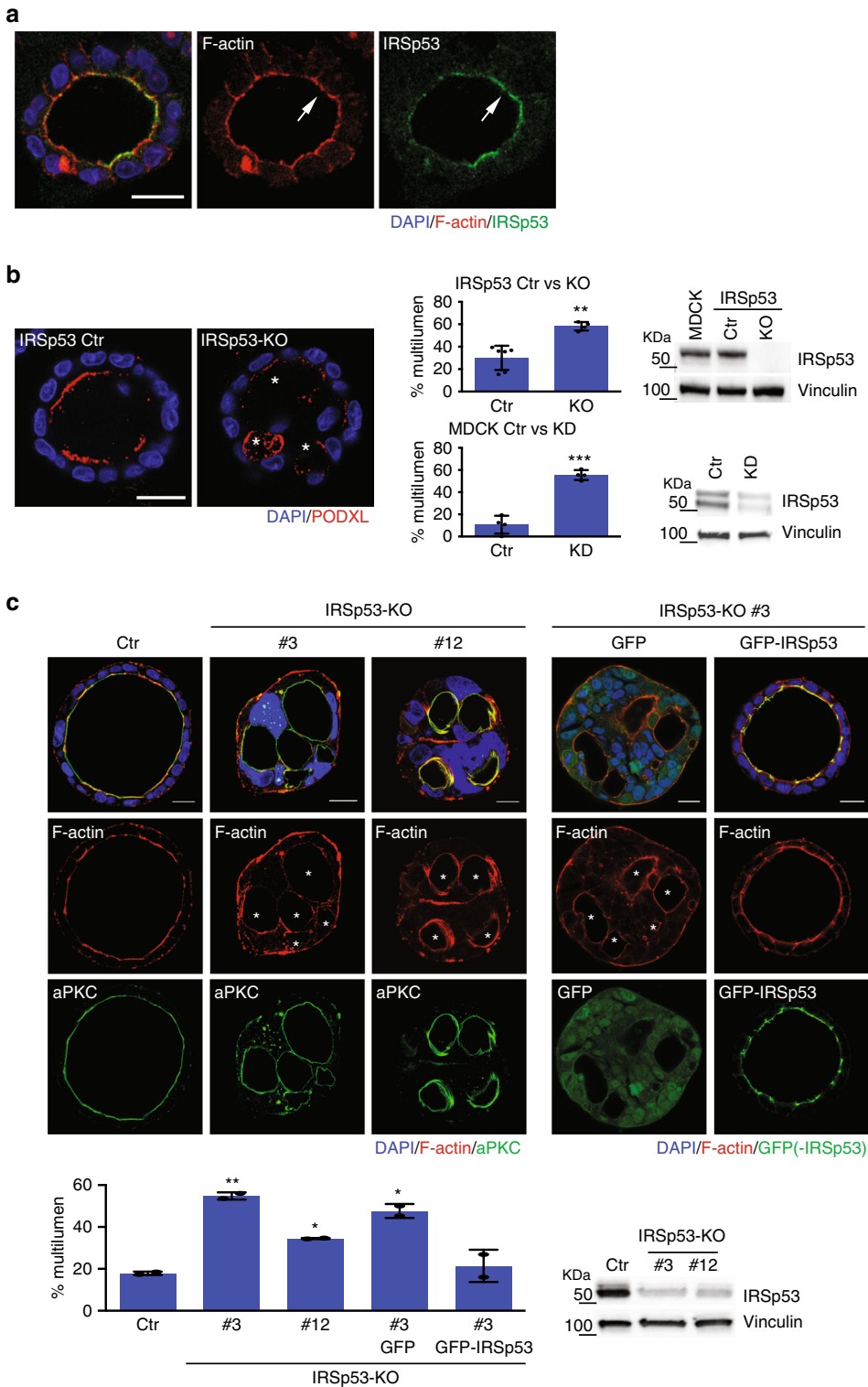

these epithelial cells. Molecularly, both its binding to negatively curved, phosphatidylinositol 4,5-bis phosphate (Ptdins(4,5)P$_2$)-rich membranes (which is compromised by the mutations inserted[37]), and its interactions with RAB35 are likely to account for these findings. Consistently, *IRSp53* ablation impaired the localization of RAB35 (Fig. 6b) and of RAB35S22N (Supplementary Fig. 4F) at the nascent AMIS during the early phases of cystogenesis. Caution should, however, be used in interpreting the finding with RAB35S22N since its expression also perturbs the correct formation of a lumen and PODXL anchoring to the

AMIS[7]. This notwithstanding, IRSp53 acts as an assembly platform that drives the correct localization of RAB35, and possibly its activation, at this site. To directly address this, we measured the amount of GTP-loaded RAB35 using immobilized RAB35-binding domain of RUSC2[38]. The loss of IRSp53 reduced the levels of active RAB35 (Fig. 6c). Altogether, these findings point to a role of IRSp53 in promoting the optimal localization and activity of RAB35.

The structure-function analysis further indicated the requirement of additional domains of IRSp53 for correct cyst

**Fig. 2 IRSp53 removal increased the formation of cysts with multiple lumens. a** Three days old MDCK epithelial cysts were fixed and stained an anti-IRSp53 antibody (green), and rhodamine phalloidin (red) to visualized F-actin and DAPI (blue). Arrows indicate IRSp53 and F-actin at the luminal side of the cyst. Scale bar, 18 μm. **b** Five days old 3D cysts of MDCK control cells (IRSp53 Ctr) or IRSp53-KO obtained by CRISPR/Cas9, or MDCK control (MDCK Ctr) or IRSp53-KD (not shown). Left: the cysts were stained with anti-podocalyxin (PODXL, red) and DAPI (blue). Asterisks, multiple lumens in IRSp53-KO cyst. Scale bar, 18 μm. Central: quantification of cysts with multiple lumens in IRSp53 Ctr vs IRSp53-KO (Top) or IRSp53 Ctr vs IRSp53-KD (Bottom). Data are expressed as means ± SD. At least 20 cysts/experiment were analysed in $n = 6$ (IRSp53 Ctr), $n = 4$ (IRSp53-KO) or $n = 4$ (IRSp53 Ctr and IRSp53-KD) independent experiments. $P$ value, student's $t$-test two-tailed. **$p < 0.01$; ***$p < 0.001$. Source data are provided as a Source Data file. Right: IRSp53 and vinculin protein levels. Mw Molecular weight markers. **c** Top left: 3D cysts embedded into Matrigel/ collagen matrix of Caco-2 control (Ctr) or IRSp53-KO clones #3 and #12 cells, obtained by CRISPR/Cas9 were stained with an anti-atypical-PKC antibody (a-PKC, green), rhodamine-phalloidin to detect F-actin (red) and DAPI (blue). Asterisks, multi-lumens in IRSp53-KO cyst. Scale bar, 10 μm. Top right: 3D cysts from IRSp53-KO clone #3 stably-infected to express GFP or murine GFP-IRSp53 cells were processed for epifluorescence to visualize GFP or GFP-IRSp53 and stained with rhodamine-phalloidin to detect F-actin (red) and DAPI (blue). Asterisks, multiple lumens in IRSp53-KO GFP cyst. Scale bar, 10 μm. Lower left: Quantification of cysts with multiple lumens. Data are expressed as mean ± SD. At least 25 cysts/experiment were analysed in $n = 2$ independent experiments. $P$ value, student's $t$-test two-tailed. *$p < 0.05$; **$p < 0.01$. Source data are provided as a Source Data file. Lower right: IRSp53 and vinculin protein levels (for GFP and GFP-IRSp53 levels-Supplementary Fig. S5A). Residual expression of IRSp53 in clones #3 and #12 is due to the aneuploidy of Caco-2 cells (see "Methods").

morphogenesis. Indeed, IRSp53 mutations in either the CRIB-PP region (required for IRSp53 binding to active CDC42[8,9,16,23]) or the SH3 domain (which mediates interactions with a variety of IRSp53 binding partners[8,9,15,16,20,23,39]) failed to rescue the defective distribution of PODXL in MDCK cells and lumen formation in Caco-2 cells, respectively, that arose from the IRSp53 ablation (Fig. 5e and Supplementary Fig. 5A). Noticeably, impairing the function of the SH3 domain also altered the luminal localization of IRSp53 (Fig. 6a). This was particularly evident in the Caco-2 cysts. In this system, IRSp53 showed a restricted, although evenly distributed, localization along the apical, luminal side. Conversely, an IRSp53-SH3-defective mutant focally accumulated to tight junctions, where it colocalized with ZO-1, with its continuous apical distribution lost (Fig. 6a). These results indicate that beside the I-BAR interaction with RAB35, SH3 interactors are also likely to be important in coordination of IRSp53 localization and activity. Among these, the actin capping and bundling protein EPS8 was previously shown to form a stoichiometrically stable complex with IRSp53 in vivo[9]. Thus, we investigated the involvement of EPS8 together with IRSp53 in orchestration of the correct lumenogenesis program. Here, we show that: (a) EPS8, like IRSp53, is recruited early at the AMIS (Supplementary Fig. 5B) and has an apically restricted luminal localization in MDCK cells (Supplementary Fig. 5C) and Caco-2 cysts (Supplementary Fig 5D); (b) loss of EPS8 in Caco-2 cells leads to the formation of cysts with multiple lumens and, at less extent, inverted of polarity (Supplementary Fig. 6A, B); (c) EPS8 is required for the correct localization of IRSp53 but not the opposite (Fig. 6d and Supplementary Fig. 6D).

**IRSp53 shapes the plasma membrane at nascent lumens.** Our findings argue that IRSp53 controls epithelial tissue polarity and lumenogenesis in vitro through the regulation of polarized trafficking and by acting as a scaffold for the assembly of apically restricted RAB35 and EPS8 complexes (Fig. 6e). However, IRSp53 can also sense and control membrane curvature and local phospholipid composition through its I-BAR domain, which will ultimately impact on the ultrastructural organization and shape of the plasma membrane at the nascent AMIS. To investigate this possibility, we performed correlative light and electron microscopy analysis in the very early phases of cystogenesis, which was monitored by the appearance at the AMIS of β-catenin and, more importantly, of PODXL (Fig. 7a, Supplementary Figs. 7 and 8). We focused on the two-cell stage of MDCK cells plated onto a Matrigel cushion. Comparison between the MDCK control (Ctr) and the IRSp53-KO cells showed that loss of IRSp53 delays or blocks the elimination of cytoplasmic bridges between newly formed apical domains at the AMIS (Fig. 7a, b) where IRSp53 is

relocalized even before PODXL (Fig. 2b, Supplementary Fig. 4B and Fig. 7). Also, albeit less frequently, this impacted on the intercellular luminal space (Fig. 7a), and resulted in the formation of PODXL-positive ectopic lumens (cytoplasmic membrane vacuoles with features of apical domains) (Supplementary Fig. 8). Of particular significance, there was the occasional but evident reduction of intercellular space and a significant increase in the number of "cytoplasmic bridges" that interrupted the separation of the luminal (apical) plasma membrane of two adjacent IRSp53-KO cells (Fig. 7a, b). These were also visible in tomographic 3D reconstructions (Fig. 7B; Supplementary Movie 4). Thus, IRSp53 ensures the integrity and structural organization of the plasma membrane at the AMIS. The increased number of membrane interconnections, caused by the loss of IRSp53 might generate distinct "mini-lumens" and plasma membrane targeting sites. These defects together with altered polarized trafficking of PODXL carriers, also enriched in IRSp53 (Supplementary Fig. 7), might lead the accumulation of PODXL in multiple foci, which would eventually evolve into multiple lumens.

**IRSp53 is required for kidney development and morphology.** To determine the functional activity of IRSp53 is physiologically relevant in living organisms, zebrafish lines were generated that carry a mutation in the fish paralog, *baiap2a* (b2a), and morpholinos were used to ablate *baiap2b* (b2b) (see below). Moreover, we analyzed tissue organization in *IRSp53*-KO mice. In both of these models, IRSp53 showed prominent expression and peculiar luminal localization in the kidneys. In the renal tubules of adult mice, IRSp53 was expressed and enriched at the luminal side (Fig. 1c). A similar, restricted pattern of luminal apical expression was seen in early-stage pronephric ducts in developing fish embryos (Fig. 1d). The analysis was therefore focused on this organ, and specifically on its tubular architecture. In zebrafish, gene duplication resulted in the generation of two distinct, but highly related, IRSp53 paralogs known as *baiap2a* (b2a) and *baiap2b* (b2b). These two gene products are expressed at variable levels in the developing embryo, with b2a as the more expressed (Supplementary Fig. 1B). A mutant zebrafish line was generated in which the gene product encoding for b2a was disrupted through introduction of a chemically-mediated nonsense mutation (Supplementary Fig. 1C), with that for b2b targeted using a mixture of splice and translation blocking morpholinos (Supplementary Fig. 1C). We also characterized an antibody that can recognize both b2a and b2b, at least when expressed in mammalian cells (Supplementary Fig. 1D). The loss of b2a in the mutant line caused a large reduction in the expression of IRSp53 in the pronephric ducts, which was consistent with b2a being

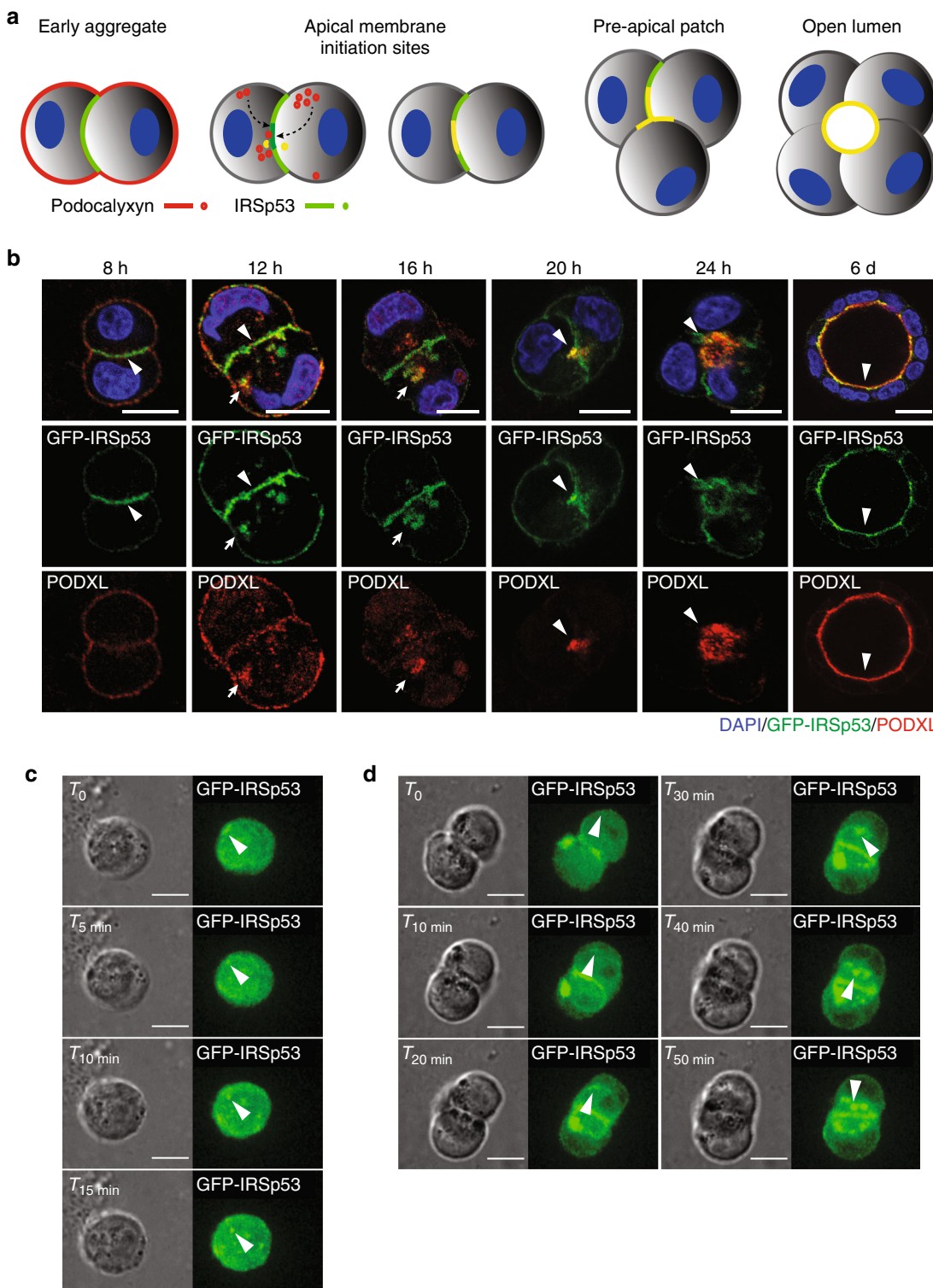

**Fig. 3 IRSP53 precedes PODXL localization to the apical membrane. a** Schematic representation of IRSp53 and PODXL trafficking during the early phases of lumenogenesis. **b** MDCK cells expressing GFP-IRSp53 were seeded as single cells on a Matrigel layer and left to grow to form three-dimensional (3D) cysts. The cysts were fixed at the indicated time points, processed for epifluorescence to visualize GFP-IRSp53 (green) and stained with anti-PODXL (red) and DAPI (blue). Arrowheads, IRSp53 preceding PODXL relocalization at the AMIS at early time points; colocalization of IRSp53 and PODXL at the AMIS at later time points; and enrichment at the luminal side in the mature cyst. Arrows, IRSp53 and PODXL colocalization in vesicle-like structures. Scale bar, 18 μm. **c**, **d** Still images of time lapse of GFP-IRSp53 during early phases of cystogenesis. MDCK expressing GFP-IRSp53 were seeded as single cells on a Matrigel. 6 h after seeding cells were subjected to time-lapse analysis using Confocal Spinning Disk microscope. Images (Bright Field and GFP channel respectively) were acquired every 5 min for 15 h (see also Supplementary Movies 2 and 3). White arrowheads indicate GFP-IRSp53 vesicular-like structures that emerge from the peripheral plasma membrane and move toward the inner (**c**) or the forming apical side at cleavage furrow immediately after the first cell division (**d**). Scale bar, 10 μm.

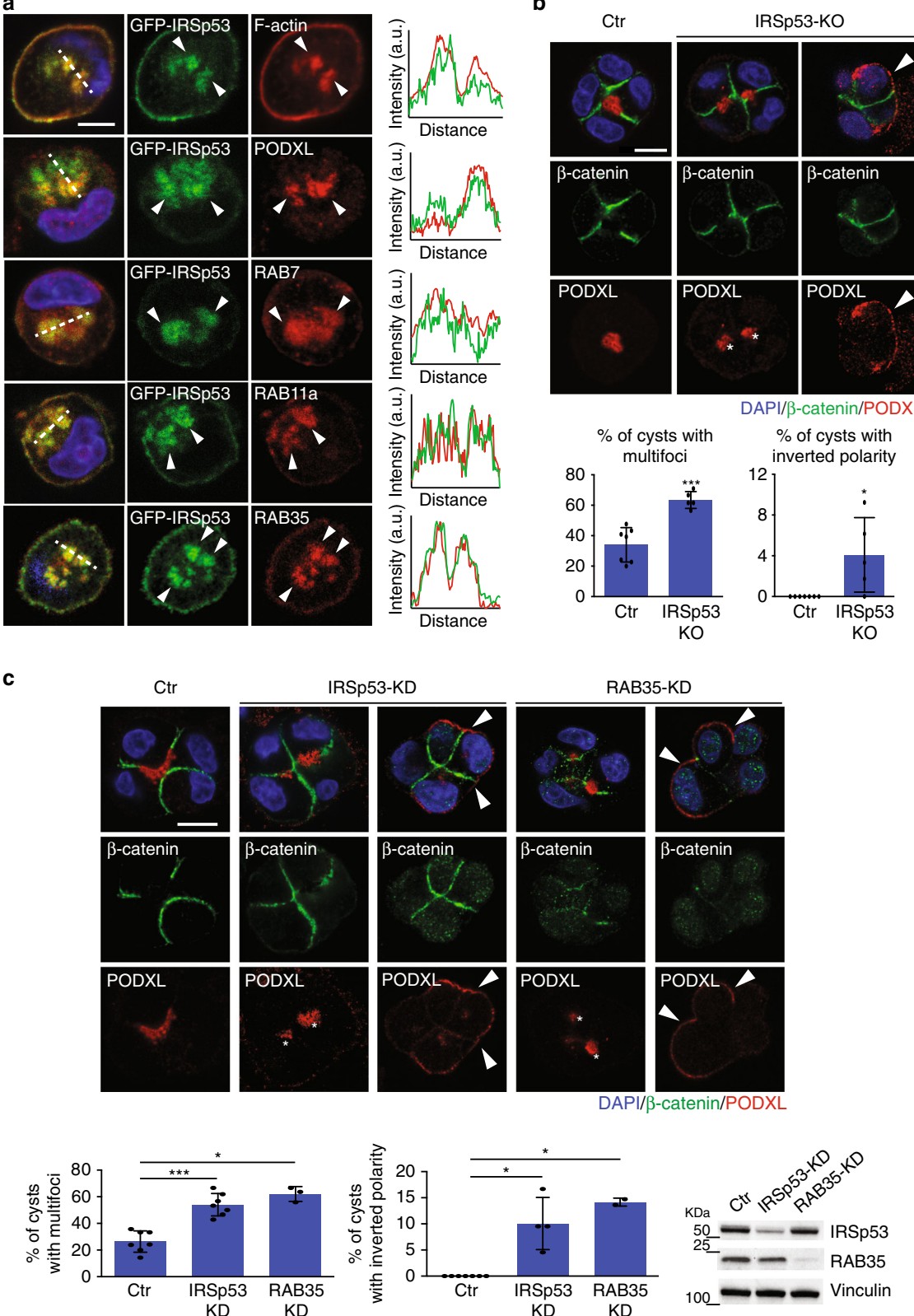

more expressed than b2b, albeit we cannot rule out that b2b was also present (Fig. 1d).

To visualize the pronephric duct during development, the WT (b2a WT) and mutant b2a (b2a −/−) lines were crossed with Tg (CldnB::GFP). This transgenic line expresses EGFP under the control of the promoter of ClaudinB, which is localized in various epithelial structures, including the developing kidneys[40]. Individual depletion of the single *baiap2* genes, so in the b2a mutant line or by injection of the b2b-morpholino in WT embryos, did not interfere with normal pronephric duct development (Fig. 8a, Supplementary Fig. 9A, B and Supplementary Fig. 10A). However, injection of b2b morpholinos in the b2a mutant lines caused

**Fig. 4 Loss of IRSp53 alters PODXL trafficking at the AMIS and apical–basal polarity. a** Left: MDCK cells, expressing GFP-IRSp53, were mock transfected or transfected with Apple-RAB11a or RFP-RAB35, and seeded as single cells on Matrigel and fixed after 5 h of growth. Cysts were processed for epifluorescence to visualize GFP-IRSp53 (green) and Apple-RAB11a or RFP-RAB35, or stained with rhodamine-phalloidin (red), anti-PODXL (red) or anti-7 (red), and DAPI (blue). Arrowheads, IRSp53 colocalization with trafficking markers. Scale bar, 10 μm. Right: Intensity profiles of the green and red channels over the vesicular structures (dashed lines in the merge channels). **b** Top: MDCK cell control (Ctr) and IRSp53-KO were seeded as single cells on Matrigel and grown for 24/36 h. The cysts were stained with anti-β-catenin (green) and anti-PODXL (red) antibodies, and DAPI (blue). Asterisks, PODXL multi-foci in IRSp53-KO cyst. Arrowheads, PODXL staining at the basal membrane in IRSp53-KO cyst. Scale bar, 10 μm. Bottom: Quantification of multi-foci cysts (left) and inverted polarity cysts (right). Data are means ± SD. Four-cell stage cysts were analysed, as at least 20 cysts/experiment in $n = 7$ (Ctr) and $n = 5$ (IRSp53-KO) independent experiments. P value, student's t-test two-tailed (multi-foci) or one-tailed (inverted polarity). *$p < 0.05$; ***$p < 0.001$. Source data are provided as a Source Data file. **c** Top: MDCK control (Ctr), IRSp53-KD or RAB35-KD cells were seeded as single cells on a Matrigel and grown for 24/36 h. The cysts were fixed and stained with anti-β-catenin (green) and anti-PODXL (red) antibodies, and DAPI (blue). Asterisks, PODXL multi-foci in IRSp53-KD and RAB35-KD cysts. Arrowheads, PODXL staining at the basal membrane in IRSp53-KD and RAB35-KD cysts. Scale bar, 10 μm. Bottom left: Quantification of multi-foci cysts (left) and inverted polarity cysts (right). Data are means ± SD. Four-cell stage cysts were analysed, as at least 20 cysts/experiment in $n = 7$ (Ctr; IRSp53-KD) and $n = 3$ (RAB35-KD) (multi-foci) or $n = 7$ (Ctr), $n = 4$ (IRSp53-KD) and $n = 2$ (RAB35-KD) (inverted polarity) independent experiments. P value, student's t-test two-tailed (multi-foci) or one-tailed (inverted polarity). *$p < 0.05$; ***$p < 0.001$. Source data are provided as a Source Data file. Bottom right: immunoblotting of IRSp53, RAB35 and vinculin levels.

---

alterations in the structure of the pronephric ducts. These ducts were less compact and dense (Supplementary Fig. 9A), with irregular or multiple lumens (Supplementary Fig. 9B) and frequently (in ~67% of embryos) showed the formation of ectopic ClaudinB-positive structures emerging from their basal surfaces (Fig. 8a, Supplementary Fig. 9B, Supplementary Fig. 10A, and Supplementary Movie 5). The multiple lumens and ectopic pseudo-lumenized structures characterized by irregular and diffused acetylated-tubulin and F-actin distributions (Supplementary Fig. 9B and Supplementary Fig. 10A) were reminiscent of the defects observed in mammalian organoids devoid of IRSp53 caused by perturbation of the correct polarity program.

Despite the absence of an overt phenotypic alteration in the pronephric ducts of singly mutated b2a embryos, we reasoned that in adult life, aging-dependence or environmental stresses might combine with the genetic alterations to cause phenotypic perturbations that are not visible during development. Consistently, histological analysis of adult zebrafish kidneys at various ages (e.g., 4, 8 months) showed several morphological defects in the kidneys of the adult b2a mutant line (Fig. 8b). The general architecture of the organs of the b2a mutants was altered, with a decrease in the density of the tubules and a relative increase in the epithelial-to-hematopoietic cell ratio, compared to those of age-matched WT animals (Fig. 8b). This was accompanied by vascular lacunae and defects in the architectural organization of the renal tubules, which were frequently coarsed or lacked an overt lumen (Fig. 8b).

Prompted by these observations, we examined the renal architecture in the IRSp53-KO mice. IRSp53-KO mice have a partially penetrant mid-gestation lethality that is due to defects in placental morphogenesis, as previously reported[41]. Some 20% to 25% of these mice, however, are born fertile and devoid of gross tissue dysmorphology[8,42,43]. Histopathological and immunohistochemical analyses of renal tissues of the IRSP53-KO mice, however, showed some relatively mild, but evident and recurrent, alterations that were reminiscent of those in the kidneys and tubules of the adult B2a mutant zebrafish (Fig. 9, Supplementary Fig. 10B). The overall morphology of the IRSp53-KO mouse kidneys was defective, with thinning of the cortex despite preserved glomerular density. The renal proximal tubules showed irregular contours that resulted in ill-defined lumina lined by irregularly spaced epithelial cells with prominent nuclear dysmorphism (Fig. 9a). Similar architectural alterations were seen in the distal tubules. Focal periodic acid Schiff–positive urinary casts were detected in the IRSp53-KO mouse kidney parenchyma (Supplementary Fig. 11A). In addition, the glomeruli of these IRSp53-KO mice showed alterations in the capillary tufts,

because of multifocal pseudo-aneurismatic vascular dilations and/or mesangial cell proliferation (Fig. 9a). The severity of these phenotypes was scored using a histological alteration score (Fig. 9b, Supplementary Fig. 10B). Immunohistochemistry analysis of both apical and basal epithelial renal markers further highlighted defects in the epithelial tubular cell apical–basal polarity, as shown by the following findings (Supplementary Fig. 11B–D): (i) ZO-1, which labeled tight apical junctions in most tubules of the control kidney, was diffuse or reduced in IRSp53-KO kidney tubules (Supplementary Fig. 11B); (ii) similarly, the apical label aPKCζ was diminished, and was sometimes absent from the lumen of mutant animals (Supplementary Fig. 11C); (iii) endomucin, which localizes to the basal side of tubules in the controls, remained largely basal when expressed, but was irregularly distributed and sometimes absent in the IRSp53-KO tubules (Supplementary Fig. 11D).

Altogether these data indicate that IRSp53 loss results in a set of subclinical morphological and histological aberrancies in the developing kidney and in adult renal tubules in zebrafish and mice, by impinging on, at least in part, the correct establishment or maintenance of epithelial polarity.

## Discussion

During the development of tubular or glandular structures of epithelial tissues, and specifically of kidneys, de-novo formation of a polarized central lumen is a key step that requires spatio-temporally coordinated interplay between actin cytoskeleton dynamics and targeted exocytosis of apical cargos, such as PODXL[31]. Although the molecular players that coordinate these events have been intensely studied[44], we still do not have a complete mechanistic understanding of this process. In addition, whether the knowledge gleaned from the analysis of in vitro morphogenesis of multicellular systems, such as cysts derived from MDCK or Caco-2 cells[1,26,27,45–47], is physiologically relevant in vivo for the development of complex organs has not been systematically tested.

Here, we show that IRSp53, a CDC42 effector that can sense Ptdins(4,5)P$_2$-rich negative membrane curvature[8,10,23–25], acts as a platform for the assembly, integration, and spatial restriction of multiple complexes that are needed for correct formation of the polarized single lumen in various epithelial systems in vitro. We further show that IRSp53 is critical in the shaping of the plasma membrane, and to ensure its integrity and continuity in the nascent lumen (Fig. 10). These functional roles in in vitro systems are mirrored by the requirement for IRSp53 for correct morphogenesis and structural organization of kidney tubules in mice

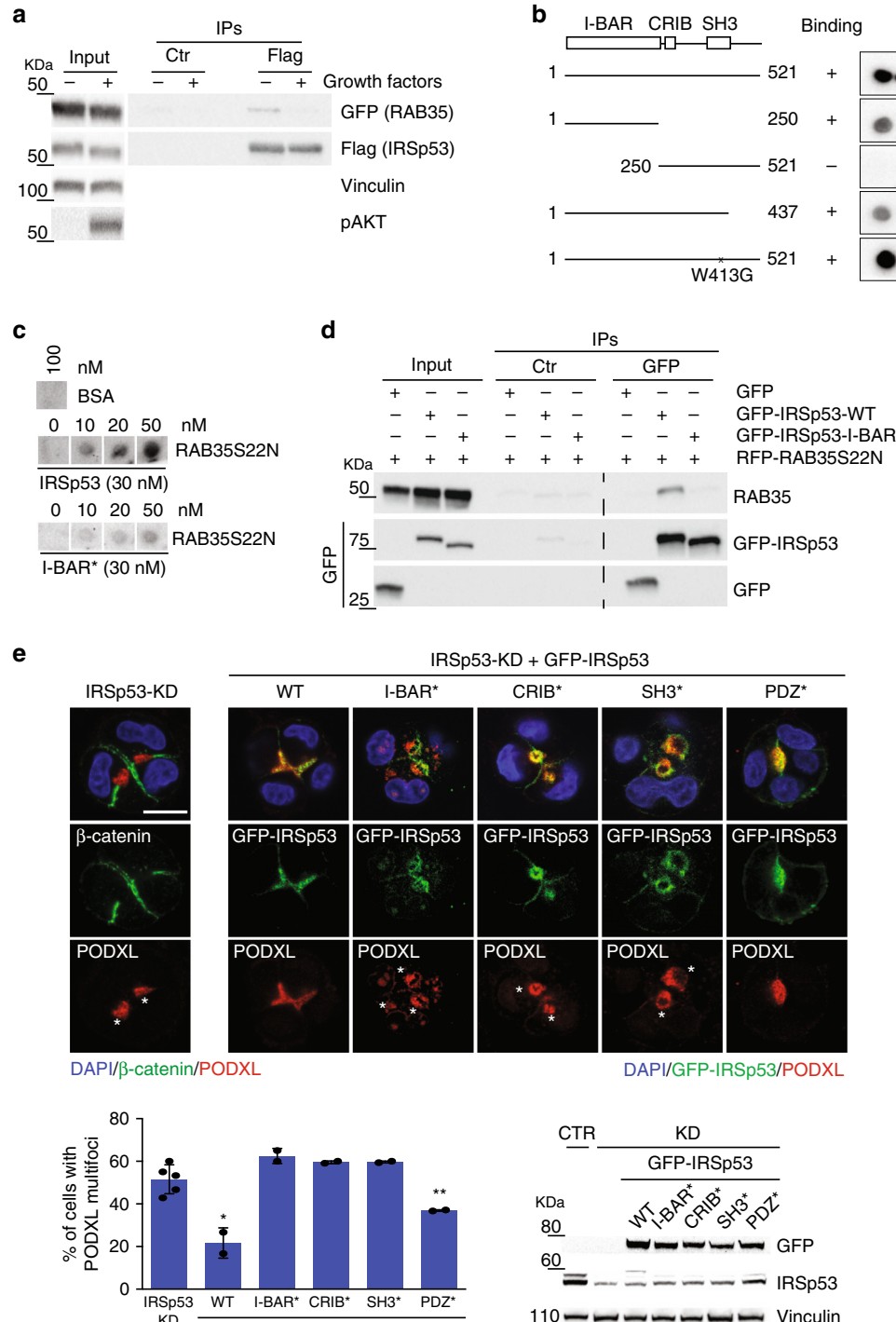

and zebrafish, where it controls the appropriate distribution of different polarity determinants.

Topologically, IRSp53 is apically restricted at the luminal membrane side of tubular and glandular epithelial tissues in human, murine, and zebrafish, and in 3D epithelial acini in vitro. This apically restricted localization is particularly prominent in adult murine and human kidney tubules, and in the pronephric ducts of zebrafish during development. Depletion of IRSp53 disrupts cystogenesis in vitro, which leads to multi-lumen formation and inversion of polarity. A remarkably similar set of alterations, which includes formation of multiple lumens and of ectopic structures that have lost their stereotypical apical–basal morphology, are also observed in the pronephric duct after

genetic removal of the two zebrafish paralogs of mammalian IRSp53. Renal tubules in IRSp53-KO mice are convoluted, not patent, and frequently display an aberrant distribution of the epithelial cells that normally surround the lumen in an orderly and stereotypical pattern. The latter phenotypes correlate with incorrect distribution of various apical determinants, including ZO-1 and aPKC. Collectively, these findings support the crucial involvement of IRSp53 in renal lumenogenesis and tubule morphogenesis across different species.

The relatively mild tubular defects in adult organisms appear to reflect the plasticity of the epithelial morphogenesis processes, which can use multiple, and often redundant, molecular pathways to ensure maintenance of the correctly organized structure and

**Fig. 5 IRSp53 directly binds RAB35 in its inactive GDP-bound state. a** Lysates (1 mg) of HeLa cells transfected with GFP-RAB35 and Flag-IRSp53, serum starved overnight and treated (+) or not (−) with growth factors, were subjected to immunoprecipitation with anti-Flag (Flag) or control (Ctr) antibodies. Inputs (20 μg) and IPs were analyzed by immunoblotting with the indicated antibodies. pAKT was used for the positive control for growth factor stimulation. **b** Structure function analysis. Equal amounts (50 nM) of recombinant purified IRSp53 full-length WT or the indicated fragments and mutant were spotted onto nitrocellulose and incubated with recombinant purified GST-RAB35S22N (50 nM). After washing, nitrocellulose membranes were immunoblotted with an anti-GST antibody. **c** Indicated amounts of recombinant GST-RAB35S22N were spotted onto nitrocellulose and incubated with equal amounts (30 nM) of either recombinant purified IRSp53 WT or IRSp53 I-BAR* (K142E, K143E, K146E, K147E). After washing, nitrocellulose membranes were immunoblotted with an anti-IRSp53 antibody. BSA (100 nM) was used as the negative control. **d** Cell lysates (2 mg) from MDCK cells transfected with GFP empty vector, GFP-IRSp53 wild-type (WT), GFP-IRSp53 I-BAR*, and RFP-RAB35S22N were subjected to immunoprecipitation with anti-GFP (GFP) or control (Ctr) antibodies. Inputs and IPs were immunoblotted with the indicated antibodies. **e** Top: MDCK IRSp53-KD or IRSp53-KD cells were infected to express murine GFP-IRSp53 wild-type (WT), I-BAR*(K142E, K143E, K146E, K147E), CRIB* (I268N), SH3* (I403P) and PDZ* (V522G), and were seeded as single cells on a Matrigel layer and left to grow for 24/36 h. The cysts were fixed and stained with anti-β-catenin (green) or anti-PODXL (red) antibodies, and DAPI (blue), or processed for epifluorescence to visualize GFP-IRSp53 (green), and stained with an anti-PODXL antibody (red) and DAPI (blue). Asterisks, PODXL multi-foci. Scale bar, 10 μm. Bottom left: Quantification of multi-foci cysts. Data are means ± SD. Three/ four-cell stage cysts were analyzed, as at least 20 cysts/experiment in n = 5 (IRSp53-KD) and n = 2 (WT, I-BAR*, CRIB*, SH3*, PDZ*) independent experiments. P value, student's t-test two-tailed. *p < 0.05; **p < 0.01. Source data are provided as a Source Data file. Bottom right: Immunoblotting to detect IRSp53, GFP-IRSp53 wild-type and mutants, and vinculin.

the functional activity of the tissues. In keeping with this concept, the genetic loss of another member of the I-BAR family proteins, known as "missing-in metastasis" (MIM, also known as MTSS1), was shown to impact on adult kidney architecture through regulation of intercellular junctions. MIM$^{-/-}$ mice show a progressive kidney pathological diseased state with dilated tubules, glomerular degeneration, and fibrosis. This set of alterations is accompanied by polyuria at the systemic level, whereas increased intercellular junctional spaces are detected by ultrastructural electron microscopy analysis[48]. Thus, different members of the I-BAR family proteins contribute to the proper structural organization of adult murine kidney, with these partly acting in a redundant fashion[48]. It is also of note that some of the alterations due to loss of IRSp53 are observed in mice devoid of key components of the molecular network of the IRSp53 interactors. A case in point is CDC42, which has a pivotal role in virtually all polarized cell processes in diverse epithelial and non-epithelial tissues by acting as a central signaling hub that connects actin dynamics, vesicular trafficking, and membrane remodeling[1,49]. Indeed, genetic loss of CDC42 in murine kidney tubules during development leads to abnormal tubulogenesis with profound defects in polarity, lumen formation, and the actin cytoskeleton, which eventually result in premature death of these animals[50–52]. These alterations are significantly more severe than those detected in *IRSp53*-KO mice, which is in keeping with the concept that CDC42 is a central hub that coordinates and uses multiple effectors and pathways to control polarity. Among these, IRSp53 appears to contribute to interconnecting of the CDC42 axis with RAB35/PODXL trafficking and EPS8-based actin dynamics. However, other CDC42-dependent pathways, such as Ptdins(4,5) P2-Annexin-CDC42[53], CDC42-PAR3-aPKC[54], and CDC42/ YAP1[52] are expected to remain functional following disruption of IRSp53, and might, therefore, compensate for the lack of IRSp53.

Redundancy and compensation mechanisms of IRSp53 functions might also be at play in processes and pathways that control endocytic internalization of PODXL, inversion of polarity and Robo2-p53 axis, which, as described in the Supplementary discussion, have been implicated in the correct lumenogenesis and renal tubule formation.

At the cellular level, during 3D cystogenesis, the loss of IRSp53 leads to formation of multiple lumens and reversal of polarity, albeit less frequently. The formation of multiple lumens occurs frequently following disruption of a variety of trafficking, polarity, and cytokinetic regulators. This indicates that none of these processes is essential for establishment of apical–basal epithelial polarity; rather, they all contribute to ensure the formation of the

single AMIS that will eventually generate a single lumen. In the case of IRSp53, its loss is likely to impact on the optimal functionality of many of these critical cell-biology processes. Indeed, IRSp53 is recruited very early on, at the onset of AMIS formation, which appears to be as a consequence of its I-BAR domain sensing of Ptdins(4,5)P$_2$-rich, negative membrane curvature that is a distinguishing early feature during de-novo formation of the AMIS around the midbody immediately after completion of the first cytokinesis[5,53]. At this site, and similar to what is seen in extending protrusions of migratory cells[12,55–59], IRSp53 accumulates together with its interactor, the actin capping protein EPS8. This protein is expected to regulate not only local cortical actin dynamics, but also to maintain IRSp53 in an "open conformation", opposing the inhibitory actions of 14-3-3[23,60]. This might facilitate the association of IRSp53 with Ptdins(4,5)P$_2$-rich membranes, and also with activated CDC42 and inactive RAB35. In keeping with this concept, removal of EPS8 not only phenocopies the loss of IRSp53 in cystogenesis assays, but also results in redistribution of IRSp53, which loses its continuous apical pattern, and becomes concentrated at tight junctions in association with ZO-1. The importance of the interaction between IRSp53 and activated CDC42 is underscored by the observation that a single-point mutation in the CDC42 binding site of IRSp53 fails to rescue the phenotype in *IRSp53*-KO MDCK and Caco-2 cells. In addition, in vitro, IRSp53 loss mimics the removal of CDC42[26,61,62], which leads to the formation of multiple lumens. Of note, the loss of CDC42 also disrupts the orientation of the spindle in dividing cells during cystogenesis[26,61,62] and controls the trafficking of polarized determinants. Both of these altered processes are thought to be responsible for the aberrant formation of multiple lumens. Within this context, IRSp53, which directly binds to CDC42, might mediate some of these specific functions. Consistent with this, trafficking of PODXL was perturbed and IRSp53 localized to PODXL-positive recycling endosomes. Further, the loss of IRSp53 significantly altered the spindle orientation during Caco-2 cystogenesis, to an extent and in a fashion not dissimilar from those seen after the loss of CDC42 (data not shown). While the precise mechanisms through which IRSp53 applies this multiple-level regulation remains to be defined, the most parsimonious interpretation of our findings is that IRSp53 acts as an important CDC42 effector, during both spindle orientation and polarized membrane trafficking.

In addition to this role, the localization of IRSp53 at the AMIS and its interaction with RAB35 suggests that it is also part of a set of molecular cues that define the identity of, and ensure the structural integrity of, the future apical membrane. Here, IRSp53

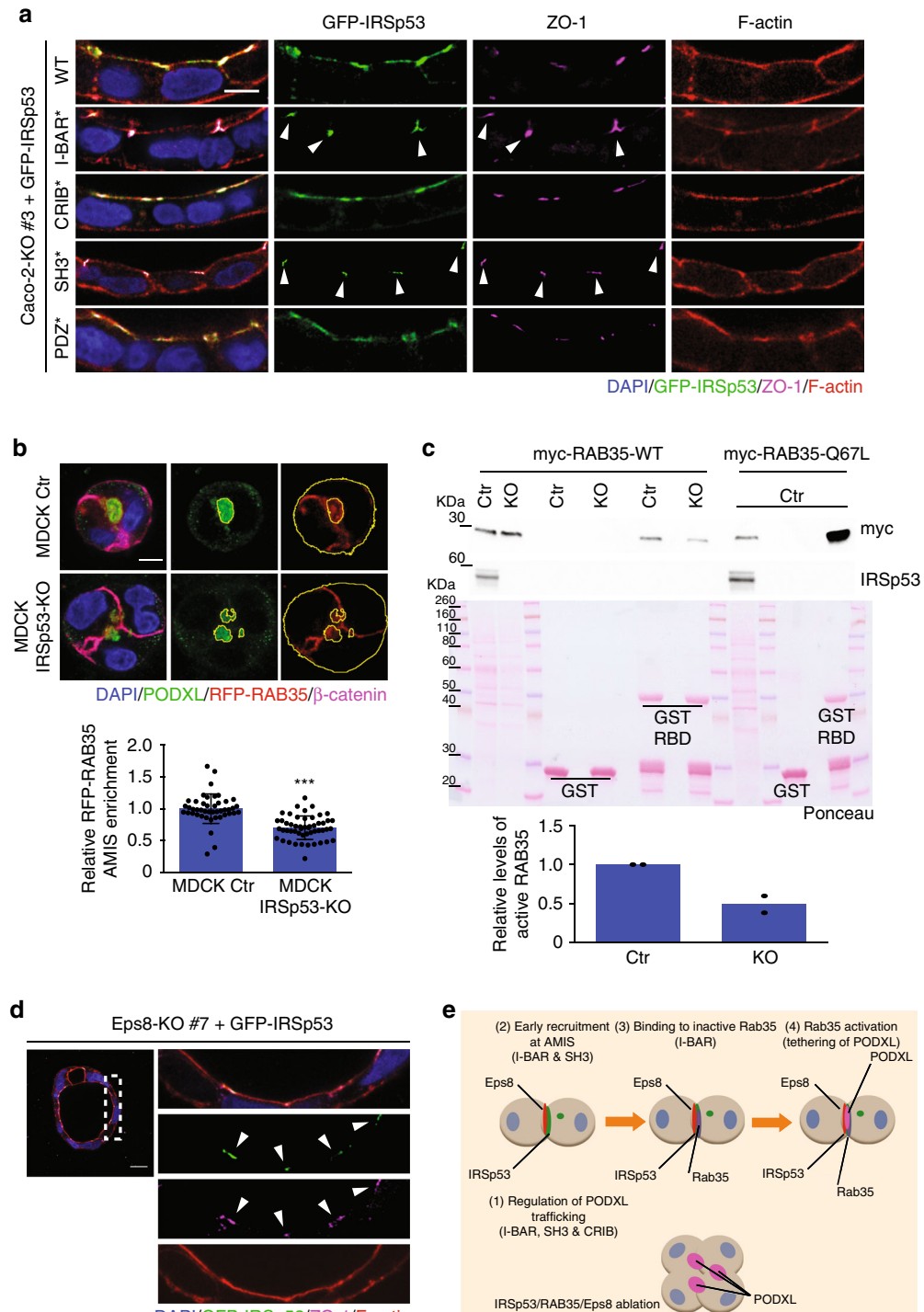

might act as a stabilizing factor for RAB35, either directly through protein:protein interactions, or indirectly through sensing and enhancing the concentration of PtdIns(4,5)P2[21], to which RAB35 is known to associate[63]. In this latter respect, it is worth noting that: (i) as for the loss of RAB35 that was recently demonstrated to function as a PODXL tethering factor at the AMIS[7], also the loss of IRSp53 disrupts the correct apical distribution of PODXL; (ii) the binding to RAB35 and Ptdins(4,5)P2-rich membrane is mediated by the I-BAR domain. In addition, a mutant in the key positively charged patches of this domain that disrupts the interaction of IRSp53 with Ptdins(4,5)P$_2$-rich lipid bilayers[37] as well as the binding to RAB35 failed to correctly localize apically

and to rescue the multi-lumen phenotype in the *IRSp53*-KO cysts; (iii) *IRSp53* ablation impairs RAB35 localization at the nascent AMIS, which demonstrates that IRSp53 drives the correct localization of RAB35. An equally plausible, although not mutually exclusive, possibility is that in addition to binding to inactive RAB35 along the AMIS, IRSp53 might associate with regulators of the RAB35 GTPase, including GEFs or GAPs, to thus contribute to the regulation of its activity. Our proteomic studies using IRSp53–BirA approaches that can detect transient interactions[64], however, failed to identify any of these proteins {deposited at PASS01464, for details see Methods}, which would argue against this mode of action. Finally, it is of note that also

**Fig. 6 Membrane- and SH3-mediated interactions guide IRSp53 localization. a** Seven days-old 3D cysts from Caco-2 *IRSp53*-KO #3 cells stably infected with murine GFP-IRSp53 WT, I-BAR*, CRIB*, SH3*, PDZ* were processed to visualize GFP-IRSp53 (green) and stained with an anti-ZO-1 antibody (magenta), rhodamine-phalloidin to detect F-actin (red), and DAPI (blue). Arrowheads indicate IRSp53 I-BAR* and SH3* at ZO-1-junctions. Scale bar, 10 μm. **b** Top: 1 day-old 3D cysts from MDCK wild-type control (Ctr) or IRSp53-KO cells stably expressing RFP-RAB35 were processed to visualize RFP-RAB35 (red), stained with anti-PODXL (green) and anti-β-catenin (magenta) antibodies, and DAPI (blue). PODXL and RFP-RAB35 signals were analysed using masks (in yellow) to quantify RFP-RAB35 localization at the AMIS. Bottom: Quantification of RFP-RAB35 at the AMIS expressed as the ratio of RFP-RAB35 at the AMIS/ RFP-RAB35 out of the AMIS. Data are means ± SD. At least 15 cysts/ experiment were analyzed in $n = 2$ independent experiments. *P* value, student's *t*-test two-tailed. ***$p < 0.001$. Source data are provided as a Source Data file. Scale bar, 10 μm. **c** Top: Lysates of MDCK control (Ctr) or IRSp53-KO (KO) transfected with myc-RAB35-WT or MDCK wild type transfected with myc-RAB35-67L were subjected to in vitro pull down with 0.2 μM of recombinant purified GST-RBD or GST as control. Inputs and IVBs were analysed by Ponceau to detect GST or GST-RBD purified proteins, and subjected to Western blotting with the indicated antibodies. Bottom: RAB35 levels quantified using ImageJ software were expressed relative to the levels of active-GTP bound RAB35 in MDCK wild type samples. Data are means from $n = 2$ independent experiments. **d** Seven day-old 3D cysts from Caco-2 *Eps8*-KO #7 cells infected with murine GFP-IRSp53 wild-type were processed to visualize GFP-IRSp53 (green), stained with an anti-ZO-1 antibody (magenta), rhodamine-phalloidin to detect F-actin (red), and DAPI (blue). Right panels represent 4× magnification of the area depicted by the dashed square (Left). Arrowheads, re-localization and enrichment of IRSp53 at the ZO-1–labeled cell–cell junctions. Scale bar, 20 μm. **e** Schematic of the functional interactions of IRSp53 with RAB35 and EPS8 at the AMIS. The IRSp53 domains required for its localization and activity on PODXL trafficking.

other I-BAR domain family members, IRTKS and MIM, have been shown to interact with RAB GTPases and to be involved in endocytosis[65]. Thus, the involvement in membrane trafficking appears to be a more general emerging function of the I-BAR family proteins.

Ultimately, the multiple functional roles of IRSp53 are likely to be integrated with its sensing and shaping of the flat plasma membrane at the nascent lumen via its I-BAR domain. Here, IRSp53 appears to be crucial in the control of the integrity, continuity, and shape of the opposing plasma membrane. Indeed, its loss leads to the formation (or preservation) of bridges between the apical domains of the plasma membrane, which interrupt the continuity of the nascent lumen (Fig. 10). It appears reasonable to propose that it is exactly this structural, membrane role in combination with its control of the trafficking of key polarity determinants that accounts for its involvement in lumen formation and tubule morphology.

Whatever the case, all in all, our findings are consistent with a multifaceted role for IRSp53 that is crucial to ensure the structural integrity and shape of the plasma membrane, and to coordinate the trafficking of apical proteins to the AMIS. They further support the concept that IRSp53 acts as a platform where trafficking determinants (e.g., RAB35), actin regulatory proteins (e.g., EPS8), and the small GTPase that is critical for polarity (i.e., CDC42) are assembled for the correct coordination of diverse processes during tubule morphogenesis and the establishment of renal tubular architecture in different species.

## Methods

All mice have been maintained in a controlled environment, at 18–23 °C, 40–60% humidity and with 12-h dark/12-h light cycles, in a certified animal facility under the control of the institutional organism for animal welfare and ethical approach to animals in experimental procedures (Cogentech OPBA). All animal studies were conducted with the approval of Italian Minister of Health (598/2015-PR and 219/2016-PR) and were performed in accordance with the Italian law (D.lgs. 26/2014), which enforces Dir. 2010/63/EU (Directive 2010/63/EU of the European Parliament and of the Council of 22 September 2010 on the protection of animals used for scientific purposes).

**Antibodies, plasmid and reagents**. A list of antibodies, details on their application and working dilutions are provided in Supplementary Tables 1 and 2.

pFUW-GFP, pFUW-GFP-IRSp53 WT and I-BAR*(K142E, K143E, K146E, K147E), CRIB* (I268N), SH3* (I403P) and PDZ* (V522G) mutants were a gift from Hans-Jürgen Kreienkamp, University of Hamburg; pRK5 Flag-IRSp53 was a gift from Sonia Krugmann; pRRL-GFP-Eps8 was generated by sub-cloning GFP-Eps8 in pRRL lentiviral vector[66]; pApple-Rab11a was a gift from Keith Mostov; GFP- and RFP-RAB35 WT, -RAB35 S22N and -RAB35 Q67L were kind gifts from Cécile Gauthier-Rouviere, Centre de Recherche en Biologie cellulaire de Montpellier; pGEX-RAB35-S22N, RAB35-Q67L were generated by sub-cloning GFP-RAB35-Q67L, -RAB35-S22N in pGEX6P1 vector; RFP-PODXL was a gift

from Fernando Martín-Belmonte, Centro de Biología Molecular Severo Ochoa, Madrid.

MDCK IRSp53 shRNAmir inducible cell line were a gift from Ann Musch and were described elsewhere[24].

**Cell culture**. MDCK cells were grown in Dulbecco's Modified Eagle Medium (DMEM, Lonza) supplemented with 5% South American serum (EuroClone) and 2 mM L-Glutamine (EuroClone). MDCK TetOFF cells were grown in Dulbecco's Modified Eagle Medium (DMEM, Lonza) supplemented with 5% South American serum (EuroClone), 2 mM L-Glutamine (EuroClone); controls were maintained in the presence of 1 μg/ml tetracycline (Sigma) to repress IRSp53 RNAi. Caco-2 cells were grown in Modified Eagle Medium (MEM, Biowest) supplemented with 20% South American serum, 2 mM L-Glutamine and 0.1 mM NEAA. HEK 293-T cells were grown in Dulbecco's modified Eagle's medium (DMEM, Lonza) supplemented with 10% South American serum (EuroClone) and 2 mM L-Glutamine (EuroClone). Cells were grown at 37 °C in 5% $CO_2$. HeLa cells were grown in Minimum Essential Medium (MEM Invitrogen) supplemented with 10% South American serum (EuroClone), 1% non-essential amino acids and 1% Sodium Pyruvate.

**Transfection**. Transfections were performed using the calcium phosphate method or liposoluble agents JETPrime (POLYPlus), INTERFERin (POLYPlus) and FugeneHD (Promega).

293 T cells were transfected using the calcium phosphate procedure. In this case DNA (10 μg for a 10 cm plate) was diluted in 439 μl of $ddH_2O$ and 61 μl of 2 M $CaCl_2$ were added. This solution was added, drop-wise, to 500 μl of HBS 2×(50 mM Hepes pH 7.5, 10 mM KCl, 12 mM dextrose, 280 mM NaCl, 1.5 mM $Na_2HPO_4$). Then, the precipitate was added to the cells and the medium replaced after 12–16 h.

JETPrime (POLYPlus) was used, according to manufacturer's instruction, to transfect MDCK cells with fluorescent-tagged CRISPR/CAS9 plasmid carrying sgRNA sequences, and fluorescent or tagged proteins for IF or IP experiments.

INTERFERin (POLYPlus) was used, according to manufacturer's instruction, to transfect MDCK cells for transient RNAi interference experiments.

FugeneHD (Promega) was used, according to manufacturer's instruction, to transfect CACO2 with fluorescence-tagged CRISPR/CAS9 plasmid carrying sgRNA sequences.

**CRISPR/CAS9 genome editing**. MDCK and Caco-2 transiently transfected with fluorescent-tagged CRISPR/CAS9 plasmid carrying sgRNA sequences (see below for sgRNA sequences and analysis of the lesions introduced); 48 h after transfection cells were sorted by FACS and re-cultured. Mass population were subjected to WB and IF analysis to verify the extent of KO. Clones were then derived by serial dilution. Analysis of the lesions introduced was performed by PCR.

MDCK
sgRNA:
WT 5′-GGCAGCAGG<u>CCTGCGCCGACCCCAACAAGATC</u>CCAGACCGC GCGGTGCAG-3′.

Sequence analysis:
Allele1
5′-GGCAGCAGGCCTGCG**T**CCGACCCCAACAAGATCCCAGACCGC GCGGTGCAG-3′.

Allele2
5′-GGCAGCAGGCCTGC <u>- -</u> CGACCCCAACAAGATCCCAGACCGCGC GGTGCAG-3′.

Both mutations cause the premature termination of the protein at the end of the I-BAR domain.

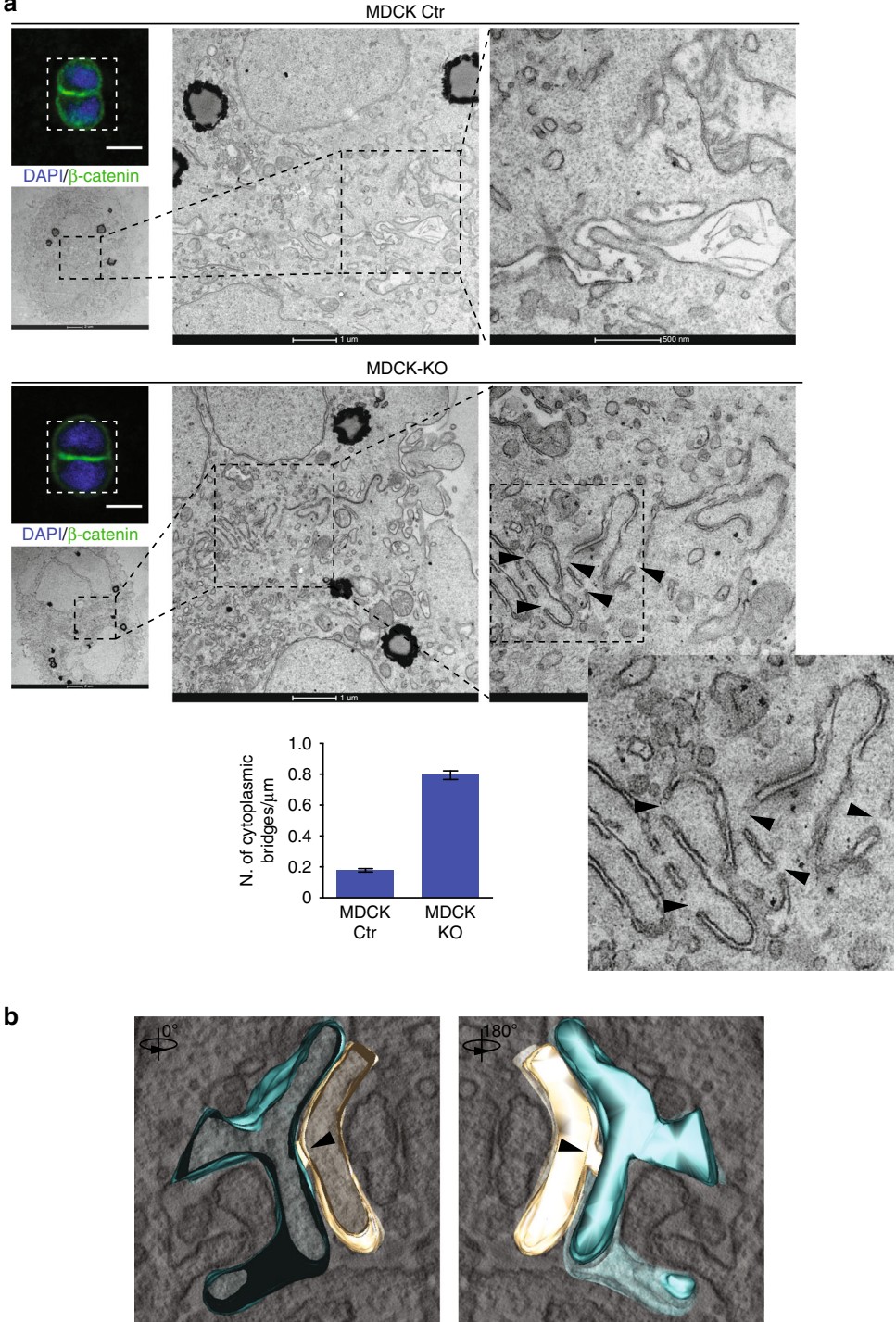

**Fig. 7 Loss of IRSp53 alters the opposing plasma membranes at the nascent AMIS. a** MDCK Ctr (top) and IRSp53-KO cells (middle) were seeded as single cells on Matrigel-coated gridded coverslips. The cysts were fixed 16 h after seeding, stained with anti-ß-catenin (green) and DAPI (blue). Two-cell stage cysts were initially identified on grids by confocal microscopy. Scale bar, 10 μm. Samples were then processed for electron microscopy, with images of the corresponding cells shown at the indicated magnifications. Arrowheads, inter-cytoplasmic bridges of the plasma membranes along the AMIS. Bottom: Quantification of the inter-cytoplasmic bridges/μm. Two-cell stage cysts were analysed from $n = 2$ independent experiments. In all, $n = 484$ (Ctr) and n = 332 (KO) different fields from $n = 6$ (Ctr; KO) samples were counted from serial sections along the Z-axis. Data are means ± s.e.m. *P* value, student's t-test two-tailed. ***$p < 0.001$. Source data are provided as a Source Data file. **b** Images of a three-dimensional tomographic reconstruction of an inter-cytoplasmic bridge at the opposing membranes of MDCK *IRSp53*-KO cells during early cystogenesis (see also Supplementary Movie 4). Arrowheads, inter-cytoplasmic bridge.

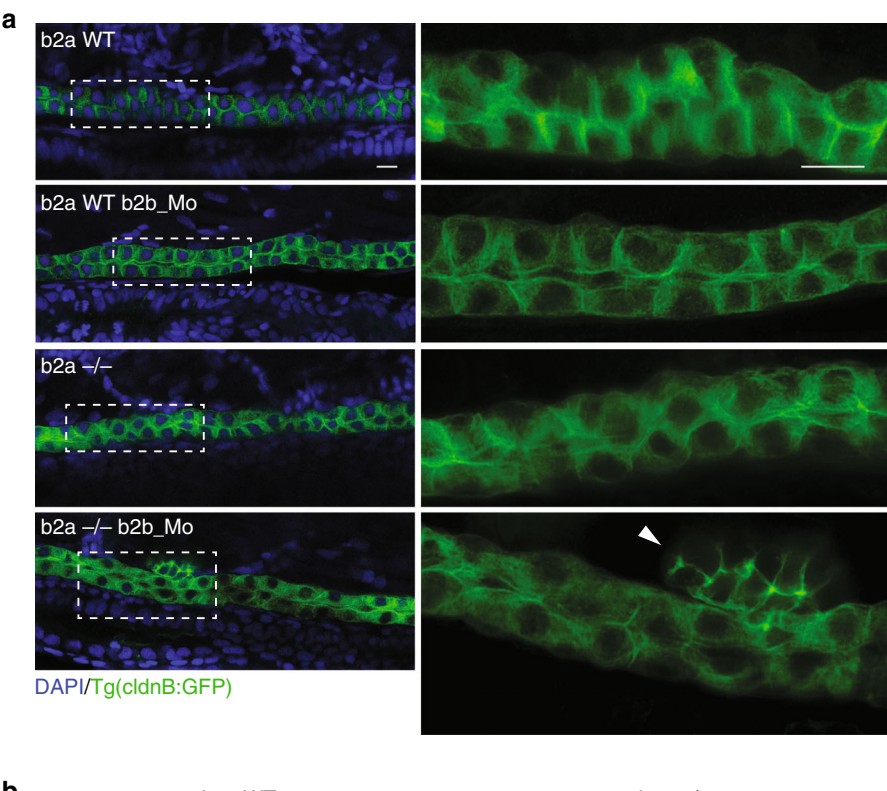

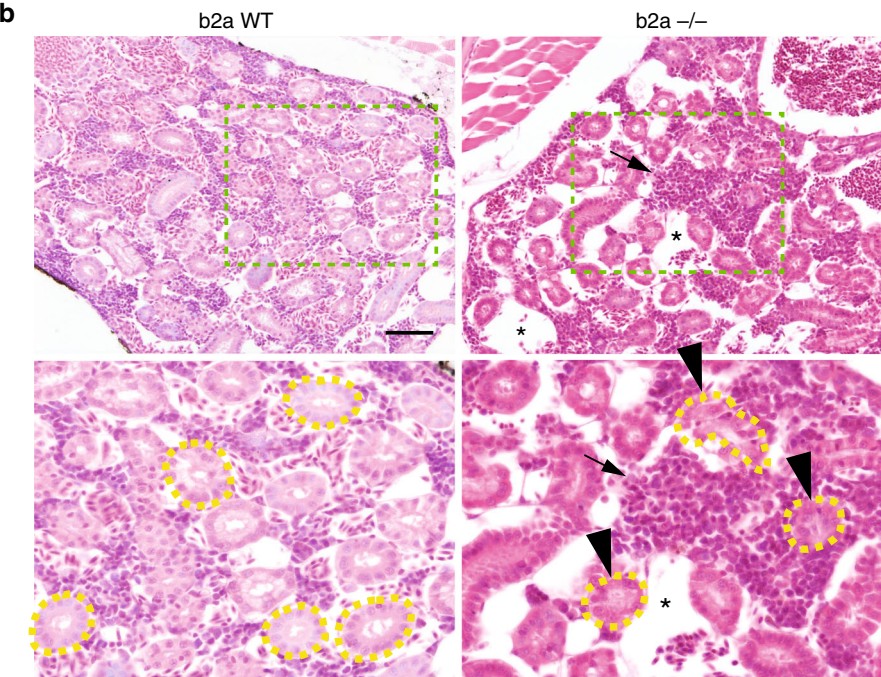

**Fig. 8 Loss of IRSp53 causes defects in pro-nephric ducts and in adult kidneys in zebrafish. a** Digital light sheet confocal images of 72 hpf pronephric ducts. Embryos obtained from either wild-type (b2a WT) or *baiap2a* (b2a −/−) mutant females in a Tg(CldnB:GFP) genetic background were treated with scrambled morpholino (b2a WT; b2a −/−) or with spliced and translation blocking morpholinos (b2a WT b2b_Mo; b2a −/− b2b_Mo), and fixed and mounted in agarose. Samples were stained with an anti-GFP antibody (green) and DAPI (blue). Z-stack projections (magnification, 3.4×) are shown on the right for the dashed, rectangular boxes on the left. Arrowhead, extra-duct structure in the b2a −/− b2b_Mo embryo. Scale bars, 10 μm. Two-thirds of the b2a −/− b2b_Mo embryos showed such defects (n = 8/12). **b** Hematoxylin and eosin staining of adult zebrafish kidneys. Adult wild-type (b2a WT) and b2a −/− strains were fixed, decalcified, and paraffin embedded. Bottom: Magnification (2×) of the light green dashed boxes from the upper images. Renal tubules are surrounded by yellow dashed lines. Arrowheads, abnormal disarrayed and not patent tubules in the b2a −/− sample, characterized by the irregular distribution and shape of the nuclei; asterisks, vascular lacunae; arrows, infiltrating cells. In all, 70% of the b2a −/− samples show such defects (n = 7/10) versus ~10% of the b2a wild-type (n = 1/9). Scale bar, 50 μm.

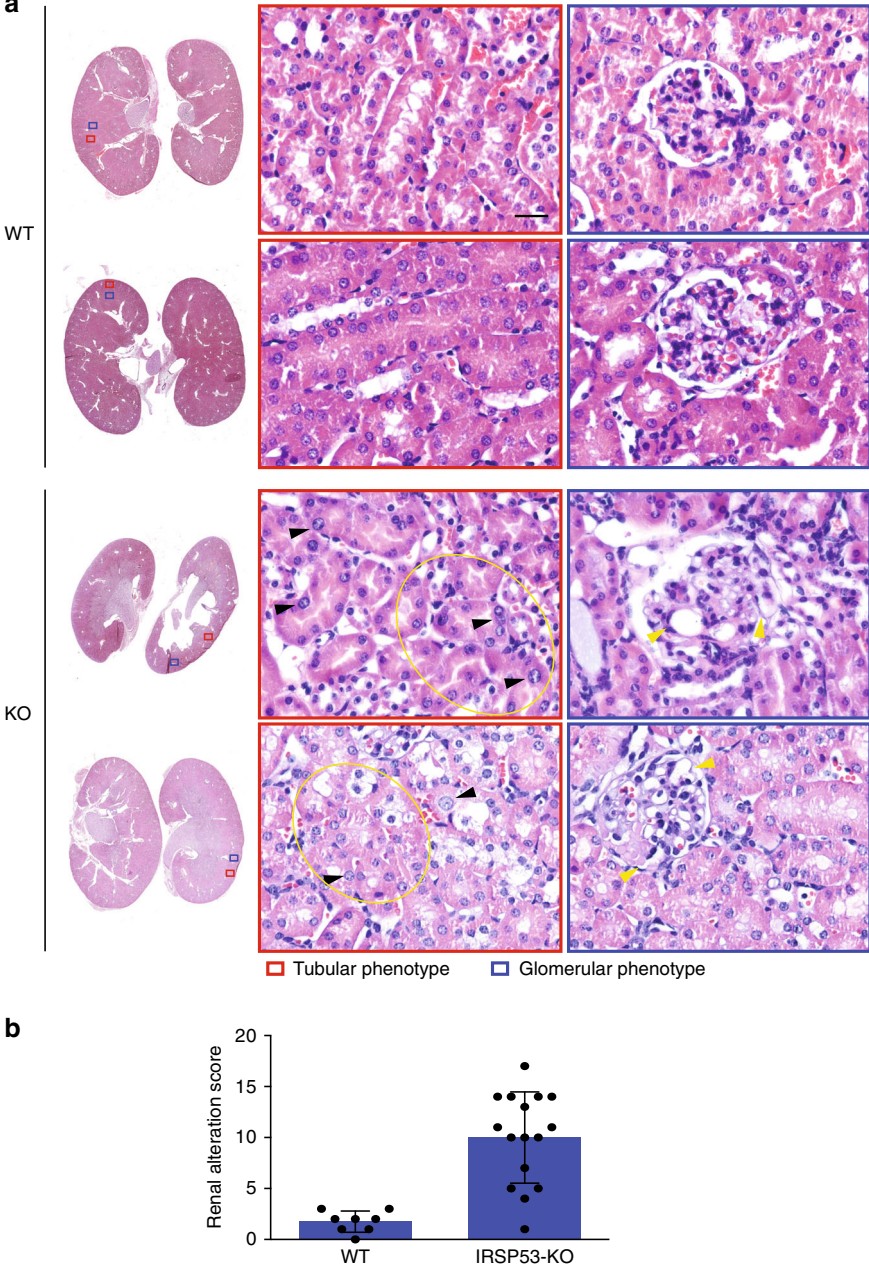

**Fig. 9 Genetic removal of IRSp53 leads to defects in murine kidney morphology. a** Hematoxylin and eosin staining of kidneys explanted from wild-type (WT) and *IRSp53*-KO adult mice were fixed, processed and paraffin embedded. Left: Low magnification images. Right: magnified images of the red (tubular phenotype) and blue (glomerular phenotype) boxes on the left. Black arrowheads, abnormal irregular distributions and shapes of the nuclei in the *IRSp53*-KO kidneys; yellow ellipses, areas with disarrayed and convoluted tubules; yellow arrowheads, glomerular tuft microcystic alterations. Scale bar, 50 μm. **b** Quantification of the renal alteration score. Kidneys from wild-type (WT) and *IRSp53*-KO mice were analyzed and scored for defects in renal morphology (see also Fig. S9B and Methods). Data are means ± SD. WT, n = 8 (4 M, 4 F); KO, n = 16 (6 M, 10 F). P value, student's t-test two-tailed. ***p < 0.001. Source data are provided in Supplementary Fig. 10 B and as a Source Data file.

Caco-2

sgRNA:

5′-AAGCAGGGCGAGCTGGAGAATTACGTGTCCGACGGCTACAAG
ACCGCACT-3′.

Clone #3:

Allele1

5′-AAGCAGGGCGAGCTGGAGAATTACG - - - - - - ACGGCTACAAGAC
CGCACT-3′.

Allele2

5′-AAGCAGGGCGAGCTGGAGAATTACG - - - - - - A - - GCTACAAGACC
GCACT-3′.

Allele3

5′-AAGCAGGGCGAGCTGGAGAATTACGTGT - - - - - - - CTACAAGACC
GCACT-3′.

Deletion in Allele 1 causes the translation of a protein lacking 2aa within the I-BAR (179-180). Deletion in Allele 2 causes premature termination of the protein at the end of the I-BAR domain (aa 311). Deletion in Allele 3 causes premature termination of the protein within the I-BAR domain (aa 184).

Clone #12

Allele1

5′-AAGCAGGGCGAGCTGGAGAATTACG - - - - - - - - - GCTACAAGACC
GCACT-3′.

Allele2

5′-AAGCAGGGCGAGCTGGAGAATTACGTGTC - GACGGCTACAAGACC
GCACT-3′.

Allele3

5′-AAGCAGGGCGAGCTGGAGAATTACG - - - - - - ACGGCTACAAGACC
GCACT-3′.

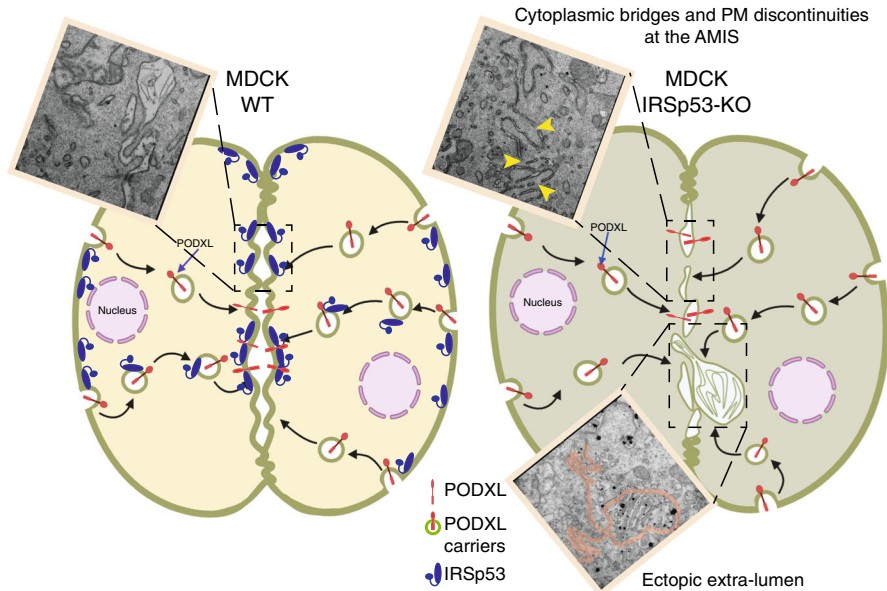

**Fig. 10 Model of how IRSp53 controls nascent lumen formation in epithelial tubular tissues.** Control MDCK cells plated on Matrigel in Matrigel-containing media undergo symmetry-breaking events immediately after the first cell division, when after cytokinetic furrow ingression a midbody is formed[4]. Around the midbody, the apical membrane initiation site (AMIS) is assembled establishing the location of the nascent lumen. AMIS assembly requires the polarized transport of highly-charged and -glycosylated transmembrane proteins, such as Podocalyxin (PODXL), that from the outer cell face become endocytosed and traffic via vesicular recycling carriers to the newly formed opposing plasma membrane. At this site, repulsive charges contribute to generate interconnected mini-lumens (leftmost inset show a Correlative Light Electron Microscopy-CLEM-analysis of MDCK cells capture after the very first cell division) that will evolve to form a single apical lumen, typical of an epithelial cyst. IRSp53 is required both for directing the trafficking of PODXL and to ensure its targeting at the AMIS via stabilizing the GTPase RAB35 (not shown), and through its interaction with the actin capping protein Eps8 (not shown). At the AMIS, IRSp53 through its membrane deforming and curvature sensing I-BAR domain ensures the structural integrity, continuity and correct shape of the opposing plasma membrane at two-cells stage. Consistent with this, IRSp53 genetic loss leads to interruption of the continuity of the PM at the AMIS (top rightmost inset, yellow arrowhead indicate membrane discontinuities), with the formation of inter-cytoplasmic bridges, and occasionally ectopic lateral lumen (bottom inset, magenta line outlines the intervening PM and the extra lumen).

Deletion in Allele 1 causes the translation of a protein lacking 3aa (179-180-181). Deletion in Allele 2 causes premature termination of the protein within the I-BAR domain (aa 186). Deletion in Allele 3 causes the translation of a protein lacking 2aa within the I-BAR (179-180).

PAM sequence; gRNA sequence.

**Short interfering RNA (siRNA) experiments.** siRNAs (small interfering RNAs) delivery was achieved by mixing from 5 to 50 nM of specific siRNAs with INTERFERin Transfection Reagent (POLYPlus) according to manufacturer's instruction. For each RNA interference experiment, negative control was performed with the same amounts of scrambled siRNAs (si CTR). Knocking down efficiency was controlled by western blot.

Oligos details as follows:

IRS 302 CCAAGGAACUCGGAGACGUUCUCUU (Invitrogen).
IRS 749 CAUAGUGGCAGUUUCUGUGCCAGCA (Invitrogen).
RAB35-1 AGAAGAUGCCUACAAAUUCUU (Invitrogen).
RAB35-2 GCUCACGAAGAACAGUAAAUU (Invitrogen).

**Lentiviral infections.** GFP and GFP-IRSp53 WT or mutants were transduced into MDCK, Caco-2, Huvec and mEC cells by lentiviral infection using pFUW-based plasmids. Lentivirus was produced transfecting HEK 293 T cells with 25 μg of pFUW- or pLLRsin-based constructs, 9 μg ENV, 16.25 μg pMDL, 6.25 μg REV (for virus packaging). Forty-eight and 72 h after transfection, supernatants were collected and passed through a 0.45 μm filter. Supernatants collected at 48 and 72 h were added to cells and medium was changed after 6 h. GFP and GFP-IRSp53 WT and mutants and GFP Eps8 expression was controlled by WB and IF.

**Cyst Matrigel culture.** MDCK cells were trypsinized to a single cell suspension at $4 \times 10^4$ cells/ml in complete medium containing 2% Matrigel (BD Biosciences). Cell-medium-Matrigel suspensions (250 μl) were plated into μ-Slide 8 well ibiTreat chambers (Ibidi, 80826), precoated with 15 μl of Matrigel (10 mg/ml). For long cultures, medium was replaced every 2 days. Cells were grown at different time points before fixation in 4%paraformaldehyde (PFA).

**Cyst Matrigel/collagen culture.** Caco-2 cells were trypsinized to a single cell suspension at $6 \times 10^4$ cells/ml and mixed with 20 mM HEPES pH7.5, 1 mg/ml Collagen I (Corning®), 50% Matrigel (BD Biosciences). Final volume was adjusted with complete medium containing. Cell-Matrix-medium suspensions (250 μl) were plated into μ-Slide 8 well ibiTreat chambers (Ibidi, 80826), and left in the incubator at 37 °C for at least 30 min to allow the mix to solidify through polymerization of Collagen I and Matrigel. The solidified mixture was then supplemented with 250 μl of complete medium. Cells were grown at different time points before fixation in 2%paraformaldehyde (PFA). For long cultures, medium was replaced every 2 days. To speed-up lumen formation in mature cysts, Cholera Toxin (0.1 μg/ml;) was added at day 6 and cysts culture fixed at day 7.

**Immunofluorescence.** Cells were plated on glass coverslips (pre-incubated with 0.5% gelatin in PBS at 37 °C for 30 min). Cells were processed for epifluorescence or indirect immunofluorescence microscopy as follow. Cells were fixed in 4% paraformaldehyde for 10′, washed with PBS and permeabilized in PBS 0,1% Triton X-100 for 10 min at room temperature (RT). To prevent non-specific binding of the antibodies, cells were then incubated with PBS in the presence of 1 % BSA for 10 min.

The coverslips were then gently deposited, face down, on 50 μl of primary antibody diluted in PBS 1% BSA, spotted on Parafilm. After 40 min of incubation at RT, coverslips were transferred into 12 well plates and washed three times with PBS. Cells were then incubated for 40 min at RT with the appropriate secondary antibody. F-actin was detected by staining with FITC or TRITC-conjugated phalloidin (Molecular Probes) at a concentration of 6.7 U/mL. After three washes in PBS, coverslips were transferred into 12 well plates and incubated in PBS containing DAPI (1:3000) for 5 min at RT. Coverslips were washed three times in PBS coverslips and mounted in Mowiol (20% Mowiol (Sigma), 5% Glycerol, 2.5% DABCO (Molecular Probes), 0.02% NaN3 in PBS) and examined by fluorescent optical microscopy or in a 90% glycerol solution containing diazabicyclo-(2.2.2) octane antifade (Sigma) and examined under confocal microscope (see below for further information regarding confocal microscopes). Confocal image acquisition was performed in sequential mode to limit channel cross-talk and corrected for residual fluorescence bleed through. Images were further processed with the Image J software (Adobe).

**Immuno-fluorescence of MDCK cysts.** Cysts seeded on Matrigel into μ-Slide 8 well ibiTreat chambers were fixed at different time point after seeding in PBS 2%

PFA for 20 min, washed 3 times in PBS 1% Glycine (quenching) and left 2 days in PBS 1x at 4 °C. Cysts were permeabilized in PBS + 0.5% Triton X-100 and quickly washed in IF-buffer (PBS, 0.2% Triton X-100, 0.1% BSA, 0.05% Tween). Samples were then incubated in blocking solution (IF-buffer + 5% BSA) for 1 h and 30 min. After 1 wash in IF-buffer, cysts were incubated with primary antibodies in IF-buffer, overnight at 4 °C. Samples were washed 3 times in IF-buffer (5 min each) and incubated with the appropriate secondary antibody and/or phalloidin for 1 h and 30 min in the dark. After 1 wash in IF-buffer (5 min) and 2 washes in PBS samples were incubated with Dapi (5 min). Samples were washed 2 times in PBS and them left in PBS for confocal analysis. Images were acquired with confocal microscopes of the Leica SP series (see below for the detailed information) equipped with a HCX PL APO 40×/0.75–1.25 oil immersion objective.

**Immuno-fluorescence of Caco-2 cysts**. Cysts embedded in Matrigel/Collagen I into μ-Slide 8 well ibiTreat chambers were fixed at different time point after seeding in PBS 2% PFA for 20 min, washed 3 times in PBS 1% Glycine (5 min each), 1 time in PBS and left 2 days in PBS, 0.03 % NaN₃ at RT. Cysts were permeabilized in PBS + 0.5% Triton X-100 and rinsed 3 times in PBS (5 min each). Blocking was performed in IF-buffer (PBS 0.2% Triton X-100, 0.05% Tween 20, 0.1% BSA, 0.03% NaN₃) + 10% Goat serum for 1 h. Cysts were incubated with primary antibodies in IF-buffer, overnight at 4 °C. Samples were washed 3 times in IF-buffer (5 min each) and incubated with the appropriate secondary antibody and/or phalloidin in IF-buffer for 1 h in the dark. Samples were washed 3 times in IF-buffer (5 min each) in the dark, incubated with Dapi (5 min) and rinsed once with PBS for 5 min in the dark. To reduce the diffusion of antibodies, samples were fixed with 2% PFA for 9 min in the dark at RT. After 3 washes (5 min each) with PBS/glycine in the dark, samples were rinsed with PBS (5 min) in the dark and stored in PBS in the dark for confocal analysis. Images were acquired with confocal microscopes of the Leica SP series (see below for the detailed information) equipped with a HCX PL APO 40×/0.75–1.25 oil immersion objective.

**Podocalyxin trafficking assays**. PODXL re-localization at VAC (Vacuolar Apical Compartment)[32,67]. MDCK cells were grown in normal medium till confluency. Monolayers were then washed 3 times in PBS 1×(w/o Ca2+) and cultured in medium without Ca2+ for 2, 5, 8 h. Cells were then fixed in 4% PFA, permeabilized and subjected to immunofluorescence analysis. In control experiments to test the anti-PODXL antibody that recognizes the N-Terminal extracellular domain of the protein (Fig. SI3B), MDCK cells monolayers were pre-incubated with anti-PODXL antibody for 1 h. After incubation cells were washed 3 time in cold PBS 1x and fixed in 4% PAF for 10 min without permeabilization (−S −P), subjected to surface stripping with 0.5% acetic acid pH3 + 0.5 M NaCl, washed 3 time in cold PBS 1x and fixed in 4% PAF for 10 min without permeabilization (+S −P) or subjected to surface stripping with 0.5% acetic acid pH3 + 0.5 M NaCl, washed 3 time in cold PBS 1×, fixed in 4% PAF for 10 min, and then permeabilized (PBS 1×, 0,1% Triton X-100, 10 min) (+S +P).

**RFP-RAB35 localization at AMIS**. MDCK cysts WT control or IRSp53-KO, stably expressing RFP-RAB35, were fixed and processed for epifluorescence to visualize RFP-RAB35 (red) and stained with anti-PODXL (green), anti-β-catenin (magenta) and Dapi (blue) and images were acquired with a Leica SP8 confocal microscope equipped with a HCX PL APO 40×/0.75–1.25 oil immersion objective. PODXL staining at AMIS and RFP-RAB35 staining were adopted to design, using ImageJ software, masks contouring the AMIS and the whole cell body (in yellow). The area of the AMIS was subtracted to the area of the whole-cell bodies to obtain the "non AMIS" cell area. RFP signal (Integrated density) was determined for each cell using ImageJ software in the AMIS vs "non AMIS" area. RFP-RAB35 ratio was calculated by dividing the signal at the AMIS over the signal of the "non AMIS" area.

**Confocal spinning-disc time-lapse microscopy**. MDCK cells were seeded as single cells on Matrigel-coated coverslips in complete medium containing 2% Matrigel (BD Biosciences). Supplementary Movie 1. Six hours after seeding, images were acquired (15 h, 5 min time interval) with the UltraVIEW VoX (Perkin Elmer) spinning disk confocal unit. Images were acquired with a 60X oil immersion objective (Nikon PLANAPo VC 1.4 NA). Supplementary Movies 2 and 3. Six hours after seeding, images were acquired (15 h, 5 min time interval) using a Confocal Spinning Disk microscope (Olympus) equipped with IX83 inverted microscope provided with an IXON 897 Ultra camera (Andor), using a 60x UPlanSApo 1.35NA objective. The system is driven by the Olympus CellSens Dimension 1.18 software (Build 16686). The experiments were performed using an environmental microscope incubator (OKOLab) set to 37 °C and 5% CO2 perfusion.

**Correlative light and electron microscopy (CLEM)**. MDCK cells were trypsinized to a single cell suspension at $4 \times 10^4$ cells/ml in complete medium containing 2% Matrigel (BD Biosciences). Cell-medium-Matrigel suspensions (350 μl) were plated into 35 mm dish with gridded coverslips (MatTek Corporation, P35G-1.5-14-CGRD), pre-coated with 20 μl of Matrigel (10 mg/ml). 2% Matrigel complete medium was added to 2 ml. After 16–18 h cells were fixed with 150 mM Hepes pH 7.5, 4% PFA, 0.05% Glutaraldehyde (5 min RT) and 150 mM Hepes pH 7.5, 4% PFA (10 min RT—3 times). Cells were then washed with PBS 1×(5 min RT—3 times), blocked and

permeabilized in PBS 1×, 1% BSA, 0.1% Saponin (1 h 30 min RT). Primary antibody was then added in PBS 1×, 1% BSA, 0.1% Saponin (4 h RT). After 3 washes in PBS 1X cells were incubated with secondary antibody (+ Dapi) in PBS 1×, 1% BSA, 0.1% Saponin (2 h RT) then washed in PBS 1×(3 times) and kept in PBS.

Samples were analyzed with SP8 confocal microscope (20X dry 0.55 NA objective) to identify cells of interests on the gridded coverslips. After selection samples were fixed with 4% paraformaldehyde and 2,5% glutaraldehyde (EMS, USA) mixture in 0.2 M Sodium Cacodylate pH 7.2 for 2 h at RT, followed by 6 washes in 0.2 M Sodium Cacodylate pH 7.2 at RT. Then cells were incubated in 1:1 mixture (volume:volume) of 2% osmium tetra oxide and 3% potassium ferrocyanide (final concentration 1% osmium tetra oxide and 1.5 % potassium ferrocyanide) for 1 h at RT followed by 6 times rinsing in 0.2 M Sodium Cacodylate buffer (pH 7.2). Then the samples were sequentially treated with 0.3% Thiocarbohydrazide in 0.2 M Sodium Cacodylate buffer (pH 7.2) for 10 min and 1% OsO4 in 0.2 M Sodium Cacodylate buffer (pH 6,9) for 30 min. Samples were rinsed with 0.1 M Sodium Cacodylate (pH 6.9) buffer until all traces of the yellow osmium fixative have been removed, washed in de-ionized water, treated with 1% uranyl acetate in water for 1 h and washed in water again[68]. The samples were subsequently subjected to de-hydration in ethanol, and embedded in Epoxy resin at RT and polymerized for at least 72 h in a 60 °C oven.

*Immunolabeling with gold particles*: Samples, analyzed with SP8 confocal microscope (20X dry 0.55 NA objective) to identify cells of interests on the gridded coverslips (as described above), were fixed with a mixture of 4% paraformaldehyde and 0.05% glutaraldehyde in 0.15 M Hepes for 5 min at RT and then replaced with 4% paraformaldehyde in 0.15 M Hepes for 30 min. Afterward, the cells were washed 3 times in PBS and incubated with blocking solution (0,5 % BSA, 0,1 % saponin, 0,27 g NH4Cl in 0.2 M HEPES pH 7.2) for 30 min at RT. Then cells were incubated with primary antibody diluted in blocking solution overnight at 4 °C. On the following day, the cells were washed 3 times with PBS and incubated with goat anti-rabbit or anti-mouse Fab' fragments coupled to 1.4 nm gold particles (diluted in blocking solution 1:100) for 2 h and washed 6 times with PBS. Meanwhile, the activated GoldEnhanceTM-EM (Nanoprobes, Yaphank NY, USA) was prepared according to the manufacturer's instructions and 100 μl were added into each sample well. The reaction was monitored by a conventional light microscope and was stopped after 5-10 min when the cells had turned "dark enough" by washing several times with PBS. Then cells were fixed, stained and processed as above.

*Sectioning*: In order to get EM images of the cell of interest, cells were grown in three-dimensional Matrigel (collagen) matrix, which was prepared in such a way to get the minimal thickness of the matrix layer (~100 μm). The cells at the early stages of cystogenesis, as judged by the initial recruitment of PODXL at the AMIS, were selected during the analysis of the MatTek dish; the optical sectioning and the Z-stacking were then performed using confocal microscope. During Z-stacking the distance between the bottom and the surface of the cell was estimated to facilitate the trimming of the sample after its embedding into Epon, to find our pair quickly and to economize slot grids. Z-stack images were printed and during trimming of the pyramid and its sharpening these images were constantly used. Embedded samples were then sectioned with diamond knife (Diatome, Switzerland) using Leica EM UC7 ultra microtome. In order to start sectioning exactly when the surface of sectioning reached the cell surface, we used display of the ultramicrotome section counter, which shows the total advance and the total number of sections cut from the moment of CLEAR setting. For the trimming and advance to ROI we used the Diatome diamond trimming blades (trimtool 45, and histo) (Diatome, Switzerland). In order to avoid mistakes we took into consideration the possible "shrinkage" of the matrix during its dehydration and embedding into Epon. Thus, when only 3 μm were left before the beginning of the cell surface we replaced the trimming histo-knife with the Ultra 35 knife (Diatome, Switzerland), and then cut two 200-nm sections and then a small series of 70 nm sections. This series then was interrupted by two 200-nm sections and then the same alternating procedure was repeated several times until the moment when according to our estimation of the deepness of the cell, the cell should be finished. Sections were analyzed with a Tecnai 20 High Voltage EM (FEI, now Thermo Fisher Scientific; The Netherlands) operating at 200 kV. Tilt series were recorded at a magnification of 9600×, 11,500×, 14,500×, 19,000×, or 25,000× using software supplied with the instrument.

*Immune-CLEM*: We followed the protocols described in ref. [69]. Briefly, an ultramicrotome (Leica EM UC7; Leica Microsystems, Vienna) was used to cut 60 nm serial thin sections and 200 nm serial semi-thick sections. Sections were collected onto 1% Formvar films adhered to slot grids. Both sides of the grids were labeled with fiduciary 10 nm gold (PAG10, CMC, Utrecht, the Netherlands). Tilt-series were collected from the samples from ±65° with 1° increments at 200 kV in Tecnai 20 electron microscopes (FEI, now Thermo Fisher Scientific, Eindhoven, the Netherlands) and recorded at a magnification of 9600×, 11,500×, 14,500×, 19,000× or 25,000× using software supplied with the instrument. The nominal resolution in our tomograms was 4 nm, based upon section thickness, the number of tilts, tilt increments, and tilt angle range. The IMOD package and its newest viewer, 3DMOD 4.0.11, were used to construct individual tomograms and for the assignment of the outer leaflet of organelle membrane contours, CLEM was performed as described[70].

**Immunohistochemistry**. Mouse tissues and organs were dissected just after the mice were sacrificed by CO₂ asphyxiation or cervical dislocation.

Human tissue samples were obtained from the University of Palermo, School of Medicine, Istituto di Patologia Generale, Palermo; the European Institute of Oncology (IEO), Milano. Human breast cancer tissue samples were collected according to the Helsinki Declaration and the study was approved by the University of Palermo Ethical Review Board (approval number 09/2018). The cases were classified according to the World Health Organization classification criteria of the Tumors of the Breast.

Four-micrometers-thick sections obtained from paraffin-embedded tissues were stained with H&E for histomorphological evaluation. For immunostaining, tissue sections were dewaxed and rehydrated. The antigen unmasking technique was performed using Novocastra Epitope Retrieval Solutions, pH6 citrate-based buffer and pH9 EDTA-based buffer in thermostatic bath at 98 °C for 30 min. After the sections were brought at room temperature, the neutralization of the endogenous peroxidase with 3% H2O2 and protein blocking by a specific protein block were performed. For immunostaining on Human kidney, colon, gastric mucosa, prostate, salivary gland and breast samples, Mouse kidney samples and Zebrafish kidney samples from wild type IRSp53 and IRSp53 −/− phenotypes, the following primary antibodies were used: Mouse anti-human IRSp53 (1,5 h at room temperature, dilution 1:200 pH6), Rabbit anti-mouse IRSp53 (overnight at 4 °C, dilution 1:50 pH9, HPA023310 Sigma), Rabbit anti-zebrafish IRSp53 (overnight at 4 °C, dilution 1:30, pH9), Rat anti-mouse Endomucin, (overnight at 4 °C, dilution 1:100); Rabbit anti-mouse PKC ζ (overnight at 4 °C, dilution 1:100 pH9, sc-216 Santa Cruz); Rabbit anti-mouse ZO-1 (1 h at room temperature, dilution 1:100 pH9, GTX108592 Genetex). The immunostaining was revealed by either a polymer detection method (Novolink Polymer Detection Systems Novocastra Leica Biosystems Newcastle Ltd Product No: RE7280-K) and following specific secondary antibodies: donkey anti-rabbit IgG (H&L) specific secondary antibody 1:500 (Novex by Life Technologies) and goat anti-rat IgG (H&L) specific secondary antibody 1:500 (Novex by Life Technologies) and AEC (3-Amino-9-Ethylcarbazole) substrate was used as chromogen. The slides were counterstained with Harris hematoxylin (Novocastra, Ltd). In double-marker immunofluorescence (IF) staining, tissue sections were incubated with the following primary antibodies: Mouse anti-human IRSp53, (1,5 h at room temperature, dilution 1:200), Rabbit anti-human ZO-1 (overnight at 4 °C, dilution 1:100 pH9, GTX108592 Genetex) and Rabbit anti-human Laminin (overnight at 4 °C, dilution 1:25 pH6, ab11575 Abcam). The following secondary antibodies were used: Alexa fluor 568-conjugated goat anti-mouse, Alexa fluor 488-conjugated goat anti-rabbit (Life Technologies). Nuclei were counterstained with DAPI (4′,6-diamidin-2-fenilindolo). Sections were analyzed with: (1) Zeiss Axio Scope A1 optical microscope (Zeiss, Germany) and microphotographs were collected using an Axiocam 503 Color digital camera with the ZEN2 imaging software (Zeiss Germany); (2) Olympus BX63 Upright microscope equipped with a motorized stage Black and white camera: Hamamatsu Orca AG (12 bit, 6.45 um pixel size) and Color Camera: Leica DFC450C (36 bit, 3.4 um pixel size); (3) Scan Scope XT device and the Aperio Digital pathology system software (Aperio; Leica).

**Standard Cell lysis for WB analysis**. After washing with PBS 1×, cells were lysed in cell lysis buffer (Hepes pH 7.5 50 mM, NaCl 150 mM, glycerol 1%, Triton X-100 1%, MgCl2 1.5 mM, EGTA 5 mM, 1 mM DTT, EDTA-Free PIC) directly on the plates using a cell-scraper. About 250 μl of buffer/10 cm plates were used. Lysates were incubated on ice for 10 min and spun at 13,200 rpm for 10 min at 4 °C. The supernatant was transferred into a new tube and protein concentration was measured by the Bradford assay (Biorad), following manufacturer's instructions.

**SDS polyacrylamide gel electrophoresis (SDS PAGE)**. Pre-casted acrylamide gels (BIORAD or Thermo Fisher Scientific) were employed for resolution of proteins.

**Western blot analysis**. Desired amounts of proteins were loaded onto 0.75–1.5 mm thick polyacrylamide gels for electrophoresis (Biorad). Proteins were transferred in Western transfer tanks (Biorad) to nitrocellulose (Schleicher and Schuell) in Western Transfer buffer 1×(diluted in 20% methanol) at constant Voltage (100 V for 1 h or 30 V overnight). PonceauS coloring was used to reveal roughly the amount of proteins transferred on the filters. Filters were blocked 1 h (or overnight) in 5% milk or 5% BSA in TBS 0.1% Tween (TBS-T).

After blocking, filters were incubated with the primary antibody, diluted in TBS-T 5% milk, for 1 h at room temperature, or overnight at 4 °C, followed by three washes of 5 min each in TBS-T and then incubated with the appropriate peroxidase-conjugated secondary antibody diluted in TBS-T for 1 h. After incubation with the secondary antibody, the filter was washed three times in TBS-T and the bound secondary antibody was revealed using the ECL (Enhanced Chemiluminescence) method (Amersham).

All uncropped blots can be found in the Source Data file.

**Co-immunoprecipitation assay**. Lysates prepared in IP buffer (40 mM Hepes pH7.5, 150 mM NaCl, 10 mM MgCl2, 2 mM EDTA, 0.3% CHAPS, 10 mM NaPyr, 50 mM NaF, 10 mM NaVan, 2 mM PMSF, 1 mM DTT, PIC) were incubated in the presence the anti-FLAG M2 affinity gel (SIGMA; Fig. 5a), anti-GFP mAb agarose (MBL; Fig. 5d) or control antibodies (Ctr) for two cycles of 1 h each at 4 °C with

rocking. Immunoprecipitates were washed 3 times in IP buffer. After washing, beads were resuspended in 1:1 volume of 2× SDS-PAGE Sample Buffer, boiled for 10 min at 95 °C, centrifuged for 1 min and then loaded onto polyacrylamide gels.

**Overlay assay**. Nitrocellulose membranes were incubated in TBST 0.1% Triton X-100 buffer and let do dry. Equal or increasing amounts of recombinant-purified proteins were spotted on membranes and let dry. Membranes, previously blocked in TBST 0.1% Triton X-100 5% Milk, were then incubated with the recombinant protein of interest, resuspended in TBST 0.1% Triton X-100 5% Milk, 1–2 h at 4 °C. After extensive washing in TBST 0.1% Triton X-100, membranes were subjected to western blot analysis with the desired antibodies.

**GST-RAB35/ IRSp53 in vitro binding**. GST-RAB35WT was loaded with either GDP (SIGMA, 1 mM) or GTPγS (SIGMA, 0.2 mM) in loading buffer (25 mM Tris pH 7.5, 100 mM NaCl, 10 mM EDTA, 5 mM MgCl2, 1 mM DTT), 1 h 37 °C. Reactions were stopped by adding 20 mM MgCl2. Equal amount (2.5 μM) of GST-RAB35S22N, GST-RAB35Q67L, GST-RAB35WT-GDP, GST-RAB35-GTPγS or GST as a control were incubated with 1 μM of recombinant purified His-IRSp53 in IVB buffer (50 mM Tris pH 7.5, 100 mM NaCl, 10 mM MgCl2, 0.1% Triton X-100, 0.05% BSA), 1 h at 4 °C. Reactions were washed 3 times in washing buffer (IVB buffer w/o BSA). After washing, beads were resuspended in 1:1 volume of 2x SDS-PAGE Sample Buffer, boiled for 10 min at 95 °C, centrifuged for 1 min and then loaded onto polyacrylamide gels.

**RBD in vitro binding**. Cell lysates from MDCK Ctr or IRSp53-KO transfected with myc-RAB35WT or MDCK Ctr transfected with myc-RAB35Q67L (lysis buffer: 50 mM HEPES pH 7.2, 150 mM NaCl, 10 mM MgCl2, 1 % Triton X-100, EDTA-free PIC) were incubated with equal amount (0.2 μM) of GST-RBD or GST as control, 1 h and 30 min at 4 °C. Reactions were washed 3 times in lysis buffer. After washing, beads were resuspended in 1:1 volume of 2x SDS-PAGE Sample Buffer, boiled for 10 min at 95 °C, centrifuged for 1 min and then loaded onto polyacrylamide gels.

**GST-fusion and His-fusion proteins production**. All the GST and His fusion proteins used were product in bacteria using *E. coli* BL21 Rosetta (DE3) competent cells transformed with the pGEX6P1 or pTRC-His vector in which the desired construct had been cloned.

**Bacterial culture**. *E. coli* BL21 Rosetta (DE3) cells picked from individual colonies, transformed with the indicated GST-fusion, were used to inoculate 200 mL of LB medium (containing ampicillin at 50 μg/mL) and were grown overnight at 37 °C. Between 10 and 100 mL of the overnight culture was diluted in 1 l of LB and was grown at 37 °C (240 rpm shaking) till it reached approximately OD = 0.4–0.6.

IPTG (1 mM) was then added used to induce the protein production. After the induction cells were pelleted down at 6000 rpm for 15 min at 4 °C and pellets were used immediately or conserved at −80 °C after washing in PBS 1×.

**GST-fusion protein**. Pellets were suspended in GST-lysis buffer (15 mL for 1 L culture). Samples were sonicated 3 times for 30 s/each on ice and were pelleted down at 13,200 rpm for 30 min at 4 °C using a JA 20 Beckman rotor or at 40,000 rpm for 45 min at 4 °C using a 55.2 Ti Beckman rotor. A total of 1 mL of glutathione-sepharose beads (Amersham), previously washed 3 times with GST-lysis buffer, was added to the supernatant and samples were incubated 1–2 h at 4 °C while rocking. Beads were then washed 3 times (with 5 min of incubation at 4 °C each) in the GST lysis solution. GST-proteins were resuspended 50% slurry in the GST-lysis solution. The quantification was achieved in an SDS PAGE gel using a titration curve with BSA.

**GST-lysis buffer**. 2x TBS
 0.5 mM EDTA
 10% Glycerol
 Protease inhibitor cocktail (Roche, Basel, Switzerland)(freshly added)
 1 mM DTT (freshly added).

**His-fusion protein**. Pellets were resuspended in His-lysis buffer (10 ml every liter of culture); samples were sonicated 3 times for 30 s/each on ice and were pelleted down at 13,200 rpm for 30 min at 4 °C using a JA 20 Beckman rotor or at 40,000 rpm for 45 min at 4 °C using a 55.2 Ti Beckman rotor. A total of 600 μL of NiNTA beads (Qiagen), previously washed 3 times with His-lysis buffer, was added to the supernatant and samples were incubated 1–2 h at 4 °C while rocking. Beads were then washed 2 times in washing buffer 1 and 1 time in washing buffer 2 (5 min, 4 °C).

**Elution**. Beads were packed in Poly-Prep® Chromatography Columns (BioRad) and eluted with Elution buffer (500 μl fractions). Fractions, evaluated by Bradford assay and SDS-PAGE, were pooled together and buffer exchange with Exchange buffer, using PD10 columns (GE Healtcare). Samples were diluted 1:2 in Dilution buffer

and loaded on RESOURCE S cation exchange chromatography column (GE Healtcare) (settings for ResS run: 1CV buffer A, than gradient 0→50%B 60CV, fractions: 1.2 ml). Fractions were pulled, concentrated in S200 10/30 column equilibrated with Modified storage buffer, flash frozen and stored at −80 °C.

**His-lysis buffer**. 50 mM Tris pH8
 300 mM NaCl
 10 mM imidazole
 1 mM ß-mercaptoethanol
 proteases inhibitors
 10% glycerol

**Washing buffer 1**. 20 mM imidazole
 600 mM NaCl
 50 mM Tris pH8
 1 mM ß-mercaptoethanol
 10% glycerol

**Washing buffer 2**. 40 mM imidazole
 300 mM NaCl
 50 mM tris pH 8
 1 mM ß-mercaptoethanol
 10% glycerol

**Elution buffer**. 200 m M imidazole
 50 mM Tris pH8
 200 mM NaCl
 1 mM ß-mercaptoethanol
 10% glycerol

**Exchange buffer**. 50 mM Tris pH6.8
 100 mM NaCl
 5% glycerol
 1 mM DTT
 0.5 mM EDTA

**Dilution buffer**. 50 mM Tris pH 6.8
 5% glycerol
 0.5 mM EDTA

**ResS buffer A**. 50 mM NaCl
 50 mM Tris pH 6.8
 5% glycerol
 1 mM EDTA
 1 mM DTT

**ResS buffer B**. 1 M NaCl
 50 mM Tris pH 6.8
 5% glycerol
 1 mM EDTA
 1 mM DTT

**Modified storage buffer**. 50 mM Tris 7.5
 200 mM NaCl
 1 mM DTT
 10% glycerol

**SILAC BioID of IRSp53 interactors**. BirA* expressing vector was purchased from Addgene (plasmid # 35700). BirA* was cloned upstream of IRSp53 or GFP, as control, in lentiviral pLVX vector (Clonetech). HeLa cells were infected with lentiviral particles generated from pLVX BirA*-IRSp53 or pLVX-BirA*-GFP. The two populations have been treated with heavy (BirA*-GFP) and light (BirA*-IRSp53) labeled DMEM medium for SILAC (Thermo Fisher Scientific) (supplemented with dialyzed fetal bovine serum) for 6 passages; swap of labeling was performed in the replicate. Cells have been treated with 50 μM biotin for 24 h and lysed in cell lysis buffer (Hepes pH 7.5 50 mM, NaCl 150 mM, glycerol 1%, Triton X-100 1%, MgCl$_2$ 1.5 mM, EGTA 5 mM, 1 mM DTT, EDTA-Free PIC), then biotinylated proteins were fished out with streptavidin-conjugated magnetic beads (Thermo Fisher Scientific). Beads were washed out from aspecific binders using 50 mM Tris pH 7.5, 1% SDS and protease inhibitors; elution was performed using SDS-PAGE Sample buffer 2×, coupled with 95 °C boiling. Eluates were then subjected to SDS-PAGE; after colloidal blue staining, bands were excised and reduced with 10 mM dithiothreitol, alkylated with 55 mM iodoacetamide and finally digested overnight with 12.5 ng/μl of trypsin (Roche). After acidification, peptide mixtures were concentrated and desalted, dried in a Speed-Vac and resuspended in 12–15 μL of solvent A (2% acetonitrile, 0.1% formic acid). A total of 5 μl of each digested

sample from the forward and reverse experiments were loaded on a LC (liquid chromatography)–ESI–MS/MS quadrupole Orbitrap QExactive mass spectrometer (Thermo Fisher Scientific). Peptides were separated on a linear gradient from 95% solvent A to 40% solvent B (80% acetonitrile, 0.1% formic acid) over 30 min and from 40 to 100% solvent B in 3 min at a constant flow rate of 0.25 μl/min on UHPLC Easy-nLC 1000 (Thermo Scientific), where the LC system was connected to a 25-cm fused-silica emitter of 75 μm inner diameter (New Objective, Inc. Woburn, MA, USA), packed in-house with ReproSil-Pur C18-AQ 1.9 μm beads (Dr Maisch Gmbh, Ammerbuch, Germany). MS data were acquired using a data-dependent top 12 method for HCD fragmentation. Survey full-scan MS spectra (300–1650 Th) were acquired in the Orbitrap with 70,000 resolution, AGC target 3e6, IT 60 ms. For HCD spectra, resolution was set to 17,500 at m/z 200, AGC target 1e5, IT 120 ms; Normalized Collision energy 25% and isolation with 2.0 m/z. Technical replicates were conducted on the LC–MS-MS part of the analysis. Raw data were processed with MaxQuant ver. 1.4.1.2. Peptides were identified from the MS/MS spectra searched against the UniProt_Human_2014_10 database using the Andromeda search engine in which trypsin specificity was set up with a maximum of two missed cleavages. Cysteine carbamidomethylation was used as fixed modification, methionine oxidation and protein N-terminal acetylation as variable modifications. Mass deviation for MS/MS peaks was set at 20 ppm. The peptides and protein false discovery rates (FDR) were set to 0.01; the minimal length required for a peptide was six amino acids; a minimum of two peptides and at least one unique peptide was required for high-confidence protein identification. The lists of identified proteins were filtered to eliminate reverse hits and known contaminants. For quantitative analysis, "re-quantify" and "second peptide" options were selected. The statistical program Perseus (ver. 1.5.1.6) was used to quantify significantly up and downregulated proteins following the criteria: (i) significance A with a Benjamini-Hochberg FDR < 0.05; (ii) ratio normalized values concordant in both forward and reverse experiments; (iii) minimum H/L ratio counts = 2. Proteomic data are available at the PeptideAtlas repository PASS01464.

**Zebrafish strains**. Adult zebrafish were maintained in a multi-rack system (from *Aquatic Habitats*) at a water temperature of 28 °C, pH 7 and conductivity 600 μS. Zebrafish embryos and larvae not older than 5 dpf were maintained at 28.5 °C in E3 water (50 mM NaCl, 0.17 mM KCl, 0.33 mM CaCl, 0.33 mM MgSO4, 0.05% methylene blue). Zebrafish strains used in this study are *AB* (referred to as wild type) *sa11319* obtained from European Zebrafish Resource Center (EZIRC) referred to as *baiap2a*^C201* and *Tg(cldnB:GFP)*. All the strains were maintained and bred according to the national guidelines (Italian decree "4 March 2014, n.26"). All experimental procedures were approved by the FIRC Institute of Molecular Oncology Institutional Animal Care and Use Committee and Italian Ministry of Health.

**Genomic DNA extraction from zebrafish embryos**. Caudal fin biopsies (fin clip) were incubated 10 min at 98 °C in 50 μl of lysis buffer (Tris-HCl 10 mM pH 8.0, EDTA 1 mM, 0.3% Tween, 0.3% NP40) followed by the ice cooling. A total of 5 μl of Proteinase K 10 mg/ml (Sigma-Aldrich) were added and samples were incubated at 55 °C O/N. The second day, 145 μl of sterile water were added, followed by 20 μl of Sodium Acetate and 200 μl of Phenol. Samples were mixed by inverting them and centrifuged at 13,000 rpm for 1 min. Supernatant was collected and precipitated O/N with 100% ethanol at −20 °C. The third day, samples were centrifuged for 30 min at 4 °C and recovered pellets were washed with 75% ethanol, centrifuged again for 5 min and resuspended in 20 μl of DNAse-free water.

***sa11319* genotyping**. Genomic DNA (gDNA) was extracted from the caudal fin biopsies of adult *sa13359* fish (generated with ENU at Sanger Institute) and a fragment of 420 bp containing *baiap2a* mutation was amplified by PCR with the specific primers forward 5′-TGTTGAGGCCATCAGCAGTA-3′ and reverse 5′-CAAACTGTGCCCAATGGAG-3′ and sequenced (Cogentech Sequencing Facility). Only heterozygous fish were maintained and in-crossed to obtain omozygous embryos and adult fish.

**RNA extraction from zebrafish, cDNA synthesis and RT-PCR**. Wild type zebrafish larvae (AB strain) were collected at 2, 6, 24 and 48 hpf and RNA was extracted using TRIZOL Reagent (Invitrogen) and RNAse Mini kit (QIAGEN). To avoid genomic DNA contamination, samples were digested with RQ1 RNase-Free DNase (Promega). The cDNA was retrotranscribed from 1 μg of RNA using SuperScript VILO cDNA Synthesis kit (Invitrogen), according to manufacturer instructions. 500 ng of cDNA were used as template for a semi-quantitative RT-PCR using the following primers: 5′-ACGGAGTGTCTCAGGGAAGA-3′ and 5′-TTCTCTCCATAGTGCCAGCC-3′ for *baiap2a*, 5′-TTGGAGAGAAATGGAC-3′ and 5′-CGTGTAGGAGAACGGGAACCA-3′ for *baiap2b*. β-actin was used as housekeeping.

**Hematoxylin and eosin staining and immunostaining on paraffin sections**. Larvae were fixed O/N at 4 °C in 4% PFA diluted in PBS and positioned in a 7 × 7 × 6 mm plastic base-molds (Kaltek) containing 1.2% low-melting agarose in PBS. Before agarose solidification, larvae were correctly oriented. After agarose block solidification, larvae were removed from the base mold and immersed in 70%

ethanol. After dehydration, agarose blocks were subjected to paraffin embedding by Leica ASP300 S Fully Enclosed Tissue Processor and 5 µm thick sections were cut using a manual rotatory microtome (Leica). Sections were stained with Harris hematoxilin solution for 2 min, washed in running water for 5 min, counterstained with Eosin-Y solution for 7 s and washed in running tap water for 5 min. Sections were dehydrated with 95% ethanol and 100% ethanol for 5 min two times. Then, they were cleared two times with xylene for 5 min and mounted on a glass slide. Sections were finally imaged using a Nikon Eclipse 90i microscope, respectively, with 20X and 100X objectives. For immunostaining, sections were incubated in sodium citrate buffer (2.94 mg/ml tri-sodium citrate pH 6, 0.05% Tween 20) at 95 °C for 45 min and cooled at RT for 1 h under chemical hood. Sections were then incubated in blocking solution (2% fetal bovine serum, 2 g bovine serum albumin, 0.05% Tween 20 in PBS 1X adjusted at 7.2 pH) for 1 h at RT followed by IRSp53 primary antibody diluted in blocking solution O/N. Samples were rinsed in PBS 1X three times for 5 min. The antibody signal was revealed with DAB and haematoxylin staining was performed, then sections were dehydrated and mounted with Eukitt-mounting medium.

**Zebrafish whole-mount IF and sections**. Three days post fertilization larvae were fixed O/N at 4 °C with 4% PFA diluted in PBS 1X and rinsed 3 times with PBS 1×. Embryos older than 24 hpf were treated with 0.25% trypsin (Sigma-Aldrich) at RT for a range of time between 2 min (for 24 hpf embryos) up to 60 min (for 5 dpf larvae). Samples were then rinsed 3 times for 5 min with washing buffer (1% Triton-X100, 0.2% DMSO in PBS 1×) and incubated for at least 1 h in blocking buffer (0.1% Triton X-100, 1% DMSO, 5% normal goat serum in PBS 1×) on a shaker. Subsequently, embryos were incubated with primary antibodies diluted in blocking buffer O/N at 4 °C. The following day, samples were rinsed rapidly twice with washing buffer and at least 3 washes of 1–2 h each with washing buffer were performed. Samples were incubated in blocking buffer for 30 min followed by secondary antibodies diluted in blocking buffer O/N at 4 °C. The final day, samples were rapidly rinsed 2 times with washing buffer and two washes of 5 min each with PBS 1X were performed. Samples were incubated 10 min with DAPI, rapidly rinsed with PBS 1X and mounted on a glass slide in 85% glycerol. The following primary antibodies were used: chicken anti-GFP 1:1000 (Abcam, ab13970). Alexa fluor-488 (Invitrogen) were used as secondary antibodies. Pronephric ducts were first acquired using a Leica SP8 confocal microscope equipped with 63X immersion oil objective, deconvolved with a Huygens software (Fig. 8a and Supplementary Movie 5) and then elaborated (Supplementary Movie 5) with a 3D rendering processing (Scientific Volume Imaging).

**Vibratome section**. To prepare 80 µm transversal sections, 3 dpf embryos previously stained were cutted in PBS with a vibratome after inclusion in 5% low-melting agarose. Sections obtained were equilibrated and mounted in glycerol 85% in PBS on glass slides and observed under a Leica TC-SP2 confocal microscope.

**Paraffin section**. 3 dpf embryos were fixed, included and processed as already described. A total of 4 µm sections were dewaxed and rehydrated. The antigen unmasking technique was performed using Novocastra Epitope Retrieval Solutions, pH6 citrate-based buffer in thermostatic bath at 98 °C for 30 min. After that, the sections were brought at room temperature and stained with chicken anti-GFP 1:1000 (Abcam) and anti-acetyl-α-tubulin 1:500 (Abcam). Alexa fluor-488 and −561 1:400 (Invitrogen) were used as secondary antibodies and Dapi to counterstain nuclei. Images were acquired using a Leica SP8 confocal microscope equipped with 63X immersion oil objective.

**Morpholino injections**. Zebrafish embryos were microinjected at 1 cell stage with an Olympus SZX9 and a Picospritzer III microinjector (Parker Instrumentation). Injection mixes were composed of Danieau solution 1×(NaCl 58 mM, KCl 0.7 mM, MgSO$_4$ 0.4 mM, Ca(NO$_3$)$_2$ 0.6 mM, HEPES 5.0 mM, pH 7.6), Phenol red 0.1% and Morpholino (MO) antisense oligos (Gene Tools). Each embryo was injected, respectively, with 1.7 ng of splice-blocking *baiap2b* MO, 5′-TTCGGGCACTACA TGAGTGACCTT-3′, and 1.7 ng of 5′-UTR *baiap2b* MO, 5′-AAAGGTCACTCAT GTAGTCGCCGAA-3′, together.

**Western blot analysis**. Larvae were lysed in sample buffer; 40 µg of total extracts were resolved by SDS-PAGE, transferred to nitrocellulose and tested with the IRSp53 (1:50) antibody. Band intensities were quantified using ImageJ.

## Microscopes

**Widefield**. Upright Olympus BX51 FL, equipped with 60× UPlanApo 1.35 NA oil and a Photometrics Cool SnapEZ camera.
Software: MetaVue.

**Confocal spinning disk**. UltraVIEW VoX (Perkin Elmer) spinning disk confocal unit, equipped with an EclipseTi inverted microscope (Nikon), a C9100-50 EMCCD camera (Hamamatsu) and driven by Volocity software (Improvision, Perkin Elmer).

Confocal Spinning Disk microscope (Olympus), equipped with IX83 inverted microscope, an IXON 897 Ultra camera (Andor) and driven by the Olympus CellSens Dimension 1.18 software (Build 16686).

**Confocal SP5inv**. Leica TCS SP5 laser confocal scanner mounted on a Leica DMI 6000B inverted microscope equipped with motorized stage, HC PL Fluotar 10×/0.30NA dry objective, HC PL FLUOTAR 20×/0.5NA dry objective, HCX PL APO 40×/ 1.25–0.75NA oil immersion objective and HCX PL APO 63×/ 1.4NA oil immersion objective.
Laser lines available:
405 nm.
Argon Laser (458 nm, 476 nm, 488 nm, 496 nm, 514 nm).
DPSS 561 nm.
HeNe 633 nm.
Software: Leica LAS AF.

**Confocal SP2 AOBS**. Leica TCS SP2 AOBS laser confocal scanner mounted on a Leica DM IRE2 inverted microscope.
Objective: HCX PL APO 63×/1.4NA oil immersion objective and HCX PL APO 40×/1.25–0.75NA oil immersion objective.
Laser lines: Violet (405 nm laser diode), blue (488 nm argon laser), yellow (561 nm laser diode) and red (633 nm laser diode).
Software: Leica Confocal Software (LCS).

**Confocal SP8 white laser**. Leica TCS SP8 laser confocal scanner mounted on a Leica DMi 8 inverted microscope equipped with motorized stage. Objectives: HC PL APO CS2 20×/0,75 dry objective, HC PL APO CS2 40×/1,30 oil immersion immersion objective and HC PL APO CS2 63×/1.40 oil immersion objective.
Laser lines available:
405 nm pulsed.
Argon Laser (458 nm, 476 nm, 488 nm, 496 nm, 514 nm).
White light laser tunable in the range: 470 nm-670 nm.
Software: Leica Application Suite X (LASX) ver. 3.5.2.18963.

**Confocal SP8**. Leica TCS SP8 laser confocal scanner mounted on a Leica DMI 8 inverted microscope equipped with motorized stage. Objectives: HC PL FLUOTAR 20×/0,55 dry objective, HC FLUOTAR 25×/0,95 water objective, HCX PL APO 40×/ 0.75–1.25 oil immersion objective, HC PL APO CS2 63×/1.40 oil immersion objective.
Lasers:
405 nm diode.
Argon (458 nm, 476 nm, 488 nm, 496 nm, 514 nm).
DPSS 561 nm.
HeNe 633 nm.
Incubation system: Okolab bold line.
Software: Leica Application Suite X, ver. 3.5.1.18803.

**Statistical analysis**. All data are presented as scatter plots with bars and the mean of independent biological replicates ± SD, with the exception of Fig. 7b ($n > 300$) where the mean ± s.e.m is reported. The number of experiments as well as the number of samples analyzed is specified for each experiment and reported in the figure legends. A two-tails, student's $t$-test with Welch corrections for two samples with un-equal variance was used to calculate the $p$ values, with the exception of Fig. 4b, *bottom right graph*, and Fig. 4c, *bottom right graph*, where a one-tail student $t$-test was used. *$p < 0.05$; **$p < 0.01$, ***$p < 0.001$.

**Reproducibility**. Experiments such as IB, IF, IHC, ponceau staining, agarose gel (in Figs. 1a–d, 2a, 3b–d, a, 5a–d, 6a, d, 8a, b, 9a; and Supplementary Figs. 1A–D, 2A–D, 3A–C; 4B–E; 5A–D; 6D, 7A, 8A, B, 9B; 10A, 11A–D) are representative of at least two independent experiments with similar results, unless otherwise indicated.

**Reporting summary**. Further information on research design is available in the Nature Research Reporting Summary linked to this article.

## Data availability

All full scan immunoblots, gels and all data set used to generate each of the graph presented in the work are provided in the Source Data file. Specifically, the source data underlying Figs. 2a–d, 4b, c, 6a, b, 7a, 9b and Supplementary Figs. 3d, e, 4a, 6a, b, and 9a are provided as a Source Data file. The authors declare that all of the data that support the findings of this study are available within the paper and its Supplementary Information files, and from the corresponding authors upon reasonable request. Proteomic data have been deposited and are available at the PeptideAtlas repository PASS01464. Source data are provided with this paper.

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

## Acknowledgements

We thank: Fernando Martin-Belmonte and Gregory Emery for critically reading the manuscript; Arnoud Echard for the GST-RAB35 WT construct; Chiara Luise and Giovanna Jodice for technical assistance with immunohistochemistry experiments; Ilaria Costa and the IFOM Imaging Facility for technical assistance in the design and performing of the confocal imaging. This study was supported by: the Associazione Italiana per la Ricerca sul Cancro (AIRC-IG#18621 to GS, AIRC-0IG#22145 to CT, and 5XMille #22759 to GS and CT); the Italian Ministry of University and Scientific Research (MIUR) to GS (PRIN: Progetti di Ricerca di Rilevante Interese Nazionale – Bando 2017#2017HWTP2K); the Italian Ministry of Health (RF-2013-02358446) to GS. SB was supported by Fellowships from AIRC. AR was a PhD student within the European School of Molecular Medicine (SEMM), University of Milan, Milan, Italy.

## Author contributions

S.B., A.R., S.M. and A.D. designed and performed all the experiments and edited the manuscript; D.C. and Ghaz.S. aid in generating cell lines and in the analysis of immunofluorescence; G.D. conducted, analyzed and interpreted all the experiments using zebrafish; A.D. and G.v.B. and A.M. perfomed all of the electron microscopy studies, I.F., G.B., F.P., S.P., G.V. and C.T. collected the specimens, and performed and interpreted all the immunohistoichemical studies in human murine and zebrafish tissues: A.C. and A.B. performed and analyzed the mass spectrometry with S.B. and A.O. performed and analyzed the light imaging data; A.D. and G.S. conceived the whole study, wrote the manuscript, and supervised the whole project.

## Competing interests

The authors declare that they have no competing interests.
