## [Peer Review File · Nature Communications]

Reviewers' comments:

Reviewer #1 (Remarks to the Author):

This is an interesting study that is initiated by a striking localization of IRSp53 to the apical luminal membrane in a variety of epithelia (Fig1). Subsequently, an epithelial culture model of MDCK cells is utilized to examine the effects of IRSp53 KO, which seems to disrupt the normal polarization and organization of these 3D cysts (Fig 2). An enrichment of IRSp53 at the forming apical interface of MDCK cysts is then shown in Fig 3, which precedes a vesicular colocalization along with apical domain destined PODXL, that is also positive for Rab7/11/35. Through KO and knockdown studies, they then establish that IRSp53 and Rab35 contribute to proper polarity and luminal integrity of MDCK cysts (Fig4). A direct interaction of recombinant IRSp53 with Rab35 is established in Fig5, though the in vivo interaction of these proteins was performed in cells over-expressing FP-tagged versions of Rab35. A functional association of these proteins and other factors (such as Eps8) are established through the use of I-BAR/SH3 domain mutant versions of IRSp53, along with Eps8-KO cells (Fig5-6). Ultra-structural analysis of these apical borders in IRSp53 KO cells were analyzed and presented in Fig 7. Well done EM show some defects in the organization of these apical membranes. The authors then extend these studies to the zebrafish model utilizing KO's of the IRSp53 paralog biap2a (Fig8, Fig1D, and supplemental material). Tissue sections of kidney from these KO studies emphasize the role biap2a has in epithelial polarization and duct formation.

Overall, these studies were performed well, with some minor technical improvements and suggestions mentioned below. The imaging is generally of high caliber and supports the authors claims.

Q1: There is clearly an enrichment of IRSp53 on what becomes the apical lumen in their model of cyst formation using MDCK cells in Fig 3. Its not clear where the IRSp53 originates from to enrich on the vesicles that seem to be transporting the PODXL to the forming apical membrane/lumen? Perhaps some time lapse studies of this process would provide some insights?

Q2: How specific is the colocalization of IRSp53, PODXL and Rab7/11/35 in Fig 3? The authors comment on how multiple markers such as Lamp1 did not show up in these vesicles, which seems confusing for Rab7 in particular? Perhaps show some additional negative colocalization data with better resolution than that shown in FigS3. How specifically defined are these vesicles?

Technical issues and suggestions:

-Can the authors address the remaining IRSp53 protein that appears in their Caco2 "KO" clones? There appears to be some protein that is still expressed in these cells (Fig 2c), though it is a clearly a functionally relevant decrease in this protein.

-Perhaps additional experimental repeats with the IRSp53 knockdown studies would improve the error reported and the statistical relevance for the inverted polarity data in Fig 4B. The data currently represent the results of just two experiments?

-The Y-axis label on the bar graph has a typo in Fig5E

- Some interaction studies of endogenous IRSp53/Rab35 proteins would be useful

Reviewer #2 (Remarks to the Author):

This is an interesting and data-heavy manuscript that identifies IRSp53 as a novel regulator of

apical lumen formation in renal epithelia. The data demonstrating that IRSp53 is required for lumenogenesis is solid, especially since authors make an extra effort to test it in couple of different cell lines (Caco2 and MDCK) as well as in two in vivo models (mice and zebrafish). However, the part that attempts to identify the molecular machinery of IRSp53 action is much less convincing, to some extent due to various technical/experimental issues. For example, in vitro overlay assay is not a good assay to analyze protein interactions. Better approaches need to be used, ideally ITC or Biocore, but, at very least, recombinant protein pull down assay. Similarly, it is not clear how cells picked for EM analysis were staged and how authors know whether they already formed AMIS. These issues are relatively easy to address, thus, authors should be encouraged to re-submit revised manuscript (if they can also address conceptual issues listed below). The manuscript also has some bigger conceptual issues. First, it is not clear how Rab35 and IRSp53 complex functions if IRSp53 binds to Rab35-GDP. All the work on Rab35 so far indicates that it functions in GTP-bound form, thus, it is unclear why it would be important to target Rab35-GDP to the AMIS. The even bigger conceptual issue is the need of Rab35 itself for lumen formation. While a couple years ago Echarid's lab used rab35 KD to suggest that it is involved in MDCK lumenogenesis, the recent study from Fukuda lab (that used Rab35 KO line) did not see any effect of Rab35 loss on apical lumen formation (the paper is not cited in the manuscript). Authors should generate Rab35 KO line (since Rab35 KD s may have off-target effects) and recapitulate the key experiments. Preferably, that should be followed by GF-Rab35 rescue. I realize that it is a lot of work, but since Rab35 is at the center of this manuscript, the issue whether rab35 is needed for lumen formation needs to be addressed.

Major Points:

- 1) Figure 2C. Why Caco2 IRSp53 cells still have IRSp53 protein?
- 2) Figure 4. Recent work (JCB, 2019) from Fukuda lab has demonstrated that Rab35 KO in MDCK cells does not lead to defects in lumenogenesis. That does not seem to fit with proposed model that Rab35 targets IRSp53. Additionally, authors demonstrate the effect of Rab35 KD on lumenogenesis. How do authors explain that? Can they rescue Rab35 defects with GFP-Rab35? Additionally, Fukuda paper needs to be cited.
- 3) Figure 5B-C. In vitro overlay assay is not a good method to analyze protein binding. More conventional and more informative recombinant protein pull-down assay (or even better quantitative binding assays, such as Biocore or ITC) should be used. As shown in supplemental figure 4D, data is poor quality and not-interpretible. Additionally, to rigorously test nucleotide dependency of Rab and effector protein binding, one needs to do GTPγS or GDP loading in purified recombinant wild-type Rab. That is especially important for this manuscript, since presumptive IRSp53 binding to Rab35-GDP is at the core of the finding.
- 4) Figure 5E. None of the mutants, except PDZ, (not only I-BAR) rescued IRSp53 KO. Thus, it is hard to conclude that Rab35 binding is required. Besides, K to E is pretty dramatic mutation that likely affects overall protein conformation. Finally, it is hard to believe that this binding is required since Rab35-KO does not affect lumenogenesis (see Fukuda paper).
- 5) Figure 6B. The data regarding the role of IRSp53 in targeting Rab35 is not totally convincing. Based on images, even in IRSp53 KO cells RFP-Rab35 does seem to be present at the AMIS (very moderate effect based on quantification) as well as everywhere else in lateral cell membrane.
- 6) Supplemental Figure 6. Why is such a huge difference in multi-lumenation in control and Eps8-KO cells between different experiments (compare bar graph on right and left in C)? Once again (see minor point 1), it is hard to explain that as a result of experimental variability, since standard errors are pretty small and presumably data was derived from several biological experimental replicates.
- 7) Since IRSp53 regulate ezrin (a known apical protein), the authors should test ezrin localization at the apical membrane in IRSp53 KO cells.
- 8) Figure 7. How do authors know that those two cells formed AMIS since they used b-catenin as marker. Just because they are two cells, that does not mean that they already formed AMIS. AMIS marker (such as podoclyxin) should be used.

Minor Point:

1) Figure 2B. Why multi-lumen formation in control cells in KD and KO experiments is so different? Since standard deviation is pretty small that is not likely due to variability between experiments.

Reviewer #3 (Remarks to the Author):

Comments for Bisi et al

This paper describes the possible role of IRSp53 in epithelial morphogenesis. The paper first characterized the morphology of kidney sections in knockout or knockdown animals, and found the abnormality in the tubular structures of the kidney. The localization of IRSp53 was analyzed in detail using the model cell system that forms cyst and the kidney sections. Then the authors proposed the possible role of IRSp53 in the apical-basal polarity of the tubular structure formation in cellular and kidney development. However, the involvement of IRSp53 in cyst formation in MDCK cells were already shown in JCB 192 (3)525-540 2011. Though this paper showed that the defect in cyst formation upon IRSp53 depletion can be rescued by IRSp53 expression. The possible molecular mechanism is also provided. The similar phenotype was observed by the IRSp53-interacting Eps8 knockout. Furthermore, the interaction of IRSp53 with Rab35 is proposed. The overall data are results of enormous effort to characterize the role of IRSp53. I would like to point several issues to be addressed for the clarity of the IRSp53 function,

Major points.

The function of kidney of IRSp53 KO animals might be analyzed in relation to clinical aspects, such as the blood concentrations of the chemicals that is known to be filtrated through kidney. Such things can be easily addressed by the biochemical analysis of plasma.

The proposal is the IRSp53 is involved in the tubulogenesis (apical-basal polarity), which is thought to occur during tubulogenesis of kidney or the cyst formation in cellular culture in 3D matrix. The tissue sectioning of kidney during the tubule-genesis (ideally consistent with the AMIS in kidney) might be stained with IRSp53 antibody for the consistency between cellular experiments and animal models.

The binding of IRSp53 to Rab35 is to the dominant negative form, which was suggested to promote the re-localization of podocalyxin (PODXL) to the apical surface of the cyst. The binding of active form of Rab35 to PDXL was reported (Nature commn, 2016 DOI: 10.1038/ncomms11166). The comparison of the PODXL localization upon IRSp53 and Rab35 knockdown in Figure 4B suggested that these two proteins are both involved in the localization of PODXL. What is the molecular mechanisms behind in term of Rab35 activity? How the inactive form of Rab35 regulate the IRSp53 function? The expression of various IRSp53 mutant in cells expressing active or dominant negative form of Rab35 will be helpful.

The binding of IRSp53 to Rab35 occurs at the basic amino-acid residues at the tip of the I-BAR domain structure, which is also involved in the membrane binding. On the other hand, the absence of Rab35 did not affect the IRSp53 localization (Figure S4). As the authors discussed, these basic amino-acid residues will contribute to the binding to PIP2, and therefore, it is not easy to conclude the role of possible Rab35 mediated regulation/recruitment of the IRSp53 by this mutation in the I-BAR domain (I-BAR*). At least it is better to discuss these points and ideally addressed experimentally.

The established function of IRSp53 is to induce filipodia like membrane protrusions. The IRSp53 knockdown/knockout resulted in the increased filopodia like structures in the CLEM analysis of

IRSp53 knockdown/knockout cells in 3D matrix. On the other hand, it is known that the apical surface of the tubules or cysts could contain a lot of filopodia-like brush borders. The enrichment of IRSp53 at the apical surface might just reflect the abundance of filopodia-like structures. The CLEM of apical surface with anti-IRSp53 antibody would be of interest.

The internalization of PODXL can be dependent on IRSp53. Then, IRSp53 will co-localize with PODXL during the internalization. Ideally it will be better to do some live-imaging to show that or stain the cells with also IRSp53 antibody in Figure S3B.

The illustration in Figure S11 indicated that the cytosol is connected upon knockdown of IRSp53. Is there any evidence for that, like dye transfer?

Minor points.

The binding of IRSp53 to VASP was reported (Oikawa et al, 2013, doi: 10.1371/journal.pone.0060528).

The binding of I-BAR to small GTPase is reported in some papers, and needs to be discussed (J Biol Chem. 2019 Apr 19;294(16):6494-6505., J Biol Chem. 2019 Apr 19;294(16):6494-6505. , J Cell Sci. 2007 May 1;120(Pt 9):1663-72., Nature. 2000 Dec 7;408(6813):732-5.)

The mutant construct of IRSp53 did not have the details. Does the * mean the point mutation in full-length? The western in Figure 5E appears to suggest these are full-length mutants. Then it is better to describe as amino-acid mutations.

The mass spec data might not support the conclusion by the authors, but should be presented in the text for help to readers. During the review, I could not see them.

In Figure 2B, the graph had no explanations on the difference in the two graphs..

In Figure 2C, the GFP-IRSP53 bands were cropped and not shown, please show these.

In Figure 3 and 4, the staining with ZO-1 and beta-catenin will help to see the relation between the cell-cell junction formation and the tubulogenesis.

In Figure 3C, the co-localization of endogenous proteins will be better to be analyzed. The co-localization at AMIS (or 2 cell stage as Figure 3B) will also need to be analyzed.

In Figure 5E, the localization of Rab35 and Eps8 should also be examined.

In methods, the JS buffer is unclear.

Experimental detail for Figure S3B (striping assay) was not clear for me.

Reviewer #4 (Remarks to the Author):

This manuscript indicates IRSp53 as one of the key regulators for the establishment of apical polarity. The authors use a variety of cell lines and animal models to substantiate their findings which together provide a compelling case for their conclusions.

I was asked to mainly comment on the technical aspects of the Correlative Microscopy in the manuscript which I will focus on below.

Abstract 2nd line: plasma membrane / actin

Page 4, line 10 from bottom: the these (remove one)

Page 4 line 7 from bottom: cortex to (twice)

Page 4 line 6 from bottom (the the (twice))

Control and WT are both used, sometimes unclear why / what the difference is

Figure S2 inconsistent A has GFP bottom, B and C have GFP first

Page 8, line 2: what is the evidence these are a; vesicles, b; internalized as opposed to newly synthesized?

Page 14/ Fig 7: technical aspects of CLEM data convincing and well interpreted. I do have a few observations that need clarification:

- There appear to be decreased intercellular spaces in the KO cells (compare insets Ctr and KO. Comment?

- Fig. S7: I would have expected a yellow (red and green) border in the control between the 2 cells if there was significant PODXL present. This can not be verified but from the EM overview the labelling looks higher in the plasma membrane outside the border than between the cells. In addition what is the structure of the red spot inside the cell.

There is a very large spread in the silver enhancement of the gold particles in the KO cell (much more than the control cell) closer inspection shows a lot of labelling on other structures.

Page 20, line 4: electron microscopy

Supplemental Material:

1.1 antibodies: no secondary antibodies are listed

1.3.1, 1.3.2 and 1.3.3: Don't understand why there is a difference in the IF protocols

1.3.4 line 2: monolayers were then washed

Line 5 antibody that recognizes

Page 12, line 1: blocked and permeabilized

Line 6: fixed with of

Line 9: please clarify 1:1 mixture, what are the end concentrations and in what buffer?

Sodium cacodylate / Na cacodylate / cacodylate buffer are used interchangeable, please use 1 consistently

Immunolabeling with gold particles

Line 5: what is the blocking solution?

Page 12, Bottom line: who is the manufacturer?

Page 13, from line 2 same as protocol above, redundant

Sectioning:

Line 2: Can you give an estimate what "minimal thickness" is?

Line 3: clarify "phase" : cell stage?

Line 4: MatTek dish

Rephrase line 3-5, incorrect grammar

Line 7: clarify: e.g. add "for retracing purposes" at the end of sentence

1.3.7.1: Tomography = Electron Tomography

Two-step CLEM used for first time, not essential, delete "two-step"

This paragraph should not be called Tomography but immune-CLEM but there is a lot of repetition in this section, needs to be cleaned up.

Point-by-Point Rebuttal letter

Reviewers' comment in black, our reply in Blue

Reviewer #1

This is an interesting study that is initiated by a striking localization of IRSp53 to the apical luminal membrane in a variety of epithelia (Fig1). Subsequently, an epithelial culture model of MDCK cells is utilized to examine the effects of IRSp53 KO, which seems to disrupt the normal polarization and organization of these 3D cysts (Fig 2). An enrichment of IRSp53 at the forming apical interface of MDCK cysts is then shown in Fig 3, which precedes a vesicular colocalization along with apical domain destined PODXL, that is also positive for Rab7/11/35. Through KO and knockdown studies, they then establish that IRSp53 and Rab35 contribute to proper polarity and luminal integrity of MDCK cysts (Fig4). A direct interaction of recombinant IRSp53 with Rab35 is established in Fig5, though the in vivo interaction of these proteins was performed in cells over-expressing FP-tagged versions of Rab35. A functional association of these proteins and other factors (such as Eps8) are established through the use of I-BAR/SH3 domain mutant versions of IRSp53, along with Eps8-KO cells (Fig5-6). Ultra-structural analysis of these apical borders in IRSp53 KO cells were analyzed and presented in Fig 7. Well done EM show some defects in the organization of these apical membranes. The authors then extend these studies to the zebrafish model utilizing KO's of the IRSp53 paralog biap2a (Fig8, Fig1D, and supplemental material). Tissue sections of kidney from these KO studies emphasize the role biap2a has in epithelial polarization and duct formation.

Overall, these studies were performed well, with some minor technical improvements and suggestions mentioned below. The imaging is generally of high caliber and supports the authors claims.

R. We are thankful to this reviewer for appreciating the overall quality of our work and experiments.

Q1: There is clearly an enrichment of IRSp53 on what becomes the apical lumen in their model of cyst formation using MDCK cells in Fig 3. It is not clear where the IRSp53 originates from to enrich on the vesicles that seem to be transporting the PODXL to the forming apical membrane/lumen? Perhaps some time lapse studies of this process would provide some insights?

R. We performed time-lapse studies of stably expressing GFP-IRSp53 and RFP-PODXL. Examples of these experiments are included in Suppl. Movie S1. We also performed additional time-lapses focusing, as per the reviewer's suggestions, on the dynamic distribution of IRSp53. As shown in the still images included in Fig. 3C-D and in Suppl. Movies S2-3 of the revised manuscript, we can document examples of IRSp53-positive vesicular-like structures that appear to emerge from the peripheral plasma membrane and move toward the forming apical side. We also noted, consistently with the ability of IRSp53 to associate with PI4,5P2-rich membrane, that GFP-IRSp53 intensities increases along what looks like the ingressing furrow in the early phase of cystogenesis (however, we have not included this movie in the revised version of the manuscript).

Q2: How specific is the colocalization of IRSp53, PODXL and Rab7/11/35 in Fig 3? The authors comment on how multiple markers such as Lamp1 did not show up in these vesicles, which seems confusing for Rab7 in particular? Perhaps show some additional negative colocalization data with better resolution than that shown in FigS3. How specifically defined are these vesicles?

R. We were surprised to detect IRSp53 in vesicles since other and we have reported that it is mainly localized along the plasma membrane in isolated cells in interphase¹⁻¹⁰. During the early phases of cystogenesis, however, IRSp53 becomes apparent in vesicle and on internalization carries. To strengthen further this observation, we also performed immune-EM CLEM analysis, which revealed the enrichment of IRSp53 along the AMIS but also in tubular/vesicular intermediates (new Fig. S7 of the revised manuscript). The partial localization into RAB7+ late endosomal vesicle, and not in Lamp1+ vesicle indicates that IRSp53 is presumably transiently associated with these structures and localizes primarily in recycling compartments. In addition, we performed a new set of colocalization experiments between EEA1 and Lamp1. In all cases, IRSp53 appears excluded from these vesicular structures (see new panel in Fig. S2D of the revised manuscript). A similar cellular pattern is also seen in the case of PODXL, which partially colocalizes with IRSp53 but not with EEA1 or Lamp1 (see Addendum Figure 1).

**Addendum Figure 1.**

A) MDCK expressing GFP-IRSp53 were seeded as single cells on Matrigel and fixed after 5h. Cysts were processed for epifluorescence to visualize GFP-IRSp53 (green) and stained with anti-EEA1 (red) and Dapi (blue). Scale bar, 10 μ m. B) MDCK expressing GFP-IRSp53 and RFP-PODXL were seeded as single cells on Matrigel and fixed after 5h. Cysts were processed for epifluorescence to visualize GFP-IRSp53 (green) and RFP-PODXL (red) and stained with anti-EEA1 (magenta) and Dapi (blue). Scale bar, 10 μ m. C) MDCK expressing GFP-IRSp53 were seeded as single cells on Matrigel and fixed after 5h. Cysts were processed for epifluorescence to visualize GFP-IRSp53 (green) and stained with anti-LAMP1 (red) and Dapi (blue). Scale bar, 10 μ m. B) MDCK expressing GFP-IRSp53 and RFP-PODXL were seeded as single cells on Matrigel and fixed after 5h. Cysts were processed for epifluorescence to visualize GFP-IRSp53 (green) and RFP-PODXL (red) and stained with anti- LAMP1 (magenta) and Dapi (blue). Scale bar, 10 μ m.

Technical issues and suggestions:

-Can the authors address the remaining IRSp53 protein that appears in their Caco2 "KO" clones? There appears to be some protein that is still expressed in these cells (Fig 2c), though it is a clearly a functionally relevant decrease in this protein.

R. The reviewer is correct in noticing the presence of residual IRSp53 in the Caco-2 KO clones. This is due to the aneuploidy of Caco-2 cells. Indeed, PCR-based analysis as described in the methods (section 1.2.3 CRISPR/CAS9 genome editing), revealed the existence of at least 3 loci. Two of these loci had missense mutations, but one displayed deletion of either 2 or 3 (depending on the clone analyzed) codons that might allow the expression of a residual amount of the mRNA. This said we estimated a stable decrease in IRSp53 levels ranging between 80 and 90% as compared to control cells. This decrease is functionally relevant as witnessed by the observations that different KO clones show a multi-lumen phenotype, which can be fully rescued by re-expression of wild type IRSp53.

We specifically commented on the residual expression in the legend to **Fig. 2 of the revised manuscript**.

-Perhaps additional experimental repeats with the IRSp53 knockdown studies would improve the error reported and the statistical relevance for the inverted polarity data in Fig 4B. The data currently represent the results of just two experiments?

R. Agree. We performed additional experiments, as requested, that confirm that the loss of IRSp53 causes a significant increase in the number of cysts that display a reversion in the polarity. We included this set of data in the **Fig. 4C of the revised manuscript**.

-The Y-axis label on the bar graph has a typo in Fig. 5E

R. We apologize for this mistake, which has been corrected in the revised version of manuscript.

- Some interaction studies of endogenous IRSp53/Rab35 proteins would be useful.

R. We agree that studies on the endogenous IRSp53/RAB35 would be of help. Unfortunately, we could not find antibodies against endogenous RAB35 suitable (and reliable) for interaction studies among endogenous proteins.

Reviewer #2

This is an interesting and data-heavy manuscript that identifies IRSp53 as a novel regulator of apical lumen formation in renal epithelia. The data demonstrating that IRSp53 is required for lumenogenesis is solid, especially since authors make an extra effort to test it in couple of different cell lines (Caco2 and MDCK) as well as in two in vivo models (mice and zebrafish). However, the part that attempts to identify the molecular machinery of IRSp53 action is much less convincing, to some extent due to various technical/experimental issues. For example, in vitro overlay assay is not a good assay to analyze protein interactions. Better approaches need to be used, ideally ITC or Biocore, but, at very least, recombinant protein pull down assay. R. To address this concern we performed a new set of experiments as suggested and confirmed that IRSp53 preferentially interacts with RAB35DN or wild type RAB35 loaded with GDP using in vitro pull-down assays with soluble and purified proteins (**Fig. S4D of the revised manuscript**)

We further tested the biochemical consequence of the loss of IRSp53 on RAB35 activity and showed (see also reply to this reviewer below) that this leads to a reduction in the level of GTP-loaded RAB35, presumably due to the perturbed/altered cellular localization of this protein (**Fig. 6C of the revised manuscript**).

Similarly, it is not clear how cells picked for EM analysis were staged and how authors know whether they already formed AMIS.

R. We performed co-staining of cells with both β -catenin and podocalyxin antibodies as shown in the revised supplementary **Fig. S8 (ex Fig. S7) of the revised manuscript**. We further tested the immunolocalization of IRSp53 by CLEM in cells co-stained with podocalyxin (see **Fig. S7 of the revised manuscript**). All these experiments indicated that we analyzed cells at the onset of AMIS formation marked by the presence of PODXL. They further confirmed that IRSp53 is enriched along the AMIS and in tubule-vesicular structures.

These issues are relatively easy to address, thus, authors should be encouraged to re-submit revised manuscript

R. We greatly appreciate the reviewer's encouragement to submit a revised version.

(if they can also address conceptual issues listed below). The manuscript also has some bigger conceptual issues. First, it is not clear how Rab35 and IRSp53 complex functions if IRSp53 binds to Rab35-GDP. All the work on Rab35 so far indicates that it functions in GTP-bound form, thus, it is unclear why it would be important to target Rab35-GDP to the AMIS.

R. We completely agree that a key functional aspect of RAB35 in cystogenesis is its activation status. However, the cellular localization of this GTPase along the plasma membrane and specifically at the AMIS is equally relevant and related to its activity. Consistently, here, we showed that:

a) IRSp53 and RAB35 are early localized at the AMIS (**Fig. S4B of the revised manuscript**), before PODXL becomes visible at this structure; b) removal of IRSp53 reduces the accumulation of RAB35 at the luminal, apical site of cysts (**Fig. 6B of the revised manuscript**). c) we have further shown with new experiments that the removal of IRSp53 also reduces the total levels of RAB35-GTP (**Fig. 6C of the revised manuscript**).

We would also like to point out that IRSp53 through its I-BAR domain preferentially binds to membranes rich in PI4,5P2 phospholipids¹¹, which have been shown to accumulate at the luminal side in cystogenesis¹². Thus, the most parsimonious interpretation of our observation in the context of previously established literature is that IRSp53 facilitates/ restricts the localization of RAB35 ready to be activated by a yet-to-be identified guanine nucleotide exchange factor at the luminal side. Notably, recent work published during the revision of this manuscript points to a specific role of the guanine nucleotide exchange factor DENND1A in controlling RAB35 activation in 3D cystogenesis¹³. At the luminal initiation site, RAB35 is, thus, expected to undergo cycles of activation and deactivation. In its GTP-bound form, it can directly associate with PODXL, functioning as tethering device for the localized delivery of PODXL-vesicles (as reported in ref.¹⁴). Consistent with this model, the removal of IRSp53, which only partially compromises the localization of RAB35, is expected to mimic a partial-loss-of-function phenotype of RAB35. Indeed, the extent of reversed polarity caused by the loss of IRSp53 is less robust as compared to the one resulting from the complete loss of RAB35. Conversely, the partial loss of RAB35 has been shown to result in the formation of multiple lumens as reported^{14,15} (see also below), similar to what we observed in the case of IRSp53 removal.

It is also relevant to point out that the multi-luminal phenotype caused by IRSp53 loss is likely the result of additional functions that this protein exerts in lumenogenesis. Indeed, our analysis of the structural organization of the opposing plasma membrane of two daughter cells during cystogenesis revealed that the loss of IRSp53 results in the formation of aberrant cytoplasmic bridges that interconnect the cytoplasm of two-adjacent cells and interrupt the continuity of the plasma membrane at the nascent lumen. The loss of IRSp53 also reduces intercellular space, and occasionally induces the formation of ectopic lumens. These alterations are likely to generate distinct mini-lumens and at PM targeting sites that combined with altered polarized trafficking of podocalyxin carriers leads to the accumulation of podocalyxin in multiple foci, which will eventually evolve into multiple lumens.

The even bigger conceptual issue is the need of Rab35 itself for lumen formation. While a couple years ago Echard's lab used rab35 KD to suggest that it is involved in MDCK lumenogenesis, the recent study from Fukuda lab (that used Rab35 KO line) did not see any effect of Rab35 loss on apical lumen formation (the paper is not cited in the manuscript). Authors should generate Rab35 KO line (since Rab35 KD s may have off-target effects) and recapitulate the key experiments. Preferably, that should be followed by GF-Rab35 rescue. I realize that it is a lot of work, but since Rab35 is at the center of this manuscript, the issue whether rab35 is needed for lumen formation needs to be addressed.

R. We thank the reviewer for pointing this out. Firstly, we would like to underline that we were and are aware of Fukuda work¹⁵ that appeared about at the same time as the study by the Echard group¹⁴ and of his more recent publication published during the revision of our work¹³. Indeed, Fukuda work had been cited in our previous version of the manuscript.

In all these works, they observe a drastic, but transient inversion of polarity following removal of RAB35 either by complete KO or KD approaches. Notably, however, after acute removal of RAB35 nearly 50 % of the cysts display inverted polarity while about 30% display defects in the distribution of PODXL¹⁵, which localizes in multiple spots and give rise to multiple lumens at late stage of cysts developments (see Figure 5 from the work by Fukuda laboratory¹⁵) Below is shown Figure 6E of the same paper, which reports that the removal of RAB35 results in polarity inversion but also the formation of multiple and aberrant lumens.

These alterations are very similar to the ones we detected in the case of IRSp53 loss.

Figure 6E legend by Mrozowska et al. JCB2016.

(E) Rab35 KO clone 18 or 20 cells were plated on Matrigel-coated glass slides and fixed with PFA at the times indicated. The cells were then stained with anti-PCX antibody, and the cysts with peripheral PCX were counted (left graph). Bars on the graph represent the number of cysts (%) with peripheral PCX scored once for each clone ($n > 150$). Note that the degree of polarity reversal, which occurs with increased culturing time, is comparable for both clones. The microscopic images on the right side of the graph show representative phenotypes of the clone 20. The white arrows and yellow arrowheads point apical PCX and peripheral PCX, respectively.

Echard's group¹⁴, in addition to inversion of polarity, also reported the formation of multiple lumens, particularly when RAB35 was incompletely eliminated

Figure 3D from Klinkert et al Nat. Comm 2016¹⁴

Thus, a partial loss-of-RAB35 function impacts on polarity and lumenogenesis

We repeated these latter experiments using two different RNAi oligos against RAB35 and indeed detected, in both cases, inversion of polarity as well as multi-lumen formation. Thus, RAB35 appears necessary for the correct localization of PODXL, polarity establishment and the formation of a single lumen consistently with what we experimentally observed.

Notably, the phenotypes caused by the removal of IRSp53 are multiple lumens and polarity inversion. Admittedly, the penetrance of polarity inversion is not as robust as in the case of RAB35 loss, albeit in a new set of experiments we carried out to reinforce the original observation we detected about 10% of IRSp53 deficient cysts with a complete inversion of polarity (See Fig. 4C of the revised manuscript). This is, however, expected since IRSp53 only partially affects RAB35 localization and activity and further impacts on other independent pathways as revealed by the structures function analysis we performed, the

identification of the critical role of Eps8 in lumenogenesis, and the CLEM analysis that indicates a specific role of IRSp53 in shaping the plasma membrane during the early phases of cyst formation.

We also reconstituted the expression of RAB35 in siRNA silenced MDCK cells as suggested. The loss of RAB35 results in reversion of polarity and in the formation of multiple lumen, which can be rescued by the re-expression of RAB35. We included this finding below as **Addendum Figure 2** for reviewer's perusal. We would not include them in the manuscript as they are mainly confirmatory of previously reported results.

This said we reworded and expanded the text in the revised version of the manuscript (pg. 10) and also inserted the new reference by Fukuda Lab.

Addendum Figure 2.

Top: MDCK control or expressing human RFP-RAB35 cells, transfected with scramble oligo (Ctr_scr or RFP-RAB35_scr) or with two different siRNA against canine RAB35 (Ctr_KD1, Ctr_KD2 or RFP-RAB35_KD1, RFP-RAB35_KD2), were seeded as single cells on a Matrigel layer and left to grow for 30 h. The cysts were fixed and stained with FITC-Phalloidin (green), anti-PODXL (magenta) and DAPI (blue) (Ctr_Scr, Ctr_KD1, Ctr_KD2) or processed for epifluorescence to visualize RFP-RAB35 and stained with FITC-Phalloidin (green), anti-PODXL (magenta) and DAPI (blue) (RFP-RAB35_Scr, RFP-RAB35_KD1, RFP-RAB35_KD2). Asterisks, PODXL multi-foci in Ctr_KD1 and Ctr_KD2 cysts. Arrowheads, PODXL staining at the basal membrane in Ctr_KD1 and Ctr_KD2 cysts. Scale bar, 10 μ m. *Bottom left:* Quantification of multi-foci cysts (left) and inverted polarity cysts (right). Data are means \pm SD. At least 20 four-cell stage cysts were analysed for each experimental condition. *Bottom right:* Immunoblotting of expression levels of RAB35 to assess down-regulation of the endogenous protein and expression of the recombinant RFP-RAB35.

Major Points:

1) Figure 2C. Why Caco2 IRSp53 cells still have IRSp53 protein?

R. The reviewer is correct in noticing the presence of residual IRSp53 in the KO clones. This is due to the aneuploidy of Caco-2 cells. Indeed, PCR-based analysis as described in the methods (section 1.2.3 CRISPR/CAS9 genome editing), revealed the existence of at least 3 loci. Two of these loci had missense mutations, but one displayed deletion of either 2 (or 3 depending on the clone analyzed) codons that might allow the expression of a residual amount of the protein. This said we estimated a stable decrease in IRSp53 levels ranging between 80 and 90%. This decrease is functionally relevant as witnessed by the observations that different KO clones show the multi-lumen phenotype, which can be fully rescued by re-expression of WT IRSp53.

We specifically commented on the residual expression in the legend to **Fig. 2 of the revised manuscript**.

2) Figure 4. Recent work (JCB, 2019) from Fukuda lab has demonstrated that Rab35 KO in MDCK cells does not lead to defects in lumenogenesis. That does not seem to fit with proposed model that Rab35 targets IRSp53. Additionally, authors demonstrate the effect of Rab35 KD on lumenogenesis. How do authors explain that? Can they rescue Rab35 defects with GFP-Rab35? Additionally, Fukuda paper needs to be cited.

R. As pointed out above the work by Fukuda and colleagues had already been included and cited. We have specifically discussed this issue above (see reply to this reviewer's general comment) and performed the RAB35 rescue experiments as requested (see **Addendum Figure 2** and the reply to the general comment by this reviewer).

3) Figure 5B-C. In vitro overlay assay is not a good method to analyze protein binding. More conventional and more informative recombinant protein pull-down assay (or even better quantitative binding assays, such as Biocore or ITC) should be used. As shown in supplemental figure 4D, data is poor quality and not-interpretable. Additionally, to rigorously test nucleotide dependency of Rab and effector protein binding, one needs to do GTPγS or GDP loading in purified recombinant wild-type Rab. That is especially important for this manuscript, since presumptive IRSp53 binding to Rab35-GDP is at the core of the finding.

R. We partially agree with this since overlay assays have been extensively used to provide evidence of direct interaction among purified proteins. Nevertheless, to provide further experimental evidence of the mechanistic role exerted by IRSp53 in conjunction with RAB35 and PODXL we performed the following set of experiments, as requested:

a. As suggested by this reviewer, we performed pull down assays using soluble, purified IRSp53 and RAB35 wild type loaded with either GDP and GTPγS as well as with RAB35S22N (the dominant negative form) and RAB35Q67L (the dominant active-GTP bound form). In all cases, we confirm that IRSp53 binds directly and preferentially to RAB35 in its inactive GDP-bound form (**Fig. S4D of the revised manuscript**).

b. We also determined the amount of active RAB35 in control and IRSp53 KO cells using an RBD (RAB35 Binding Domain)-assay and showed that the loss of IRSp53 results in a sizeable reduction of the levels of RAB35-GTP (**Fig. 6C of the revised manuscript**). This finding correlates with the reduced apical and luminal localization of RAB35 detected following the removal of IRSp53 during cysts formation (**Fig. 6B of the revised manuscript**) and suggests that IRSp53 likely acts by stabilizing the localization of RAB35 and, in doing so, it facilitates the activation of RAB35.

4) Figure 5E. None of the mutants, except PDZ, (not only I-BAR) rescued IRSp53 KO. Thus, it is hard to conclude that Rab35 binding is required. Besides, K to E is pretty dramatic mutation that likely affects overall protein conformation. Finally, it is hard to believe that this binding is required since Rab35-KO does not affect lumenogenesis (see Fukuda paper).

R. The K to E mutations (I-BAR*: K142E, K143E, K146E, K147E¹¹) have been previously characterized by circular dichroism and shown not to alter the overall structure of the I-BAR domain³⁴ but to impair its binding to negatively charged phospholipids as well as, in this work, to RAB35. Our structure/ functional analysis further indicates that IRSp53 requires its functional I-BAR domain for its role in lumen formation. However, we acknowledged explicitly in the manuscript that:

"molecularly, both its (IRSp53) binding to negatively curved, phosphatidylinositol 4,5-bis phosphate (Ptdins(4,5)P₂)-rich membranes (which is compromised by the mutations inserted¹¹), and/ or its interactions with RAB35 likely account for these findings".

Notably, as discussed above, the loss of IRSp53 is expected to impair only partially RAB35 functions, a condition that others^{14, 15} and we (in this manuscript) have demonstrated to result in the formation of cysts with multiple lumens¹⁴⁻¹⁵.

We must also point out that IRSp53 appears to act in this process as a platform for the assembly of diverse protein complexes that include EPS8 and CDC42. Indeed, mutants of IRSp53 that impair the interaction with these proteins are no longer able of rescuing the phenotype due to the silencing of IRSp53. These results are consistent with a complex biochemical role exerted by IRSp53, which also controls the structures and shape of the opposing membrane of two daughter cells at the onset of cystogenesis. Dissecting the specific contributions of these interactions and functions of IRSp53 in epithelial polarity and lumenogenesis will certainly require further work that is beyond the scope of the current manuscript.

5) Figure 6B. The data regarding the role of IRSp53 in targeting Rab35 is not totally convincing. Based on images, even in IRSp53 KO cells RFP-Rab35 does seem to be present at the AMIS (very moderate effect based on quantification) as well as everywhere else in lateral cell membrane.

R. The reduction of RAB35 at the apical side in IRSp53 KO cells is expected and in line with the concept that IRSp53 loss mimics a partial-loss-of-function of this GTPase. Indeed, RAB35 is known to undergo lipid modifications that target it to the plasma membrane. Thus, a reduction of this protein but not a total absence is what one could have reasonably expected to occur in IRSp53 deficient cells. This reduction in localization is also accompanied by a reduction in RAB35-GTP level and activity, as we have shown in the new set of data (**Figure 6B-C of the revised manuscript**). These defects likely contribute to the lumenogenesis phenotypes observed in IRSp53 KO cysts.

6) Supplemental Figure 6. Why is such a huge difference in multi-lumenation in control and Eps8-KO cells between different experiments (compare bar graph on right and left in C)? Once again (see minor point 1), it is hard to explain that as a result of experimental variability, since standard errors are pretty small and presumably data was derived from several biological experimental replicates.

R. The two graphs measured two different phenotypes. The graph on the left (**Fig. S6A**) is the quantification of the number of cysts with inverted polarity, the one on the right (**Fig. 6SB**) is the quantification of the number of cysts that display multiple lumens.

7) Since IRSp53 regulate ezrin (a known apical protein), the authors should test ezrin localization at the apical membrane in IRSp53 KO cells.

R. We performed this experiment and showed that the loss of IRSp53 does not significantly affect the luminal, apical localization of activated phospho-ERM, suggesting that additional determinants are critical for the correct distribution of Ezrin during cystogenesis. We included this figure below, as **Addendum Figure 3**, but not in the revised paper due to space constraints.

**Addendum Figure 3.**

Caco-2 control (Ctr) or IRSp53-KO (clone #3), were embedded as single cells into Matrigel/Collagen matrix and left growing to form cysts. Cysts were fixed and stained with anti-p-ERM (Green), rhodamine phalloidin to detect F-actin (Red), and Dapi (blue). Arrowheads indicate **pERM luminal localization**, asterisks indicate multiple lumens. Scale bar, 20 μ m.

8) Figure 7. How do authors know that those two cells formed AMIS since they used β -catenin as marker. Just because they are two cells, that does not mean that they already formed AMIS. AMIS marker (such as podocalyxin) should be used.

R. Agree. To address this issue, we co-stained cells with both β -catenin and Podocalyxin antibodies (see the revised supplementary Fig. S8). We further tested the immunolocalization of IRSp53 by CLEM in cells co-stained with Podocalyxin (see Fig. S7 of the revised manuscript). All these experiments indicated that we analyzed cells at the onset of AMIS formation marked by the presence of PODXL. They further confirmed that IRSp53 is enriched along the AMIS and in tubule-vesicular-like structures. Using Podocalyxin as a marker of AMIS, we also noticed that the kinetic of AMIS formation is remarkably reproducible with the onset starting between 16-18 hours after the initial seeding. The exact same timing, which represents a relative robust proxy of the stage of AMIS formation and cystogenesis, has been used in all CLEM experiments.

Minor Point:

1) Figure 2B. Why multi-lumen formation in control cells in KD and KO experiments is so different? Since standard deviation is pretty small that is not likely due to variability between experiments.

R. We agree that the relative number of control cysts with multi-lumen in this set of experiment was particularly elevated. This was the first set of experiments that were performed on control cells (subjected to the same CRISPR treatment as IRSp53 KO) that were unfortunately kept in culture for a prolonged amount of time. We therefore repeated this analysis on freshly thawed cells. Under these conditions, the % of control cysts displaying a multi-luminal phenotype is about 15-20% of cases, which are the levels normally detected in wild type MDCK. We included this set of measurements in the revised Figure 2B.

Reviewer #3

This paper describes the possible role of IRSp53 in epithelial morphogenesis. The paper first characterized the morphology of kidney sections in knockout or knockdown animals, and found the abnormality in the tubular structures of the kidney. The localization of IRSp53 was analyzed in detail using the model cell system that forms cyst and the kidney sections. Then the authors proposed the possible role of IRSp53 in the apical-basal polarity of the tubular structure formation in cellular and kidney development. However, the involvement of IRSp53 in cyst formation in MDCK cells were already shown in JCB 192 (3)525-540 2011. Though this paper showed that the defect in cyst formation upon IRSp53 depletion can be rescued by IRSp53 expression. The possible molecular mechanism is also provided. The similar phenotype was observed by the IRSp53-interacting Eps8 knockout. Furthermore, the interaction of IRSp53 with Rab35 is proposed. The overall data are results of enormous effort to characterize the role of IRSp53. I would like to point several issues to be addressed for the clarity of the IRSp53 function, R. We thank the reviewer to appreciate the efforts that we put in characterizing the role of IRSp53 in epithelial polarity.

Major points.

The function of kidney of IRSp53 KO animals might be analyzed in relation to clinical aspects, such as the blood concentrations of the chemicals that is known to be filtrated through kidney. Such things can be easily addressed by the biochemical analysis of plasma.

R. We thank the reviewer for this comment. We performed a set of physiological analysis both on the urine and the blood of control and IRSp53 mice. The analysis of water consumption and urine production, indeed, revealed that KO mice drink significantly less water and produce less urine as compared to WT control mice. Despite this difference, the analysis of a number of urine biomarkers and the renal profile analysis of the plasma that was carried out by Comparative Clinical Pathology Services, LLC, USA) revealed no significant differences between the two genotypes (see **Addendum Figure 4 to this reviewer**). Clearly more work would need to be done in order to investigate whether the volumetric difference in urine productions and water consumption can be attributed to the set of alterations in the morphology and polarity of kidney structures. It is noticeable, nevertheless, that these structural alterations of the kidneys are evident and quantitatively significant but qualitatively not dramatic possibly due to molecular mechanisms (e.g. the expression of other members of the I-BAR family, as reported in ref.¹⁶) that compensate for the loss of IRSp53. We discussed this finding in the revised version of the manuscript (**pg. 19 of the revised manuscript**). We are reluctant, however, to include them as a figure in a manuscript that, as reviewer #2 indicated, is already extremely data heavy.

Addendum Figure 4

WT n. 7 (4M; 3F)
KO n. 7 (4M; 3F)

Addendum Figure 4.

A) Water intake. Water consumption in *IRSp-53* WT (n = 7; 4M, 3F) and KO (n = 7; 4M, 3F) mice was determined by the difference in weight of the water containers measured every 24 hours during two weeks and expressed as the daily average water intake \pm SD. *** p < 0.005. B) Urine production. Urine was collected from *IRSp-53* WT (n = 7; 4M, 3F) and KO (n = 7; 4M, 3F) mice two times/day (morning and afternoon) for two weeks and the urine output expressed as the daily average volume of urine collected \pm SD. *** p < 0.005. C) Frozen urine samples collected by *IRSp-53* WT and KO mice were pulled and analysed by the Comparative Clinical Pathology Services (LLC, USA). No significant differences were highlighted by the analysis of the kidney biochemical profile markers analysed. NAG: N-acetyl- β -D-glucosaminidase. D) Frozen plasma samples collected by *IRSp-53* WT and KO mice were pulled and analysed by the Comparative Clinical Pathology Services (LLC, USA). No significant differences were highlighted by the analysis of the kidney biochemical profile markers analysed. BUN: blood urea nitrogen.

The proposal is that *IRSp53* is involved in the tubulogenesis (apical-basal polarity), which is thought to occur during tubulogenesis of kidney or the cyst formation in cellular culture in 3D matrix. The tissue sectioning of kidney during the tubulogenesis (ideally consistent with the AMIS in kidney) might be stained with *IRSp53* antibody for the consistency between cellular experiments and animal models.

R. We performed *IRSp53* localization during embryogenesis in Zebrafish samples (**Fig. 1D**), which represent a nearly ideal model to document the localization of a protein during early stages of kidney morphogenesis. Importantly, *BAIAP2A*, the zebrafish orthologue of *IRSp53*, is expressed and localized to the apical cells of the pronephric duct during early larval stages of development.

The binding of *IRSp53* to Rab35 is to the dominant negative form, which was suggested to promote the re-localization of podocalyxin (PODXL) to the apical surface of the cyst. The binding of active form of Rab35 to PODXL was reported (Nature commn, 2016 DOI: 10.1038/ncomms11166). The comparison of the PODXL localization upon *IRSp53* and Rab35 knockdown in Figure 4B suggested that these two proteins are both involved in the localization of PODXL. What are the molecular mechanisms behind in terms of Rab35 activity? How the inactive form of Rab35 regulate the *IRSp53* function? The expression of various *IRSp53* mutant in cells expressing active or dominant negative form of Rab35 will be helpful.

R. To address the first question, we determined the amount of active RAB35 in control and *IRSp53* KO cells using an RBD (RAB binding Domain)-assay, and showed that the loss of *IRSp53* results in a reduction in the activation of this GTPases (**Fig 6C of the revised manuscript**). This finding correlates with the reduced apical and luminal localization of RAB35 detected following the removal of *IRSp53* (**Fig. 6B of the revised manuscript**) during cysts formation (as noticed by the reviewer) and suggests that *IRSp53* likely acts by stabilizing the localization of RAB35 and, in doing so, it facilitates its activation.

It is also important to underline that we have no evidence that the inactive form of RAB35 regulates *IRSp53* function. Indeed, the removal of RAB35 has no effect on *IRSp53* cellular distribution during cystogenesis (**Fig. S4C of the revised manuscript**). We also would like to refer this reviewer to the reply to Reviewer #2 for additional details as to the possible mechanisms of action of the newly identified *IRSp53/RAB35* axis.

The binding of *IRSp53* to Rab35 occurs at the basic amino-acid residues at the tip of the I-BAR domain structure, which is also involved in the membrane binding. On the other hand, the absence of Rab35 did not affect the *IRSp53* localization (Figure S4). As the authors discussed, these basic amino-acid residues will contribute to the binding to PIP2, and therefore, it is not easy to conclude the role of possible Rab35 mediated regulation/recruitment of the *IRSp53* by this mutation in the I-BAR domain (I-BAR*). At least it is better to discuss these points and ideally addressed experimentally.

R. Agree. We explicitly discuss (**pg.13 of the revised manuscript**) that this set of mutations will also affect the ability of *IRSp53* to bind to PI4,5P2-rich membranes and further stated that the inability of this mutant to restore the defect due to the loss of *IRSp53* is likely the results of the concomitant impairment in the binding to RAB35 and to PI4.5P2-rich plasma membrane. We also agree that it would be ideal to decouple experimentally the binding to PI4,5P2 from that to RAB35, but so far attempts in this direction have failed, suggesting that the same surface might be implicated in these interactions.

The established function of *IRSp53* is to induce filipodia like membrane protrusions. The *IRSp53* knockdown/knockout resulted in the increased filipodia like structures in the CLEM analysis of *IRSp53* knockdown/knockout cells in 3D matrix.

R. We would like to point out that the removal of *IRSp53* does not increase the number of filipodia. Actually, as reported in **Figure 7, the newly added Supplementary Fig. 7 and in Supplementary Fig. 8**, the loss of *IRSp53* results in the formation of cytoplasmic bridges and frequently in the close juxtaposition of the opposing membrane of two daughter cells (**please see Fig. 7A bottom panel**). The opposing membrane of two daughter cells do display a convoluted, apparently actin-free structure, rather than protrusive filipodia.

On the other hand, it is known that the apical surface of the tubules or cysts could contain a lot of filipodia-like brush borders. The enrichment of *IRSp53* at the apical surface might just reflect the abundance of filipodia-like structures. The CLEM of apical surface with anti-*IRSp53* antibody would be of interest.

R. We performed the experiments suggested. The results are included in **Supplementary Figure S7 of the revised manuscript** and revealed that *IRSp53* localizes at the newly forming AMIS under condition where a clear brush border is not yet developed. Having said this, we previously observed an enrichment of *IRSp53* in brush borders on the apical microvilli of intestinal cells similarly to the localization of *Eps8*¹⁷, a nearly stoichiometric binding partner of *IRSp53*.

The internalization of PODXL can be dependent on *IRSp53*. Then, *IRSp53* will co-localize with PODXL during the internalization. Ideally it will be better to do some live-imaging to show that or stain the cells with also *IRSp53* antibody in Figure S3B.

R. We monitored the localization of *IRSp53* during the internalization of PODXL using a calcium switch set up in MDCK monolayers. As shown in **Figure S3B of the revised manuscript**, we can detect *IRSp53* on vesicular apical compartments together with PODXL. This finding together with the functional internalization assays of **Fig. S3C-E of the revised manuscript**

indicates that IRSp53 also partly regulates PODXL internalization, and further points to functions of IRSp53 independent on the role exerted on RAB35.

The illustration in Figure S11 indicated that the cytosol is connected upon knockdown of IRSp53. Is there any evidence for that, like dye transfer?

R. The suggested experiments would be an ideal demonstration of the existence of cytoplasmic bridges. However, introducing a cytoplasmic dye only in one of the daughter cells after the first cell division is a formidable task that we have not yet been able to achieve. This is also why we opted to perform 3D tomography by EM of these structures (Fig. 7B). To our knowledge, this approach (which has never been done before during cyst formation) represents the most direct way to demonstrate the presence of interconnected cytoplasmic bridges.

Minor points.

The binding of IRSp53 to VASP was reported (Oikawa et al, 2013, doi: 10.1371/journal.pone.0060528)

The binding of I-BAR to small GTPase is reported in some papers, and needs to be discussed (J Biol Chem. 2019 Apr 19;294(16):6494-6505., J Biol Chem. 2019 Apr 19;294(16):6494-6505. , J Cell Sci. 2007 May 1;120(Pt 9):1663-72., Nature. 2000 Dec 7;408(6813):732-5.)

R. We included the suggested references documenting the interaction of IRSp53 with either RhoGTPases (Cdc42 and RAC1) or VASP in the introduction of the revised manuscript. We discussed instead the interaction of MIM and IRTKs with RAB proteins reported in¹⁸ in the discussion as follows:

"Finally, it is of note that also other I-BAR domain family members, IRTKS and MIM, have been shown to interact with RAB GTPases and to be involved in endocytosis of CXCR4 receptor¹⁸. Thus, the involvement in membrane trafficking appears to be a more general emerging function of the I-BAR family proteins."

The mutant construct of IRSp53 did not have the details. Does the * mean the point mutation in full-length? The western in Figure 5E appears to suggest these are full-length mutants. Then it is better to describe as amino-acid mutations.

R. Details of the construct were included in the Methods section. This said, we specified the nature of this mutation in the legend to Figure 5

The mass spec data might not support the conclusion by the authors, but should be presented in the text for help to readers. During the review, I could not see them.

R. We mentioned this finding in the discussion and uploaded the full repertoire of interacting proteins on the PeptideAtlas repository (<http://www.peptideatlas.org/PASS/PASS01464>) to provide evidence that no guanine nucleotide exchange factor for RAB35 were identified as IRSp53 interactors under the conditions tested as specified in the supplementary information. We could include a summary list (e.g., Addendum Figure 5-below) of the most relevant hits; however, we feel that this list would need to be validated through a set of additional experiments before being published. For example, we did validate the interaction with ZO-1. However, to understand how IRSp53 acts through this protein would require a much larger set of experiments that, we feel, is beyond the scope of the present work.

Addendum Figure 5

Gene name	Protein name	Notes and functions
BAIAP2	Brain-specific angiogenesis inhibitor 1-associated protein 2	BirA* fusion
CGN	Cingulin	Actin binding, involved in thigh junction assembly
TJP1	Thight junction protein ZO-1	Involved in thight junction assembly
DLG1	Disks large homolog 1	Scaffold protein, involved in junction assembly, signal transduction, cell proliferation
INPPL1	Phosphatidylinositol 3,4,5-trisphosphate 5-phosphatase 2	Negatively regulates PI3K pathway to produce PIP2
LIN7C; LIN7A (MALS)	Protein lin-7 homolog C; protein lin-7 homolog A	Involved in cell polarity
INADL (PATJ)	InaD-like protein	Scaffold protein, involved in cell polarity and thight junction assembly
MPP5	MAGUK p55 subfamily member 5	Involved in cell polarity and thight junction assembly
NCK1	Cytoplasmic protein NCK1	Scaffold protein, cytoskeletal adaptor
SHANK2; CORTBP1	SH3 and multiple ankyrin repeat domains protein 2	Scaffold activity in postsynaptic density
ERBB2IP	Protein LAP2	Adapter for ERBB2 receptor
MPDZ	Multiple PDZ domains protein	NMDAR signaling
APC	Adenomatous polyposis coli protein	Tumor suppressor involved in Wnt signaling
MUC13	Mucin-13	Cell signaling
EGFR	Epidermal growth factor receptor	Receptor tyrosin kinase
SEC24B	Transport protein Sec24B	Vesicle trafficking from ER to Golgi
SLC3A2	4F2 cell-surface antigen heavy chain	Amino-acids transport
SEC16A	Transport protein Sec16A	Vesicle trafficking from ER to Golgi
RAPH1 (lamellipodin)	Ras-associated and pleckstrin homology domains-containing protein 1	Mediator of membrane signals, lamellipodia dynamic
WASF2	Wiskott-Aldrich syndrome protein family member 2	Part of WAVE complex, actin cytoskeleton dynamic
ABII	Abl interactor 1	EGFR signaling, actin cytoskeleton dynamic
EPS8	Epidermal growth factor receptor kinase substrate 8	EGFR signaling, actin cytoskeleton dynamic
BAIAP2L1	Brain-specific angiogenesis inhibitor 1-associated protein 2-like 1	Adapter protein, membrane and actin dynamic
TACC2	Transforming acidic coiled-coiled-containign protein 2	Centrosomal microtubules organization
TXNL1	Thioredoxin-like protein 1	Oxidoreductase activity
TP53BP2	Apoptosis-stimulating of p53 protein 2	Apoptosis stimulation
UBAP2L	Ubiquitin-associated protein 2-like	Hematopoietic stem cells activity regulator

■ Cell-cell junction/polarity ■ Adaptors ■ Signaling ■ Trafficking ■ Actin architecture/dynamic ■ Other

Addendum Figure 5.

List of the top proteins that interact with IRSp53 identified by mass spectrometry using BioID approach combined with in-cell stable isotope labelling (SILAC) The list is obtained based on the normalized heavy/light ratio of the peptides recovered). The list of the proteins is organized according to their function. MS data are analysed as described in the Material and Methods section. IRSp53 known binding partners are highlighted in yellow. A complete list of the results as well of the parameters used is

included in the Proteomic data available at the PeptideAtlas repository (<http://www.peptideatlas.org/PASS/PASS01464>).

In Figure 2B, the graph had no explanations on the difference in the two graphs.

R. The two graphs report the % of cysts that display a multi-lumen phenotype either in cells in which IRSp53 had been knocked out by CRISPR technology or were transiently silenced by RNAi. We more explicitly indicated this by adding additional labelling to the revised figure (**See revised Figure 2**). In addition, we would like to point out that the % of cysts with multiple lumens in control of the CRISPR experiments was particularly high. As specified in the reply to "minor point 1" of reviewer#2, the analysis of cysts with multiple lumen in KO cells was one of the first experiment that were performed. We realized that the control KO cells had been kept in culture for an excessive amount of passages. We, therefore, repeated this analysis on freshly thawed cells. Under these conditions, the % of control cysts displaying a multi-luminal phenotype was about 15-20%, which are the levels normally detected in MDCK. We included this set of measurements **in the revised Fig. 2B**.

In Figure 2C, the GFP-IRSP53 bands were cropped and not shown, please show these.

R. The GFP and GFP-IRSp53 are shown **in Fig. S5A**, as also indicated in the legend of Figure 2C.

In Figure 3 and 4, the staining with ZO-1 and beta-catenin will help to see the relation between the cell-cell junction formation and the tubulogenesis.

R. Agree. We performed β -catenin staining in **Fig. 4 and Fig. S4A of the revised manuscript** to highlight the relation between cell-cell junction and lumenogenesis. We also performed ZO-1 staining in Caco-2 (**Fig. 6A of the revised manuscript**).

In Figure 3C, the co-localization of endogenous proteins will be better to be analyzed. The co-localization at AMIS (or 2 cell stage as Figure 3B) will also need to be analyzed. In Figure 5E, the localization of Rab35 and Eps8 should also be examined.

R. We agree in principle with the reviewer. In this particular case, we did make sure that GFP-IRSp53 and the endogenous protein had the same cellular localization. The GFP-IRSp53 signal is, however, significantly stronger and easier to follow, thus analysis of its expression is more reliable. It must also be pointed out that GFP-IRSp53 wild type fully restored the cystogenesis defects of IRSP53 KO cells. We, instead, used RFP-Rab7 and RFP-RAB35 because the antibodies recognizing the endogenous proteins works poorly under the condition of the cystogenesis assay, making the interpretation of the experimental outcome less certain.

The localization of EPS8 is not affected by the removal of IRSp53 (**Fig. S6D of the revised manuscript**) and EPS8 invariably decorates the apical luminal side as shown **in Fig. S5B-D**.

We also agree that it would be informative testing RAB35 localization in these experiments. However, given the lack of availability of a reliable antibody against endogenous RAB35, performing this experiment would have entailed generating 6 different and independent lines expressing RFP-RAB35 to endogenous levels: an effort we are currently undertaking but that seems incompatible with a timely publication of this work.

In methods, the JS buffer is unclear.

R. JS is a lab jargon that originally indicated a buffer used by J. Schlessinger for performing cellular biochemistry experiments. We removed this nomenclature in the revised text.

Experimental detail for Figure S3B (striping assay) was not clear for me.

R. we described the stripping assay more in details in the method section. This assay is based on the ability of acidic solution to remove (strip) proteins (in this case antibody against PODXL) from the cell surface. To clarify this point we also included a more detailed description of the assay in the legend to **Fig. S3A of the revised manuscript**.

Reviewer #4

This manuscript indicates IRSp53 as one of the key regulators for the establishment of apical polarity. The authors use a variety of cell lines and animal models to substantiate their findings which together provide a compelling case for their conclusions. I was asked to mainly comment on the technical aspects of the Correlative Microscopy in the manuscript which I will focus on below.

Abstract 2nd line: plasma membrane / actin

Page 4, line 10 from bottom: the these (remove one)

Page 4 line 7 from bottom: cortex to (twice)

Page 4 line 6 from bottom (the the (twice))

R. We thank the reviewer for pointing out these oversights. Corrections have been made.

Control and WT are both used, sometimes unclear why / what the difference is

R. We eliminated the term WT and opted for the use of Control (or its abbreviation Ctr) in each of the experiments with the exception of the experiments on animal models in which WT is referred to the genetic background.

Figure S2 inconsistent A has GFP bottom, B and C have GFP first

R. Agree. We moved the GFP on the top **in Fig. S2A**, as suggested.

Page 8, line 2: what is the evidence these are a; vesicles, b; internalized as opposed to newly synthesized?

R. The process of internalization of PODXL from a peripheral plasma during the early step of cystogenesis has been previously documented by many different laboratories^{14, 19-23} and used widely to provide evidences of the role of endosomal trafficking of this polarity determinants in lumenogenesis. We used identical conditions and cellular systems with respect to the one previously published. We also more directly measured PODXL internalization into vesicular apical compartments in monolayers (a system used to analyze PODXL trafficking) by incubating cells with an antibody that recognizes the extracellular portion of PODXL, thereby allowing the monitoring of its trafficking upon Calcium removal (**Fig. S3A of the revised manuscript**). This

evidence and considerations indicate that during cystogenesis in general and specifically in MDCK cyst formation, PODXL becomes enriched in vesicular recycling compartments.

This said we rephrased the text in amore cautionary way, as follows:

... In the first 8 h to 12 h after plating of the control cells, PODXL localized to the peripheral surface of the cells, **before accumulating in vesicular-like, recycling structures as previously shown^{14, 19-23}**....

Page 14/ Fig 7: technical aspects of CLEM data convincing and well interpreted. I do have a few observations that need clarification:

- There appear to be decreased intercellular spaces in the KO cells (compare insets Ctr and KO. Comment?)

R. We also noticed that frequently the intercellular space is reduced, consistent with a role of IRSp53 in shaping and organizing the plasma membrane. We added a comment on this aspect on **pg. 15 of the revised manuscript**.

- Fig. S7: I would have expected a yellow (red and green) border in the control between the 2 cells if there was significant PODXL present. This can not be verified but from the EM overview the labelling looks higher in the plasma membrane outside the border than between the cells. In addition, what is the structure of the red spot inside the cell.

R. In the **revised Fig. S8 (previously named Fig. S7)**, we showed the individual fluorescent channels in order to highlight the presence of PODXL that begins accumulating at the opposing plasma membrane of two daughter cells. We have also added a **new Fig. S7** showing the distribution of PODXL and IRSp53 by CLEM analysis. The red spot in this figure represents PODXL-positive structures likely indicative of an extra lumen being formed.

There is a very large spread in the silver enhancement of the gold particles in the KO cell (much more than the control cell) closer inspection shows a lot of labelling on other structures.

R. We would like to point out that this is not silver enhancement but gold enhancement and thus, due to feature of this methods, small dots represent the background.

Page 20, line 4: electron microscopy

R. We corrected the typo.

Supplemental Material:

1.1 antibodies: no secondary antibodies are listed

R. We added the list of secondary antibodies used in chapter 1.1 Antibodies, plasmid and reagents on the Materials and Methods section of the revised Supplementary Information

1.3.1, 1.3.2 and 1.3.3: Don't understand why there is a difference in the IF protocols

R. The various protocols differ for the % of PFA, time of fixation and in some cases addition of Glycine for the cysts fixation. These are used and optimized based on the different nature of the samples (e.g. 3D cell cysts or cells seeded on Matrigel plugs in Matrigel containing media requires different procedure with respect to cells seeded on 2D substrate in DMEM). These different protocols were developed from established, published procedures from different laboratories and shown to be the best for each of the conditions tested.

1.3.4 line 2: monolayers were then washed

R. Corrected.

Line5 antibody that recognizes

R. Corrected.

Page 12, line 1: blocked and permeabilized

R. Corrected.

Line 6: fixed with of

R. Corrected.

Line 9: please clarify 1:1 mixture, what are the end concentrations and in what buffer?

R. 1:1 refers to volume:volume for a final concentration of 1.0 % osmium tetra oxide and 1.5 % potassium ferrocyanide. We specified this information on **pg. 12 of the revised manuscript**.

Sodium cacodylate / Na cacodylate / cacodylate buffer are used interchangeable, please use 1 consistently

R. We consistently used Sodium cacodylate buffer in the revised manuscript.

Immunolabeling with gold particles

Line 5: what is the blocking solution?

R. Blocking solution: 0,5 % BSA, 0,1 % saponin, 0,27 g NH₄Cl in 0.2 M HEPES (pH 7.2). We inserted this information into the manuscript.

Page 12, Bottom line: who is the manufacturer?

R. GoldEnhanceTM-EM is from Nanoprobes, Yaphank NY, USA. We added this information to the revised manuscript.

Page 13, from line 2 same as protocol above, redundant

R. We removed this redundant part

Sectioning:

Line 2: Can you give an estimate what "minimal thickness" is?

R. The average thickness of the Matrigel layers is ~ 100 micrometers. We added this information to the revised manuscript.

Line 3: clarify "phase" : cell stage?

R. We clarified as follows:

"The cells at the early stages of cystogenesis, as judged by the initial recruitment of PODXL at the AMIS, were selected during the analysis of the MatTek dish;"

Line 4: MatTek dish

R. Corrected

Rephrase line 3-5, incorrect grammar

R. Corrected

Line 7: clarify: e.g. add "for retracing purposes" at the end of sentence

R. The text has been implemented with clarifications as requested by the reviewer.

1.3.7.1: Tomography = Electron Tomography

R. Corrected

Two-step CLEM used for first time, not essential, delete "two-step"

R. Corrected

This paragraph should not be called Tomography but immune-CLEM but there is a lot of repetition in this section, needs to be cleaned up.

R. We renamed and partly re-wrote this paragraph (now called Immune-CLEM).

References

1. Miki, H.; Yamaguchi, H.; Suetsugu, S.; Takenawa, T., IRSp53 is an essential intermediate between Rac and WAVE in the regulation of membrane ruffling. *Nature* **2000**, *408* (6813), 732-5.
2. Nakagawa, H.; Miki, H.; Nozumi, M.; Takenawa, T.; Miyamoto, S.; Wehland, J.; Small, J. V., IRSp53 is colocalised with WAVE2 at the tips of protruding lamellipodia and filopodia independently of Mena. *J Cell Sci* **2003**, *116* (Pt 12), 2577-83.
3. Funato, Y.; Terabayashi, T.; Suenaga, N.; Seiki, M.; Takenawa, T.; Miki, H., IRSp53/Eps8 complex is important for positive regulation of Rac and cancer cell motility/invasiveness. *Cancer Res* **2004**, *64* (15), 5237-44.
4. Connolly, B. A.; Rice, J.; Feig, L. A.; Buchsbaum, R. J., Tiam1-IRSp53 complex formation directs specificity of rac-mediated actin cytoskeleton regulation. *Mol Cell Biol* **2005**, *25* (11), 4602-14.
5. Suetsugu, S.; Kurisu, S.; Oikawa, T.; Yamazaki, D.; Oda, A.; Takenawa, T., Optimization of WAVE2 complex-induced actin polymerization by membrane-bound IRSp53, PIP(3), and Rac. *J Cell Biol* **2006**, *173* (4), 571-85.
6. Mattila, P. K.; Pykalainen, A.; Saarikangas, J.; Paavilainen, V. O.; Vihinen, H.; Jokitalo, E.; Lappalainen, P., Missing-in-metastasis and IRSp53 deform PI(4,5)P2-rich membranes by an inverse BAR domain-like mechanism. *J Cell Biol* **2007**, *176* (7), 953-64.
7. Ahmed, S.; Goh, W. I.; Bu, W., I-BAR domains, IRSp53 and filopodium formation. *Semin Cell Dev Biol* **2010**, *21* (4), 350-6.
8. Goh, W. I.; Lim, K. B.; Sudhakaran, T.; Sem, K. P.; Bu, W.; Chou, A. M.; Ahmed, S., mDia1 and WAVE2 interact directly with IRSp53 in filopodia and are involved in filopodium formation. *J Biol Chem* **2011**.
9. Disanza, A.; Bisi, S.; Winterhoff, M.; Milanese, F.; Ushakov, D. S.; Kast, D.; Marighetti, P.; Romet-Lemonne, G.; Muller, H. M.; Nickel, W.; Linkner, J.; Waterschoot, D.; Ampe, C.; Cortellino, S.; Palamidessi, A.; Dominguez, R.; Carlier, M. F.; Faix, J.; Scita, G., CDC42 switches IRSp53 from inhibition of actin growth to elongation by clustering of VASP. *EMBO J* **2013**, *32* (20), 2735-50.
10. Kast, D. J.; Yang, C.; Disanza, A.; Boczkowska, M.; Madasu, Y.; Scita, G.; Svitkina, T.; Dominguez, R., Mechanism of IRSp53 inhibition and combinatorial activation by Cdc42 and downstream effectors. *Nat Struct Mol Biol* **2014**, *21* (4), 413-22.
11. Suetsugu, S.; Murayama, K.; Sakamoto, A.; Hanawa-Suetsugu, K.; Seto, A.; Oikawa, T.; Mishima, C.; Shirouzu, M.; Takenawa, T.; Yokoyama, S., The RAC binding domain/IRSp53-MIM homology domain of IRSp53 induces RAC-dependent membrane deformation. *J Biol Chem* **2006**, *281* (46), 35347-58.
12. Martin-Belmonte, F.; Gassama, A.; Datta, A.; Yu, W.; Rescher, U.; Gerke, V.; Mostov, K., PTEN-mediated apical segregation of phosphoinositides controls epithelial morphogenesis through Cdc42. *Cell* **2007**, *128* (2), 383-97.
13. Kinoshita, R.; Homma, Y.; Fukuda, M., Rab35-GEFs, DENND1A and folliculin differentially regulate podocalyxin trafficking in two- and three-dimensional epithelial cell cultures. *J Biol Chem* **2020**.
14. Klinkert, K.; Rocancourt, M.; Houdusse, A.; Echard, A., Rab35 GTPase couples cell division with initiation of epithelial apico-basal polarity and lumen opening. *Nature communications* **2016**, *7*, 11166.
15. Mrozowska, P. S.; Fukuda, M., Regulation of podocalyxin trafficking by Rab small GTPases in 2D and 3D epithelial cell cultures. *J Cell Biol* **2016**, *213* (3), 355-69.
16. Chou, A. M.; Sem, K. P.; Lam, W. J.; Ahmed, S.; Lim, C. Y., Redundant functions of I-BAR family members, IRSp53 and IRTKS, are essential for embryonic development. *Scientific reports* **2017**, *7*, 40485.
17. Croce, A.; Cassata, G.; Disanza, A.; Gagliani, M. C.; Tacchetti, C.; Malabarba, M. G.; Carlier, M. F.; Scita, G.; Baumeister, R.; Di Fiore, P. P., A novel actin barbed-end-capping activity in EPS-8 regulates apical morphogenesis in intestinal cells of *Caenorhabditis elegans*. *Nat Cell Biol* **2004**, *6* (12), 1173-9.
18. Li, L.; Baxter, S. S.; Zhao, P.; Gu, N.; Zhan, X., Differential interactions of missing in metastasis and insulin receptor tyrosine kinase substrate with RAB proteins in the endocytosis of CXCR4. *J Biol Chem* **2019**, *294* (16), 6494-6505.
19. Bryant, D. M.; Roignot, J.; Datta, A.; Overeem, A. W.; Kim, M.; Yu, W.; Peng, X.; Eastburn, D. J.; Ewald, A. J.; Werb, Z.; Mostov, K. E., A molecular switch for the orientation of epithelial cell polarization. *Dev Cell* **2014**, *31* (2), 171-87.
20. Datta, A.; Bryant, D. M.; Mostov, K. E., Molecular regulation of lumen morphogenesis. *Curr Biol* **2011**, *21* (3), R126-36.
21. McRae, R.; Lapierre, L. A.; Manning, E. H.; Goldenring, J. R., Rab11-FIP1 phosphorylation by MARK2 regulates polarity in MDCK cells. *Cell Logist* **2017**, *7* (1), e1271498.
22. Lim, H.; Yu, C. Y.; Jou, T. S., Galectin-8 regulates targeting of Gp135/podocalyxin and lumen formation at the apical surface of renal epithelial cells. *FASEB J* **2017**, *31* (11), 4917-4927.

23. Strilic, B.; Kucera, T.; Eglinger, J.; Hughes, M. R.; McNagny, K. M.; Tsukita, S.; Dejana, E.; Ferrara, N.; Lammert, E., The molecular basis of vascular lumen formation in the developing mouse aorta. *Dev Cell* **2009**, *17* (4), 505-15.

REVIEWERS' COMMENTS:

Reviewer #1 (Remarks to the Author):

Overall they addressed my concerns with the exception of interaction studies with the endogenous IRSp53/Rab35 proteins. They have done so much additional work addressing the demands of the other 3 Reviewers that I won't insist on this

Reviewer #2 (Remarks to the Author):

Overall authors made a considerable effort to address most of my concerns/suggestions. Consequently, this is much improved manuscript. I do still have few concerns that should be addressed before paper can be published. I know that typically NComms only allow one revision. However, I think that this is pretty interesting manuscript and I would highly recommend that authors should be given a chance to incorporate suggested experiments/edits.

1) This is an interesting and data-heavy manuscript that identifies IRSp53 as a novel regulator of apical lumen formation in renal epithelia. The data demonstrating that IRSp53 is required for lumenogenesis is solid, especially since authors makes an extra effort to test it in couple of different cell lines (Caco2 and MDCK) as well as in two in vivo models (mice and zebrafish). However, the part that attempts to identify the molecular machinery of IRSp53 action is much less convincing, to some extent due to various technical/experimental issues. For example, in vitro overlay assay is not a good assay to analyze protein interactions. Better approaches need to be used, ideally ITC or Biocore, but, at very least, recombinant protein pull down assay.

R. To address this concern we performed a new set of experiments as suggested and confirmed that IRSp53 preferentially interacts with RAB35DN or wild type RAB35 loaded with GDP using in vitro pull-down assays with soluble and purified proteins (Fig. S4D of the revised manuscript)

I appreciate the attempt to demonstrate the binding between Rab35 and IRSp53 using pull-down assays. However, the binding seems pretty weak, especially considering that 2.5 μ M of purified protein was used. Most importantly, the GST is clearly much less than GST-Rab35 (even if authors state that it was loaded the same amount). I wonder what would happen if GST negative control would be loaded in the same amount as GST-Rab35. Typically I would not be as picky, but the preference for GDP-Rab35 is very unusual observation that goes against what we typically think about how Rabs work. That is also a key part of the manuscript and needs to be rigorously demonstrated.

2) The manuscript also has some bigger conceptual issues. First, it is not clear how Rab35 and IRSp53 complex functions if IRSp53 binds to Rab35-GDP. All the work on Rab35 so far indicates that

it functions in GTP-bound form, thus, it is unclear why it would be important to target Rab35-GDP to the AMIS.

R. We completely agree that a key functional aspect of RAB35 in cystogenesis is its activation status. However, the cellular localization of this GTPase along the plasma membrane and specifically at the AMIS is equally relevant and related to its activity. Consistently, here, we showed that: a) IRSp53 and RAB35 are early localized at the AMIS (Fig. S4B of the revised manuscript), before PODXL becomes visible at this structure; b) removal of IRSp53 reduces the accumulation of RAB35 at the luminal, apical site of cysts (Fig. 6B of the revised manuscript). c) we have further shown with new experiments that the removal of IRSp53 also reduces the total levels of RAB35-GTP (Fig. 6C of the revised manuscript). We would also like to point out that IRSp53 through its I-BAR domain preferentially binds to membranes rich in PI4,5P2 phospholipids, which have been shown to accumulate at the luminal side in cystogenesis. Thus, the most parsimonious interpretation of our observation in the context of previously established literature is that IRSp53 facilitates/ restricts the localization of RAB35 ready to be activated by a yet-to-be identified guanine nucleotide exchange factor at the luminal side. Notably, recent work published during the revision of this manuscript points to a specific role of the guanine nucleotide exchange

factor DENND1A in controlling RAB35 activation in 3D cystogenesis

At the luminal initiation site, RAB35 is, thus, expected to undergo cycles of activation and deactivation. In its GTP-bound form, it can directly associate with PODXL, functioning as tethering device for the localized delivery of PODXL-vesicles (as reported in ref.14). Consistent with this model, the removal of IRSp53, which only partially compromises the localization of RAB35, is expected to mimic a partial-loss-of- function phenotype of RAB35. Indeed, the extent of reversed polarity caused by the loss of IRSp53 is less robust as compared to the one resulting from the complete loss of RAB35. Conversely, the partial loss of RAB35 has been shown to result in the formation of multiple lumens as reported (see also below), similar to what we observed in the case of IRSp53 removal. It is also relevant to point out that the multi-luminal phenotype caused by IRSp53 loss is likely the result of additional functions that this protein exerts in lumenogenesis. Indeed, our analysis of the structural organization of the opposing plasma membrane of two daughter cells during cystogenesis revealed that the loss of IRSp53 results in the formation of aberrant cytoplasmic bridges that interconnect the cytoplasm of two-adjacent cells and interrupt the continuity of the plasma membrane at the nascent lumen. The loss of IRSp53 also reduces intercellular space, and occasionally induces the formation of ectopic lumens. These alterations are likely to generate distinct mini-lumens and at PM targeting sites that combined with altered polarized trafficking of podocalyxin carriers leads to the accumulation of podocalyxin in multiple foci, which will eventually evolve into multiple lumens.

I fully agree with authors that IRSp53 is needed for lumenogenesis. That part of the paper is solid. My issue is with author statement that IRSp53 binds preferably to GDP-Rab35 and that this binding somehow translates to recruitment of GTP-Rab35 to the lumen formation site. The proposal that IRSp53 recruits GDP-Rab35 to the lumen formation site, thus, allowing then activation by Rab35 GEFs is interesting one. Unfortunately, authors do not have data supporting this idea (except not very good pull-down assay). Authors, at very least, should show that Rab35-S22N is localized at lumen formation site and that this localization is dependent on IRSp53.

3) The even bigger conceptual issue is the need of Rab35 itself for lumen formation. While a couple years ago Echarde's lab used Rab35 KD to suggest that it is involved in MDCK lumenogenesis, the recent study from Fukuda lab (that used Rab35 KO line) did not see any effect of Rab35 loss on apical lumen formation (the paper is not cited in the manuscript). Authors should generate Rab35 KO line (since Rab35 KD s may have off-target effects) and recapitulate the key experiments. Preferably, that should be followed by GF-Rab35 rescue. I realize that it is a lot of work, but since Rab35 is at the center of this manuscript, the issue whether Rab35 is needed for lumen formation needs to be addressed.

R. We thank the reviewer for pointing this out. Firstly, we would like to underline that we were and are aware of Fukuda work¹⁵ that appeared about at the same time as the study by the Echarde group¹⁴ and of his more recent publication published during the revision of our work¹³. Indeed, Fukuda work had been cited in our previous version of the manuscript.

In all these works, they observe a drastic, but transient inversion of polarity following removal of RAB35 either by complete KO or KD approaches. Notably, however, after acute removal of RAB35 nearly 50 % of the cysts display inverted polarity while about 30% display defects in the distribution of PODXL, which localizes in multiple spots and give rise to multiple lumens at late stage of cysts developments (see Figure 5 from the work by Fukuda laboratory)

Below is shown Figure 6E of the same paper, which reports that the removal of RAB35 results in polarity inversion but also the formation of multiple and aberrant lumens.

The authors are correct that original paper from Fukuda lab (2016) did show that Rab35 KO and KD leads to multiple lumen formation. Strangely, Fukuda lab again published paper in 2019 where they now show that Rab35 KO has no effect on lumen formation. I do realize that it is not authors fault that the same lab published contradictory findings. However, they should point that controversy in the discussion.

Reviewer #3 (Remarks to the Author):

The manuscript was significantly improved and I recommend the publication in Nature Communications.

Reviewer #4 (Remarks to the Author):

Identified as reviewer 4, analysing the technical execution of the imaging I, like the other reviewers, acknowledge the quality of the imaging. Having gone through the revised manuscript I am happy that all my comments were incorporated into the new version and I have no additional comments.

Point-by-point reply to Reviewers

Reviewer points in black, our reply in Blue

We thank all the reviewers for their constructive criticisms and for appreciating our effort and the quality and relevance of our work.

Reviewers #1, 3 and 4 were satisfied by our reply and had no further comments after our revision.

Reviewer #2 had the following remaining minor issues:

1) I appreciate the attempt to demonstrate the binding between Rab35 and IRSp53 using pull-down assays. However, the binding seems pretty weak, especially considering that 2.5 μM of purified protein was used. Most importantly, the GST is clearly much less than GST-Rab35 (even if authors state that it was loaded the same amount). I wonder what would happen if GST negative control would be loaded in the same amount as GST-Rab35. Typically I would not be as picky, but the preference for GDP-Rab35 is very unusual observation that goes against what we typically think about how Rabs work. That is also a key part of the manuscript and needs to be rigorously demonstrated.

R. The reviewer is correct in noticing that the amounts of GST is less than GST-RAB35. Indeed, the pull-down experiments using GST fusion proteins were performed using EQUI-MOLAR amounts (i.e., the same number of moles of the negative GST control and RAB35-GST fusion protein). The GST has a Mw of $\sim 26\text{KDa}$, which is nearly half of the Mw of the RAB35-GST ($\sim 49\text{KDa}$). Thus, using the following standard equation

$$n(\text{number of moles}) = \text{mass}(\mu\text{g}) / \text{Mw}$$

the total amounts in μg that one must use to have equimolar amounts of baits is for the GST control nearly half of that of RAB35-GST fusion proteins. This is exactly what we have done and what can be observed by examining our western blot. (See the fig. S4D snapshot below). Specifically, we used a total of n (number of moles) $\cong 0.625$ nanomoles in 250 μL of volume (a value that corresponds to a concentration of GST fusion proteins of 2.5 μM , as specified in the text).

In the case of GST than we had to use according to the equation above,

$$n = \text{mass} / \text{Mw} \longrightarrow \text{mass} = 0.625 \text{ nano moles} \times 26 \text{ KDa} \cong 1.63 \mu\text{g}$$

In the case of RAB35-GST

$$n = \text{mass} / \text{Mw} \longrightarrow \text{mass} = 0.625 \text{ nano moles} \times 49 \text{ KDa} \cong 3.06 \mu\text{g}$$

which is nearly double of the amount of GST and that it is exactly what we reported in our experiment (shown in Fig. S4D and included below). The amount of GST is equimolar with respect to RAB35-GST.

Having stressed this, we do agree that the interaction between IRSp53 and RAB35 is of a relative low affinity, as we have already emphasized in the text, but the binding of IRSp53 to RAB35, and a preference for RAB35-DN is supported by:

1. In Coimmunoprecipitation IRSp53 binds with relatively low apparent affinity, but specifically and preferentially RAB35-WT under conditions of serum deprivation (Fig. 5)
2. Recombinantly purified IRSp53 binds directly to recombinantly purified RAB35-S22N or RAB35-Q67L, displaying increase apparent, albeit low affinity for RAB35-S22N with respect to RAB35-Q67L in dot blot assays (Supplementary Fig. 4E).
3. Recombinantly purified IRSp53 directly interacts, specifically but weakly, with purified RAB35 through its I-BAR domain (Fig. 5B)
4. A mutant in a set of positively charged amino acids on IRSp53 I-BAR is no longer able to co-immunoprecipitate with RAB35-S22N in vitro as well as in coimmunoprecipitation experiments (Fig. 5C and D).

2) I fully agree with authors that IRSp53 is needed for lumenogenesis. That part of the paper is solid. My issue is with author statement that IRSp53 binds preferably to GDP-Rab35 and that this binding somehow translates to recruitment of GTP-Rab35 to the lumen formation site. The proposal that IRSp53 recruits GDP-Rab35 to the lumen formation site, thus, allowing then activation by Rab35 GEFs is interesting one. Unfortunately, authors do not have data supporting this idea (except not very good pull-down assay). Authors, at very least, should show that Rab35-S22N is localized at lumen formation site and that this localization is dependent on IRSp53.

R. We had performed the required experiment previously. The results indicate that the loss of IRSp53 impairs the luminal localization of RAB35-S22N (the dominant negative form of this GTPase), similar to what we reported for mRFP-RAB35 WT (Fig. 6B). We included this piece of data as Supplementary Figure 4F. However, since the expression of RAB35-S22N, as expected and reported for a "dominant negative" (e.g., Klinkert et al 2016), disrupts the lumenogenesis process and perturbs podocalyxin anchoring to the AMIS, caution is required in interpreting the outcome of this experiment.

3) The authors are correct that original paper from Fukuda lab (2016) did show that Rab35 KO and KD leads to multiple lumen formation. Strangely, Fukuda lab again published paper in 2020 where they now show that Rab35 KO has no effect on lumen formation. I do realize that it is not authors fault that the same lab published contradictory findings. However, they should point that controversy in the discussion.

R. We are glad that the apparent conceptual issue is resolved by a more careful analysis of data published by others. We would also like to stress that the data might not be "entirely" controversial since it is conceivable that the effects of RAB35 interference on lumen formation depends of whether its activity is completely and chronically blocked, as after CRISPR KO, or partially impaired, as after RNAi or IRSp53 KO. Some of these issues have already been discussed and highlighted in a recent review, which we will cite in our revised manuscript. In addition, we explicitly described this point in the main text (in the chapter entitled "IRSp53 is at the core of RAB35 and EPS8 pathways in the control of lumen formation" reported below:

"The RAB GTPases are key regulators of PODXL transcytosis³⁹. Among these, we focused our attention on RAB35. RAB35 is involved in clathrin-mediated endocytosis⁴⁰ and was recently shown to localize early at the AMIS, where it serves as a physical anchor for the targeting of PODXL in MDCK cell cystogenesis^{10,39}. Of note, the complete genetic loss of RAB35 was shown to primarily lead to a complete but transient inversion of polarity accompanied by accumulation of PODXL in intracellular vesicles and only subsequently to the formation of multilumen³². Conversely, a partial reduction of its expression caused the formation of multiple lumen¹⁰. Collectively, these findings indicate that tampering with RAB35 activity impact on cystogenesis, PODXL trafficking and establishment of apico-basal polarity."